# Modelling present-day basal melt rates for Antarctic ice shelves using a parametrization of buoyant meltwater plumes

**Werner M. J. Lazeroms[1], Adrian Jenkins[2], G. Hilmar Gudmundsson[2], and Roderik S. W. van de Wal[1]**

[1]Institute for Marine and Atmospheric Research Utrecht, Utrecht University, Utrecht, The Netherlands
[2]British Antarctic Survey, Natural Environment Research Council, Cambridge, United Kingdom

*Correspondence to:* Werner M. J. Lazeroms (w.m.j.lazeroms@uu.nl)

**Abstract.** Basal melting below ice shelves is a major factor in mass loss from the Antarctic Ice Sheet, which can contribute significantly to possible future sea-level rise. Therefore, it is important to have an adequate description of the basal melt rates for use in ice-dynamical models. Most current ice models use rather simple parametrizations based on the local balance of heat between ice and ocean. In this work, however, we use a recently derived parametrization of the melt rates based on a buoyant meltwater plume travelling upward beneath an ice shelf. This plume parametrization combines a nonlinear ocean temperature sensitivity with an inherent geometry dependence, which is mainly described by the grounding-line depth and the local slope of the ice-shelf base. For the first time, this type of parametrization is evaluated on a two-dimensional grid covering the entire Antarctic continent. In order to apply the essentially one-dimensional parametrization to realistic ice-shelf geometries, we present an algorithm that determines effective values for the grounding-line depth and basal slope in any point beneath an ice shelf. Furthermore, since detailed knowledge of temperatures and circulation patterns in the ice-shelf cavities is sparse or absent, we construct an effective ocean temperature field from observational data with the purpose of matching (area-averaged) melt rates from the model with observed present-day melt rates. Our results qualitatively replicate large-scale observed features in basal melt rates around Antarctica, not only in terms of average values, but also in terms of the spatial pattern, with high melt rates typically occurring near the grounding line. The plume parametrization and the effective temperature field presented here are therefore promising tools for future simulations of the Antarctic Ice Sheet requiring a more realistic oceanic forcing.

## 1 Introduction

The Antarctic Ice Sheet is characterized by vast areas of floating ice at its margins, comprising ice shelves, both large and small, that buttress the outflow of ice from inland. The stability of these ice shelves is governed by a delicate mass balance, consisting of an influx of ice from the glaciers, iceberg calving at the ice front, snowfall and ablation at the surface, and basal melting due to oceanic heat exchange in the ice-shelf cavities. Recent studies suggest that Antarctic ice shelves are experiencing rapid thinning (Pritchard et al., 2009, 2012; Paolo et al., 2015), an effect which can be traced back to an increase in basal melting (Depoorter et al., 2013; Rignot et al., 2013). This is especially apparent in West Antarctica, where relatively warm water from the Amundsen and Bellinghausen seas is able to flow into the ice-shelf cavities and enhance melting from below. Increased basal melt rates and thinning of ice shelves decrease the buttressing effect, enhancing the ice flow and associated mass loss from the Antarctic glaciers and ice sheet. The disintegration of the ice shelves can significantly affect future sea-level rise, as suggested by recent numerical simulations (Golledge et al., 2015; Ritz et al., 2015; DeConto and Pollard, 2016).

In order to correctly predict the evolution of the ice sheet, it is necessary to have accurate models of the dynamics of ice shelves, in which basal melting at the interface between ice and ocean plays an important role. State-of-the-art ice-sheet models for large-scale climate simulations (see e.g. De Boer et al. 2015) provide a complete description of the flow and thermodynamics of ice. However, due to the complex nature of the system and high computational cost of climate simulations, these models inevitably contain approximations and parametrizations of many physical processes, among which basal melting is no exception. In particular, it is challenging to resolve the ocean dynamics within the ice-shelf cavities at

a continental scale, which severely restricts the level of detail possible in basal melt parametrizations. Most recent simulations (e.g. De Boer et al. 2015; DeConto and Pollard 2016) determine the basal melt rate from the local heat flux at the ice-ocean interface (Beckmann and Goosse, 2003), driven by a far-field temperature and a number of tuning factors. Others include a dependence on the thickness of the water column beneath the ice shelf in order to reduce melting near the grounding line (Asay-Davis et al., 2016).

As demonstrated by observational data (e.g. Rignot et al. 2013), the basal melt rates around Antarctica show a complex spatial pattern, which can be inferred to depend heavily on both the geometry of the ice-shelf base and the ocean temperature. It is unlikely that a description of basal melt based on local fluxes at the ice-ocean interface can capture this complex pattern without being either significantly tuned or used in conjunction with extremely detailed ocean-shelf-cavity models. On the other hand, the ocean dynamics and associated melt rates within individual ice-shelf cavities have been studied in rather high detail in recent years. For example, Holland et al. (2008) showed that basal melt rates obtained from a general ocean circulation model respond quadratically to changing ocean temperatures. These studies shed light on the minimal requirements of basal melt parametrizations, i.e. a nonlinear temperature sensitivity, an inherent geometry dependence corresponding to the unresolved ocean circulation, and a depth-dependent pressure freezing point, yielding higher melt rates at greater depths and the possibility of refreezing at lesser depths, closer to the margins of the ice shelves.

Taking these requirements into account, we develop a more advanced parametrization for the basal melt rates, based on the theory of buoyant meltwater plumes, which was first applied to the ice-shelf cavities by MacAyeal (1985). In this theory, it is assumed that the main physical mechanism driving the ocean circulation within the cavity is the positive buoyancy of meltwater, which travels upward beneath the ice-shelf base in the form of a turbulent plume. Melting at the ice-ocean interface is influenced by the fluxes of heat and meltwater through the ocean boundary layer, which depend on the plume dynamics. The upward motion of the plume induces an inflow of possibly warmer ocean water into the ice shelf cavity, creating more melt. Entrainment from the surrounding ocean water affects the momentum and thickness of the plume as it moves up the ice-shelf base. Depending on the stratification of the ocean water inside the cavity, the plume may reach a level of neutral buoyancy from which it is no longer driven upward.

The dynamics of the plume can be captured by a quasi-one-dimensional model of the mass, momentum, heat and salt fluxes within the plume, as shown schematically in Fig. 1. In particular, this work is based on the plume model of Jenkins (1991), from which a basal melt parametrization has recently been derived (Jenkins, 2011, 2014). This parametrization is based on an empirical scaling of the plume

model results in terms of ambient ocean properties and the geometry of the ice-shelf cavity. The geometry dependence is mainly determined by the grounding-line depth and the slope of the ice-shelf base. The aim of this particular study is to apply the plume parametrization to a two-dimensional grid covering all of Antartica, in order to investigate if this type of parametrization is able to give realistic present-day values and capture the complex pattern of basal melt rates shown in observations (Rignot et al., 2013).

In the following section, we describe the details of the plume model and the basal melt parametrization derived from it (Sections 2.1 and 2.2). An important part of the work is the development of an algorithm that translates the parametrization from a one-dimensional to a two-dimensional geometry, as described in Section 2.3. In Section 3.1, we show results from the numerical evaluation of the (still 1-D) parametrization along flow lines of two well-known Antarctic ice shelves, namely Filchner-Ronne and Ross. Finally, Sections 3.2 and 3.3 discuss the application of the 2-D plume parametrization to the entire Antarctic continent, resulting in a two-dimensional map of basal melt rates under the ice shelves. Special attention is given to the construction of an effective ocean temperature field from observations by inversion of the modelled basal melt rates. The results are compared with those from simple heat-balance models (Beckmann and Goosse, 2003; DeConto and Pollard, 2016).

## 2 Modelling basal melt

In this section, we start with a description of the basic physics underlying basal melt models. We summarize the quasi-one-dimensional plume model of (Jenkins, 1991) and the development of the plume parametrization (Jenkins, 2011, 2014) resulting from this model, as shown in previous work. The main contribution of the current study is the method used to extend this plume parametrization to two-dimensional input data, necessary for use in a 3-D ice-sheet–ice-shelf model.

First of all, we briefly discuss a common feature of many basal melt parametrizations, namely the dependence on the local balance of heat at the ice-ocean interface. In its simplest form, this is a balance between the latent heat of fusion and the heat flux through the sub-ice-shelf boundary layer, which can be expressed as follows (Holland and Jenkins, 1999; Beckmann and Goosse, 2003):

$$\rho_i \dot{m} L = \rho_w c_w \gamma_T (T_a - T_f), \tag{1a}$$

where $\rho_i, \rho_w$ are the densities of ice and ocean water, respectively, $\dot{m}$ is the melt rate, $L$ is the latent heat of fusion for ice, $c_w$ is the specific heat capacity of ocean water, $\gamma_T$ is a turbulent exchange velocity and $T_a$ is the temperature of the ambient ocean water. In this model, the melting is driven by the difference between $T_a$ and the depth-dependent freezing

point,

$$T_f = \lambda_1 S_w + \lambda_2 + \lambda_3 z_b, \tag{1b}$$

where $S_w$ is salinity of the ocean water, $z_b$ is the depth of the ice-shelf base, and $\lambda_1, \lambda_2, \lambda_3$ are constant parameters. As explained by Holland and Jenkins (1999), more details can be included in this basal melt model, e.g. heat conduction into the ice and a balance equation for salinity (see also Section 2.1). Nevertheless, many ice models contain basal melt parametrizations based on Eqs. (1) (see e.g. De Boer et al. 2015; DeConto and Pollard 2016). These models typically use either constant or temperature dependent values for $\gamma_T$, leading to a melt rate that depends either linearly or quadratically on the temperature difference $T_a - T_f$. The latter case is consistent with the findings of Holland et al. (2008), who obtained a similar quadratic relationship from the output of an ocean general circulation model applied to the ice-shelf cavities. The non-linearity arose because the exchange velocity $\gamma_T$ in Eq. (1a) was expressed as a linear function of the ocean current driving mixing across the boundary layer, which is itself a function of the thermal driving. Holland et al. (2008) further explain how this non-linear temperature dependence is related to the input of meltwater with an associated decrease in salinity and increase in buoyancy.

Hence, the exchange velocity plays an important role in correctly determining the heat balance at the ice-ocean interface, or, more precisely, the heat transfer through the ocean boundary layer beneath the ice shelves. However, a local heat-balance model as expressed by (1) is too simplistic to capture the effects of the ocean circulation on the basal melting, e.g. those depending on the ice-shelf geometry. The plume model and parametrization discussed in the remainder of this section are considered the next step in modelling the physics for general ice-shelf geometries without having to rely on full ocean circulation models, for which there are also insufficient input data to obtain a universal Antarctic solution.

## 2.1 Plume model

The parametrization used in this study is based on the plume model developed by Jenkins (1991). Here we summarize the key assumptions and physics behind this model. The ice-shelf cavity is modelled by a two-dimensional geometry (Fig. 1), in which the ice-shelf base has a (local) slope given by the angle $\alpha$. This geometry is assumed to be uniform in the direction perpendicular to the plane and constant in time and can be seen as a vertical cross-section along a flow line of the ice shelf. We can define a coordinate $X$ along the ice-shelf base with slope $\alpha$ and consider the development of a meltwater plume initiating at the grounding line ($X = 0$) and moving up along the ice-shelf base due to positive buoyancy with respect to the ambient ocean water.

The situation depicted in Fig. 1 essentially yields a two-layer system of the meltwater plume with varying thickness

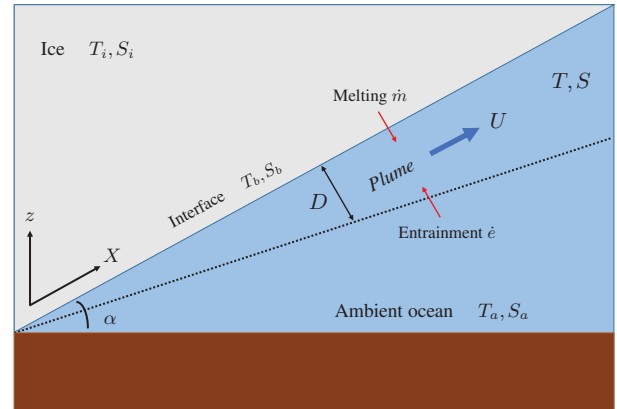

**Figure 1.** Schematic picture of the plume model. The plume travels upward under the ice-shelf base along the path $X$ with speed $U$ and thickness $D$ while being influenced by melting and entrainment. Note that, in general, the slope angle $\alpha$ can vary in the direction of $X$.

$D$, velocity $U$, temperature $T$ and salinity $S$ lying above the ambient ocean with temperature $T_a$ and salinity $S_a$. As explained in Jenkins (1991), the typically small values of the slope angle $\alpha$ allow us to consider conservation of mass, momentum, heat and salt within the plume in a depth-averaged sense. Moreover, as the plume travels upward in the direction of $X$, it is affected by entrainment (at rate $\dot{e}$) of ambient ocean water, as well as the fluxes of meltwater (with melt rate $\dot{m}$) and heat at the ice-ocean interface (with temperature $T_b$ and salinity $S_b$). These considerations yield the following quasi-one-dimensional system of equations for $(D, U, T, S)$ as a function of the coordinate $X$ along the shelf base, denoting the balance of mass, momentum, heat and salt within the plume:

$$\frac{\mathrm{d}DU}{\mathrm{d}X} = \dot{e} + \dot{m}, \tag{2a}$$

$$\frac{\mathrm{d}DU^2}{\mathrm{d}X} = D\frac{\Delta\rho}{\rho_0}g\sin\alpha - C_d U^2, \tag{2b}$$

$$\frac{\mathrm{d}DUT}{\mathrm{d}X} = \dot{e}T_a + \dot{m}T_b - C_d^{1/2}\Gamma_T U(T - T_b), \tag{2c}$$

$$\frac{\mathrm{d}DUS}{\mathrm{d}X} = \dot{e}S_a + \dot{m}S_b - C_d^{1/2}\Gamma_S U(S - S_b), \tag{2d}$$

where $g$ is the gravitational acceleration, $C_d$ is the (constant) drag coefficient, $\Delta\rho = \rho_a - \rho$ is the difference in density between plume and ambient ocean, and $C_d^{1/2}\Gamma_T, C_d^{1/2}\Gamma_S$ are the turbulent exchange coefficients (Stanton numbers) of heat and salinity at the ice-ocean interface. The above formulation makes explicit the linear dependence of the turbulent exchange velocities on the ocean current ($\gamma_T = C_d^{1/2}\Gamma_T U$, $\gamma_S = C_d^{1/2}\Gamma_S U$). The system of equations (2) is closed using suitable expressions for the entrainment rate $\dot{e}$, an equation of state $\rho = \rho(T, S)$, the balance of heat and salt at the ice-ocean interface and the liquidus condition. The expres-

sion for the entrainment rate is assumed to have the following form (Bo Pedersen, 1980):

$$\dot{e} = E_0 U \sin\alpha, \tag{3}$$

with $E_0$ a dimensionless constant. Hence, the entrainment rate increases linearly with the plume velocity, is zero for a horizontal ice-shelf base, and grows with increasing slope angle. Furthermore, a linearized equation of state yields:

$$\frac{\Delta\rho}{\rho_0} = \beta_S(S_a - S) - \beta_T(T_a - T), \tag{4}$$

where $\beta_S$ is the haline contraction coefficient and $\beta_T$ the thermal expansion coefficient. The boundary conditions at the ice-ocean interface are given by:

$$C_d^{1/2}\Gamma_T U(T - T_b) = \dot{m}\left(\frac{L}{c_w} + \frac{c_i}{c_w}(T_b - T_i)\right), \tag{5a}$$

$$C_d^{1/2}\Gamma_S U(S - S_b) = \dot{m}(S_b - S_i), \tag{5b}$$

$$T_b = \lambda_1 S_b + \lambda_2 + \lambda_3 z_b \tag{5c}$$

i.e. the first equation balances the turbulent exchange of heat with heat conduction and latent heat of fusion $L$ in the ice, where $c_w$ and $c_i$ are the specific heat capacities of ocean water and ice, respectively, and $T_i$ is the ice temperature. Similarly, Eq. (5b) is a balance between turbulent exchange of salt and diffusion into the ice. Eq. (5c) is the (linearized) liquidus condition that puts the interface temperature equal to the pressure freezing point at the local depth $z_b$ of the ice-shelf base, equivalent to Eq. (1b).

Equations (2)-(5) form a closed set that can be solved to obtain the prognostic variables $(D, U, T, S)$ of the plume as a function of the plume path $X$, given the ice-shelf draft $z_b(X)$ with slope angle $\alpha(X)$, the ambient ocean properties $T_a(z)$ and $S_a(z)$, and the ice properties $T_i$ and $S_i$. Of particular interest for the current work, however, are the ice-ocean interface conditions (5), which essentially determine the melt rate $\dot{m}$, the key quantity of this study. In other words, the melt rate is determined by the fluxes of heat and salt at the interface, which in turn are linked to the development of the plume. Note that these boundary conditions can be simplified (McPhee, 1992; McPhee et al., 1999) to only two equations containing the freezing temperature $T_f$ of the plume, rather than the interface properties $T_b$ and $S_b$:

$$C_d^{1/2}\Gamma_{TS} U(T - T_f) = \dot{m}\left(\frac{L}{c_w} + \frac{c_i}{c_w}(T_f - T_i)\right), \tag{6a}$$

$$T_f = \lambda_1 S + \lambda_2 + \lambda_3 z_b, \tag{6b}$$

where $C_d^{1/2}\Gamma_{TS}$ is an effective heat exchange coefficient. This simplified formulation can be used together with the prognostic equations (2) by substituting $T_b$ with $T_f$ in (2c) (note that $T_b$ and $T_f$ are not necessarily equal), whereas $S_b$ disappears from the problem by substituting (5b) in (2d).

Strictly speaking, Eq. (6) is only valid after assuming a constant ratio $\Gamma_T/\Gamma_S$ of the exchange coefficients, as explained by Jenkins et al. (2010), who also show that both Eqs. (5) and Eqs. (6) give similar results when used to describe basal melt rates under Ronne Ice Shelf. Also note the similarity between Eqs. (6) and the simple melt model described by Eqs. (1), the difference being the inclusion of heat conduction and the parametrization $\gamma_T = C_d^{1/2}\Gamma_{TS} U$ as well as the plume variables $T$ and $S$ instead of ambient ocean properties. Hence, the turbulent exchange in this model is directly determined by the plume velocity that appears as a prognostic variable.

Without giving further details, we mention that the plume model described above can be evaluated for different ice-shelf geometries (i.e. vertical cross-sections along flow lines) and different vertical temperature and salinity profiles of the ambient ocean (Jenkins, 2011, 2014). In this model, the general physical mechanism governing the development of the plume is the addition of meltwater at the ice-ocean interface, which increases its buoyancy. Changes in buoyancy affect plume speed and that, combined with its temperature and salinity, determines the subsequent input of meltwater.

## 2.2    Basal melt parametrization along a flow line

Evaluating the aforementioned plume model for different geometries and ocean properties leads to a wide variety of solutions for the basal melt rates. The question arises whether there exists an appropriate scaling with external parameters that combines these results into a universal melt pattern. Here we will summarize how such a scaling can be found, leading to the basal melt parametrization of Jenkins (2014) for the quasi-one-dimensional geometries along flow lines described in the previous section; more details can be found in Appendix A. It is important to note that the following derivation is based on simple geometries with a constant basal slope and constant ambient ocean properties, though the resulting parametrization can easily be applied to more general cases, as shown in Section 3.1. Section 2.3 will discuss the extension of this parametrization to more realistic two-dimensional geometries.

The basal melt parametrization used in this study consists of a general expression for a dimensionless melt rate $\hat{M}$ as a function of the dimensionless coordinate $\hat{X}$ measured from the grounding line (Fig. 2). This dimensionless coordinate is essentially the vertical distance of the ice-shelf base from the grounding line, scaled by a temperature- and geometry-dependent length scale $l$:

$$\hat{X} = \frac{z_b - z_{gl}}{l}, \qquad l = f(\alpha) \cdot \frac{T_a - T_f(S_a, z_{gl})}{\lambda_3}, \tag{7}$$

where $z_{gl}$ is the grounding-line depth and $f(\alpha)$ a slope-dependent factor. Hence, $\hat{X} = 0$ corresponds to the grounding line and any shelf point downstream from the grounding line corresponds to a value $0 < \hat{X} < 1$ depending on $T_a$, $S_a$, $z_{gl}$ and $\alpha$. This scaling also implies that the ice-shelf front

is not necessarily located at $\hat{X} = 1$, but its location is highly dependent on the input variables. Similarly, the melt rate is scaled as follows:

$$\hat{M} = \frac{\dot{m}}{M}, \qquad M = M_0 \cdot g(\alpha) \cdot [T_a - T_f(S_a, z_{gl})]^2 \qquad (8)$$

with a different slope-dependent factor $g(\alpha)$ and a constant parameter $M_0$. The dimensionless curve $\hat{M}(\hat{X})$ in Fig. 2 is now defined by polynomial coefficients that were found empirically from the plume model results (Jenkins 2014; Appendix A). In summary, to obtain the basal melt rate $\dot{m}$ at any point beneath the ice-shelf, one requires the local depth $z_b$, local slope $\alpha$, grounding-line depth $z_{gl}$ and ambient ocean properties $T_a$ and $S_a$ to calculate $\hat{X}$ and find the corresponding value on the dimensionless curve $\hat{M}(\hat{X})$, which then has to be multiplied by the physical scale given in (8) (see Appendix A for details). The physical quantities and constant parameters required for evaluating the parametrization are summarized in Table 1.

Although the scaling defined by (7) and (8) is found in a purely empirical way, it is possible to derive the various factors analytically, as sketched in Appendix A. The empirical procedure and the physical meaning of the different factors are outlined in the following. A general solution to the problem is challenging to find as there are at least four length scales that determine the plume evolution (Jenkins, 2011). The first governing length scale is associated with the pressure dependence of the freezing point that imposes an external control on the relationship between plume temperature, plume salinity and the melt rate. Lane-Serff (1995) discussed how this length scale, $(T_a - T_f)/\lambda_3$, approximately determines the distribution of melting and freezing beneath an ice shelf. Jenkins (2014) extended the analysis of Lane-Serff (1995) by making the transition point between melting and freezing dependent on the ice-shelf basal slope, resulting in the length scale (7) with slope factor $f(\alpha)$.

The second of these four length scales is associated with the ambient stratification, which determines how far the plume can rise before reaching a level of neutral buoyancy. Magorrian and Wells (2016) discuss the plume behaviour and resulting melt rates when this length scale dominates. Critically, with the pressure dependence of the freezing point assumed to be negligible, as required in the analysis of Magorrian and Wells (2016), no freezing can occur. The third length scale can be formulated by comparing the input of buoyancy from freshwater outflow at the grounding line with the input of buoyancy by melting at the ice-ocean interface (Jenkins, 2011). This length scale indicates the size of the zone next to the grounding line where the impact of ice shelf melting on plume buoyancy can be ignored and conventional plume theory (Morton et al., 1956; Ellison and Turner, 1959) applied, and is generally small compared with typical ice shelf dimensions. The final length scale is that at which the Coriolis force takes over from friction as the primary force balancing the plume buoyancy in the momentum budget. Jenkins

(2011) discussed these length scales in the context of which would take over as the dominant control on plume behaviour beyond the initial zone near the grounding line where the initial source of buoyancy dominates, and showed the length scale associated with the pressure dependence of the freezing point, $(T_a - T_f)/\lambda_3$, to be most important for typical ice-shelf conditions.

Hence, we obtain the second factor of the length scale $l$ in (7) used in the parametrization. However, this length scale contains two more ingredients. First, as discussed by Jenkins (2011), the entrainment rate in the mass conservation equation (2a) explicitly depends on the slope $\alpha$, whereas the melt rate is only affected indirectly, so there is a geometrical factor that scales the elevation of the plume temperature above the local freezing point:

$$\frac{E_0 \sin \alpha}{C_d^{1/2} \Gamma_{TS} + E_0 \sin \alpha}. \qquad (9)$$

This factor gives rise to the slope dependence $f(\alpha)$ in $l$, which is essentially an empirically derived scaling of the transition point between melting and freezing (Appendix A). The second ingredient is related to the coefficient $\Gamma_{TS}$, which appears in $f(\alpha)$ through the simplified interface conditions (6). Jenkins (2014) retained the more complex melt formulation (5) in the plume model while seeking empirical scalings based on an effective $\Gamma_{TS}$. As discussed by Holland and Jenkins (1999), the factor relating $\Gamma_T$ and $\Gamma_{TS}$ is itself a function of the plume temperature, so Jenkins (2014) expressed the effective $\Gamma_{TS}$ as an empirical function of $\Gamma_T$, $T_a - T_f$ and (9) including a constant initial value $\Gamma_{TS0}$ (see Appendix A). When distance along the plume path is scaled with this slightly more complex factor (see Eq. (A10)), the melt rates produced by the plume model conform to a universal form, first rising to a peak at $\hat{X} \approx 0.2$, before falling and transitioning to freezing at $\hat{X} \approx 0.56$ (Fig. 2).

With the distance along the plume path appropriately scaled, all that remains is to scale the amplitude of the melt rate curves produced by the plume model and find the melt rate scale $M$ in (8). As in Jenkins (2011) the appropriate physical scales are: 1) the temperature of the ambient ocean water relative to the freezing point; 2) the factor in Eq. (9) scaling the temperature elevation of the plume above freezing; 3) a factor that scales the plume speed, given by the ratio of plume buoyancy to frictional drag:

$$\left( \frac{\sin \alpha}{C_d + E_0 \sin \alpha} \right) \left( \frac{C_d^{1/2} \Gamma_{TS}}{C_d^{1/2} \Gamma_{TS} + E_0 \sin \alpha} \right). \qquad (10)$$

The second term in parenthesis is the factor that scales the plume temperature relative to the ambient temperature and thus controls plume buoyancy. It replaces the initial buoyancy flux at the grounding line used in the scaling of Jenkins (2011). The final expression includes factors and powers that are derived empirically (though some theoretical arguments

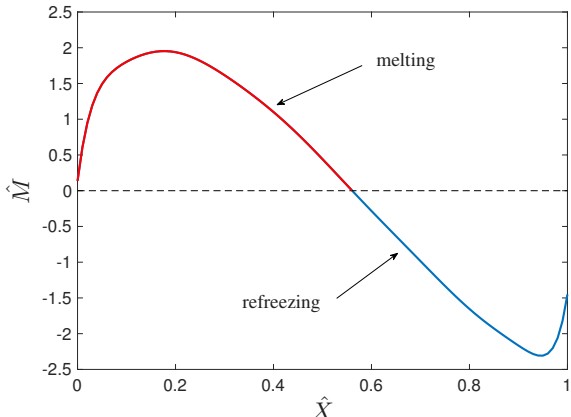

**Figure 2.** Dimensionless melt curve $\hat{M}(\hat{X})$ used in the basal melt parametrization. Higher melt rates typically occur close to the grounding line with a maximum at $\hat{X} \approx 0.2$. A transition from melting (red) to refreezing (blue) may occur further away from the grounding line, depending on the position of the ice front. Note that the value of $\hat{X}$ depends on the distance to the grounding line, as well as the temperature difference $T_a - T_f$ and the local slope $\alpha$ (see Appendix A). In other words, $\hat{X} = 0$ corresponds to the grounding line, but the dimensionless position of the ice-shelf front depends on the length scale and is not necessarily equal to $\hat{X} = 1$.

can be applied, cf. Appendix A), giving rise to the form of $M$ with slope factor $g(\alpha)$ in (8). In summary, the result of this scaling procedure is an approximately universal melt rate curve, which can then be represented by a single polynomial expression that is accurate to about 20% for melt rates ranging over many orders of magnitude (Jenkins, 2014).

### 2.3 Basal melt parametrization in 2-D: effective plume path

As explained in the previous section, an important feature of the basal melt parametrization is its dependence on non-local quantities, in particular the grounding-line depth $z_{gl}$ from which the plume originated. Therefore, in order to apply the parametrization to realistic geometries, one needs to know for each ice-shelf point the corresponding grounding-line point(s) serving as the origin of the plume(s) reaching that particular shelf point. For the quasi-one-dimensional settings considered so far, this is not an issue, since the plume can only travel in one direction. However, for general ice-shelf cavities, an arbitrary shelf point can be reached by plumes from multiple directions, corresponding not only to different values for $z_{gl}$, but also to different slope angles $\alpha$. This means that the plume parametrization cannot be directly applied to such geometries. An algorithm is needed to determine effective values for $z_{gl}$ and $\alpha$. The development of this algorithm is the main focus of the current work and discussed below.

As a starting point, we consider the usual topographic data in terms of two-dimensional fields for the ice thickness $H_i$, bedrock / seabed elevation $H_{\text{bed}}$ and elevation of the upper

ice surface $H_s$ used by ice-dynamical models. The following algorithm is valid for any topographic data on a rectangular grid with any resolution $\Delta x \times \Delta y$. First of all, the topographic data are used to define an ice mask based on the criterion for floating uniform ice, as shown in Table 2. Furthermore, the depth of the ice base is determined to be:

$$z_b = H_s - H_i. \tag{11}$$

In order to apply the basal melt parametrization to this two-dimensional data, effective values for $z_{gl}$ and $\alpha$ must be determined for every *ice-shelf* point $(i, j)$ with basal depth $z_b(i, j)$, where the indices $i$ and $j$ denote the position on the grid. This is done by first searching for "valid" grounding-line points in 16 directions on the grid, starting from *any* shelf point $(i, j)$, as depicted in Fig. 3a. Note that we can calculate a local basal slope $s_n(i, j)$ at the point $(i, j)$ in the $n$-th direction as follows:

$$s_n(i, j) = \frac{z_b(i, j) - z_b(i + i_n, j + j_n)}{\sqrt{(i_n \Delta x)^2 + (j_n \Delta y)^2}}, \tag{12}$$

where $(i_n, j_n)$ denotes a direction vector on the grid, i.e. $(i_n, j_n) = (1, 0)$ denotes right, $(i_n, j_n) = (0, 1)$ denotes up, etc., and $\Delta x$ and $\Delta y$ denote the horizontal grid size in the $x$- and $y$-direction, respectively. To determine whether a grounding-line point found in one of the 16 directions is valid for the calculation of the basal melt, the following two criteria are applied:

1. Assuming that a buoyant meltwater plume can only reach the point $(i, j)$ from the $n$-th direction if the basal slope in that direction is positive, the algorithm only searches in directions for which $s_n(i, j) > 0$.

2. If the first criterion is met for the $n$-th direction, the algorithm searches in this direction for the nearest *ice-sheet* point. More precisely, the associated direction vector $(i_n, j_n)$ is added to the grid indices and the mask value in the resulting point is checked. This process is repeated until either an ice-sheet point, an ocean point or the domain boundary is encountered. An ice-sheet point found in this way is only considered to be a valid grounding-line point if it lies deeper than the original ice-shelf point at $(i, j)$, assuming again that a buoyant meltwater plume from the grounding line can only go up. The second criterion then becomes $z_n(i, j) < z_b(i, j)$, where $z_n(i, j)$ is the grounding-line depth in the $n$-th direction.

Note, however, that in determining the second criterion, the depth difference between the encountered sheet point and the adjacent shelf point can be considerable, especially for coarser resolutions. In such cases, the algorithm tries to obtain a better estimate of the true grounding-line depth in this direction, $z_n(i, j)$, by interpolating along either the bed or the ice base, as shown in Fig. 3b and c. The two cases shown

**Table 1.** Physical quantities and constant parameters serving as input for the basal melt parametrization.

| External quantities | | Units |
|---|---|---|
| $z_b$ | Local depth of ice-shelf base | m |
| $\alpha$ | Local slope angle | — |
| $z_{gl}$ | Depth of grounding line | m |
| $T_a$ | Ambient ocean temperature | °C |
| $S_a$ | Ambient ocean salinity | psu |
| Constant parameters | | Values |
| $E_0$ | Entrainment coefficient | $3.6 \times 10^{-2}$ |
| $C_d$ | Drag coefficient | $2.5 \times 10^{-3}$ |
| $C_d^{1/2}\Gamma_T$ | Turbulent heat exchange coefficient | $1.1 \times 10^{-3}$ |
| $\lambda_1$ | Freezing point-salinity coefficient | $-5.73 \times 10^{-2}$ °C |
| $\lambda_2$ | Freezing point offset | $8.32 \times 10^{-2}$ °C |
| $\lambda_3$ | Freezing point-depth coefficient | $7.61 \times 10^{-4}$ K m$^{-1}$ |
| $M_0$ | Melt rate parameter | 10 m yr$^{-1}$ °C$^{-2}$ |
| $C_d^{1/2}\Gamma_{TS0}$ | Heat exchange parameter | $6.0 \times 10^{-4}$ |
| $\gamma_1$ | Heat exchange parameter | 0.545 |
| $\gamma_2$ | Heat exchange parameter | $3.5 \times 10^{-5}$ m$^{-1}$ |

in these figures account for either a positive or a negative basal slope beyond the grounding line. One should note that this additional step assumes the grounding line to be located halfway between the sheet and shelf points, which could be improved by more sophisticated interpolation techniques.

Following the above procedure yields for each ice-shelf point $(i,j)$ a set of grounding-line depths $z_n$ and local slopes $s_n$ in the directions that are "valid" according to the aforementioned two criteria. Mind that not all directions may yield a (valid) grounding-line point, in particular those towards the open ocean. Now, in order to determine the *effective grounding-line depth* $z_{gl}(i,j)$ and *effective slope angle* $\alpha(i,j)$ necessary for calculating the basal melt in the shelf point $(i,j)$, we simply take the average of the values found for $z_n$ and $s_n$:

$$z_{gl}(i,j) = \frac{1}{N_{ij}} \sum_{\text{valid } n} z_n(i,j), \tag{13a}$$

$$\tan[\alpha(i,j)] = \frac{1}{N_{ij}} \sum_{\text{valid } n} s_n(i,j), \tag{13b}$$

where $N_{ij}$ denotes the number of valid directions found for the shelf point $(i,j)$. On the other hand, if no valid values for $z_n$ and $s_n$ are found for a particular shelf point, we take $z_{gl} = z_b$ and $\alpha = 0$, leading to zero basal melt in that point (see Appendix A).

In summary, the method described above yields two-dimensional fields for the effective grounding-line depth $z_{gl}$ and effective slope $\tan(\alpha)$, given topographic data in terms of $H_i$, $H_s$ and $H_{bed}$ and a suitable ice mask, such as the one defined in Table 2. These fields, in turn, serve as input for the basal melt parametrization described in the previous section, together with appropriate data for the ocean

**Table 2.** Definition of the ice mask. The ice-shelf criterion is that for uniform ice with density $\rho_i$ floating on ocean water with density $\rho_w$. The minimum ice thickness used here is $H_{i,\min} = 2$ m.

| Mask value | Type | Criterion |
|---|---|---|
| 0 | ice sheet | $(\rho_i/\rho_w)H_i > -H_{\text{bed}}$ |
| 1 | ice shelf | $(\rho_i/\rho_w)H_i \leq -H_{\text{bed}}$ |
| 2 | ocean / no ice | $H_i \leq H_{i,\min}$ |

temperature $T_a$ and salinity $S_a$ (discussed in Section 3.2). We thus obtain a complete method for calculating the basal melt for all Antarctic ice shelves, given the topography and ocean properties, which can also be used in conjunction with ice-dynamical models. In the following, however, we use the Bedmap2 dataset (Fretwell et al., 2013) to define the present-day topography of Antarctica and disregard the ice dynamics. More specifically, the original Bedmap2 data is remapped to a rectangular grid with grid size $\Delta x = \Delta y = 20$ km, using the mapping package OBLIMAP 2.0 (Reerink et al., 2016). The resulting topographic data can be used as input for the algorithm described here, leading to the fields for $z_{gl}$ and $\tan(\alpha)$ shown in Fig. 4, which are used for the basal melt calculations discussed in Section 3. Note that the mask in Fig. 4a does not exactly match the Bedmap2 mask because a constant $\rho_i$ was used in formulation of Table 2 as is common in many ice sheet models. This might cause discrepancies in the position of the grounding line, which, however, are likely compensated by the rather coarse resolution. In Fig. 4b one can see that the lowest values of $z_{gl}$ are obtained towards the inland regions of Filchner-Ronne ice shelf and Amery ice shelf. The values for the local slope are typically high near

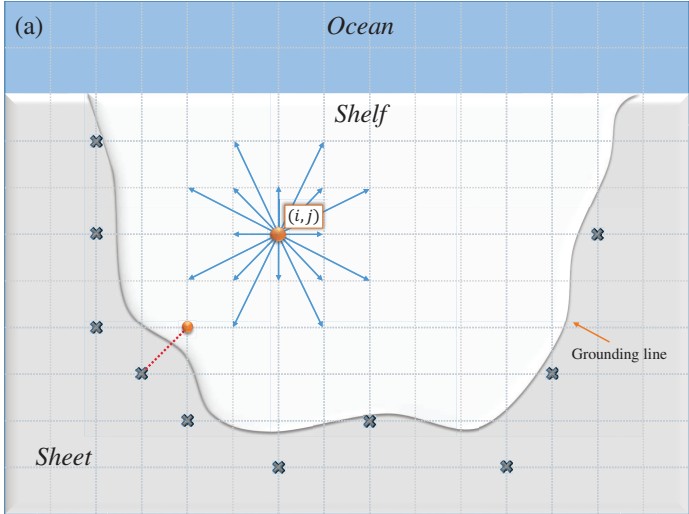

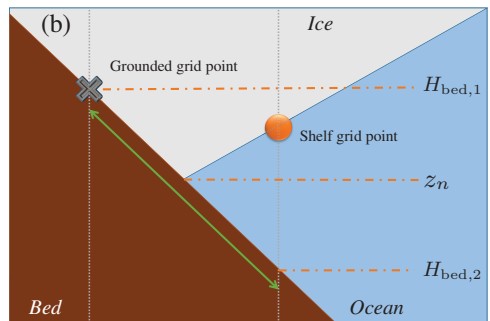
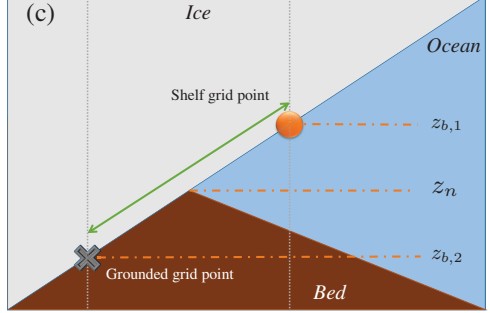

**Figure 3.** Schematic of the algorithm for finding the average grounding-line depth and associated slope angle used by the basal melt parametrization. (a) Top view of an ice-shelf on a horizontal grid. The algorithm searches in 16 directions on the grid from the shelf point $(i,j)$. Possible grounded points found in this way are marked by $\times$. (b) Vertical slice along the $n$-th direction (e.g. the red dotted line in (a)). If the grounded point is higher than the previous shelf point, the grounding-line depth $z_n$ is found by interpolation along the bed $(z_n = \frac{1}{2}(H_{\text{bed},1} + H_{\text{bed},2}))$. (c) Interpolation along the ice base if the grounded point in the $n$-th direction is deeper than the previous shelf point $(z_n = \frac{1}{2}(z_{b,1} + z_{b,2}))$.

the grounding line and in some places also near the ice front, as shown in Fig. 4c.

One should note that, although we attempt to directly translate the concept of a quasi-1-D plume to a multitude of plumes in two dimensions, there are important physical effects not taken into account by this approach. Most importantly, a realistic two-dimensional plume has an additional degree of freedom because it also develops in the cross-flow direction, causing the width to be a dynamic variable in addition to the thickness $D$. This can have significant consequences for the mass budget currently described by (2a). Hattermann (2012) explored the possibility of adding a variable plume width to the original plume model and Hattermann et al. (2014) showed that such a 2-D formulation improves the prediction of melt rates for a realistic ice-shelf geometry compared to the 1-D model. Although this appears to be an important extension of the plume model that should be taken into account, the aim of the current work is to explore the capabilities of the original 1-D plume parametrization in pre-

dicting melt rates around Antarctica. The current approach is meant to be a simple method to parametrize the net circulation within an ice-shelf cavity as the average effect of multiple plumes, in order to be applied around the entire ice sheet. Further extensions for obtaining a 2-D plume model are beyond the scope of this work.

## 3 Results

Here we present various results obtained by evaluating the basal melt parametrization described in the previous section. First, we investigate the main characteristics of the original 1-D parametrization of Section 2.2 by evaluating it along flow lines of the Filchner-Ronne and Ross ice shelves. In Sections 3.2 and 3.3, we turn to the full 2-D geometry of Antarctica using the algorithm described in Section 2.3, first by constructing an appropriate effective ocean temperature field from observational data.

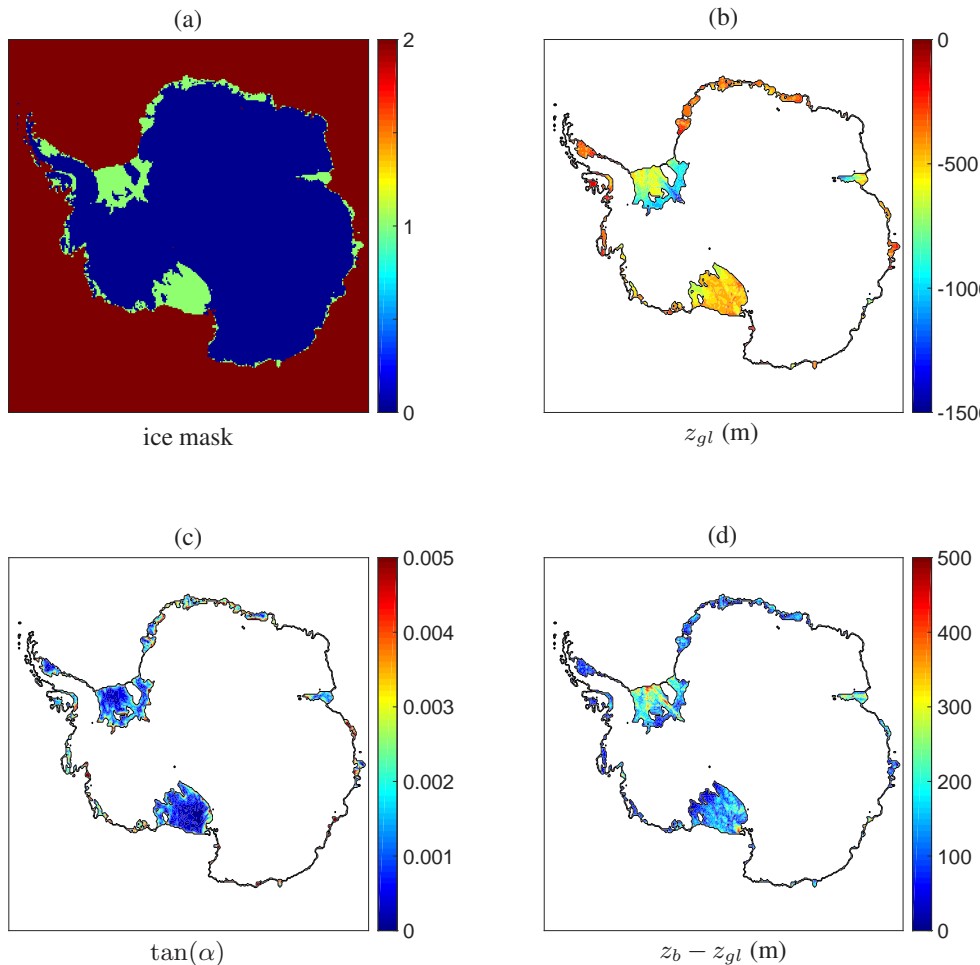

**Figure 4.** Effective plume paths under the Antarctic ice shelves as calculated by the algorithm of Section 2.3 using the Bedmap2 topographic data remapped on a 20 km by 20 km grid. (a) Ice mask according to Table 2. (b) The effective grounding-line depth $z_{gl}$. (c) The effective slope $\tan(\alpha)$. (d) The difference between local ice-base depth and associated grounding-line depth, $z_b - z_{gl}$.

### 3.1 Comparison of basal melt parametrizations along flow lines

Topographic data along flow lines for both Filchner-Ronne ice shelf (FRIS) and Ross ice shelf are taken from Bombosch and Jenkins (1995) and Shabtaie and Bentley (1987), respectively. This data can be used to determine the quantities $z_b$, $\alpha$ and $z_{gl}$ necessary for calculating the basal melt with the parametrization of Section 2.2. Furthermore, we define a uniform ambient ocean temperature $T_a = -1.9\,°\text{C} + \Delta T$, where $\Delta T$ is varied between runs, and a constant ambient ocean salinity $S_a = 34.65$ psu. The results of these calculations are shown in Fig. 5 and compared with those of the full plume model described in Section 2.1. Moreover, we compare with two simple basal melt parametrizations based on Eqs. (1), namely the linear (i.e. in $T_a - T_f$) parametrization by Beckmann and Goosse (2003) with constant $\gamma_T$ and the quadratic parametrization by DeConto and Pollard (2016) with $\gamma_T = \kappa_T |T_a - T_f|$. Apart from the values listed in Ta-

ble 1, additional model parameters used for these calculations are given in Table 3.

Fig. 5 shows that both the current parametrization and the original plume model yield approximately the same melt-rate patterns as a function of the horizontal distance from the grounding line. These patterns roughly correspond to the dimensionless melt curve in Fig. 2, i.e. maximum melt near the grounding line and possibly refreezing further away along the flow line. This is most apparent in Fig. 5c, which shows a transition from melting to freezing, since the relatively deep draft of FRIS allows higher values of the dimensionless coordinate $\hat{X}$. On the other hand, Fig. 5d does not show refreezing because the draft of Ross ice shelf is much shallower. Increasing the ocean temperature (through $\Delta T$) can significantly enhance basal melt and remove the area of refreezing, as shown in Figs. 5e and 5f. In these cases, additional melt peaks occur in regions of high basal slope. Moreover, although the general agreement is good, the discrepancies between the current parametrization and the plume model are

**Table 3.** Additional model parameters used for evaluating the plume model and the simple parametrizations described in Section 3.1. BG2003 refers to Beckmann and Goosse (2003) and DCP2016 refers to DeConto and Pollard (2016).

| Constant parameters | | Values |
|---|---|---|
| $L$ | Latent heat of fusion for ice | $3.35 \times 10^5$ J kg$^{-1}$ |
| $c_w$ | Specific heat capacity of water | $3.974 \times 10^3$ J kg$^{-1}$ K$^{-1}$ |
| $c_i$ | Specific heat capacity of ice | $2.009 \times 10^3$ J kg$^{-1}$ K$^{-1}$ |
| $\beta_S$ | Haline contraction coefficient | $7.86 \times 10^{-4}$ |
| $\beta_T$ | Thermal expansion coefficient | $3.87 \times 10^{-5}$ K$^{-1}$ |
| $g$ | Gravitational acceleration | $9.81$ m s$^{-2}$ |
| $\rho_i$ | Density of ice | $9.1 \times 10^2$ kg m$^{-3}$ |
| $\rho_w$ | Density of ocean water | $1.028 \times 10^3$ kg m$^{-3}$ |
| $\gamma_T$ | Turbulent exchange velocity (BG2003) | $5.0 \times 10^{-7}$ m s$^{-1}$ |
| $\kappa_T$ | Turbulent exchange coefficient (DCP2016) | $5.0 \times 10^{-7}$ m s$^{-1}$ K$^{-1}$ |

largest when the basal slope changes rapidly, because the parameterization responds immediately to the change while the full model has an inherent lag as the plume adjusts to the new conditions. On the whole, we see that the melt patterns given by the plume parametrization can be quite complex, while the two simple parametrizations give nearly constant curves (i.e. independent of the position with respect to the grounding line).

It is interesting to investigate the temperature sensitivity of the four models in terms of the horizontally averaged melt rate as a function of $\Delta T$, as shown in Figs. 5g and 5h. In the case of FRIS, the plume model and parametrization are much more sensitive to the ocean temperature than the two simpler models. However, the average melt rates for Ross ice shelf are rather similar for all four models and all values of $\Delta T$. Hence, the difference in the temperature sensitivity depends significantly on the ice-shelf geometry, where the plume parametrization appears to have a larger potential for capturing diverse melt values than the simpler models. Note that in both cases, the temperature dependence of the plume parametrization is slightly nonlinear, similar to the DeConto and Pollard (2016) parametrization, while the Beckmann and Goosse (2003) parametrization has a linear temperature dependence. Following the discussion of Holland et al. (2008), the temperature dependence of the plume parametrization should therefore be more realistic than the one of Beckmann and Goosse (2003). However, the quadratic parametrization of DeConto and Pollard (2016) tends to significantly underestimate the melt rates as well, despite its nonlinearity. It appears that the geometry dependence of the plume parametrization is an important factor for the temperature sensitivity of the calculated basal melt rates. In Section 3.3 we show that these geometrical effects are indeed crucial for obtaining realistic melt rates with the 2-D parametrization, but first we discuss the matter of determining a suitable input field for the ocean temperature.

## 3.2 Effective ocean temperature

The previous section dealt with the 1-D basal melt parametrization along a flow line using a uniform ambient ocean temperature for the entire ice-shelf cavity. While a uniform temperature might appear a reasonable first approximation for a single ice shelf, it is far from realistic to apply a single ocean temperature for multiple ice shelves around the entire Antarctic continent. Therefore, in order to apply the parametrization to the 2-D geometry defined by Fig. 4, a suitable 2-D field for the ocean temperature $T_a$ is required. In principle, the same is true for the salinity $S_a$, but we will assume that the horizontal variations in ocean salinity around Antarctica are so small that the pressure freezing point $T_f$ is only affected by variations in depth. In the following, we will therefore take a uniform salinity $S_a = 34.6$ psu. One should realize that vertical variations in $S_a$, which are not accounted for in the current parametrization, would be important in reality, as discussed in Section 4.

Two problems arise when considering a 2-D ocean temperature field for forcing the parametrization. First of all, such a field should ideally be based on observational data, but ocean temperature measurements in the Antarctic ice-shelf cavities are sparse. A more feasible approach would be to compute an interpolated field based on ocean temperature data in the surrounding ocean, which inevitably contains artefacts resulting from the non-uniform and predominantly summertime sampling. Secondly, even if a complete dataset of ocean temperatures were available, it is not immediately clear which temperatures (i.e. at which depth) are characteristic for the ocean water reaching the grounding lines (e.g. Jenkins et al. 2010). In principle, detailed knowledge of the bottom topography and the ocean circulation would be required for this, which goes beyond the scope of the current modelling approach.

In view of these issues, we construct an *effective ocean temperature* field with which the current plume parametrization yields melt rates that are as close as possible to present-day observations, averaged over entire ice shelves. In other words, this can be regarded as the inverse problem of com-

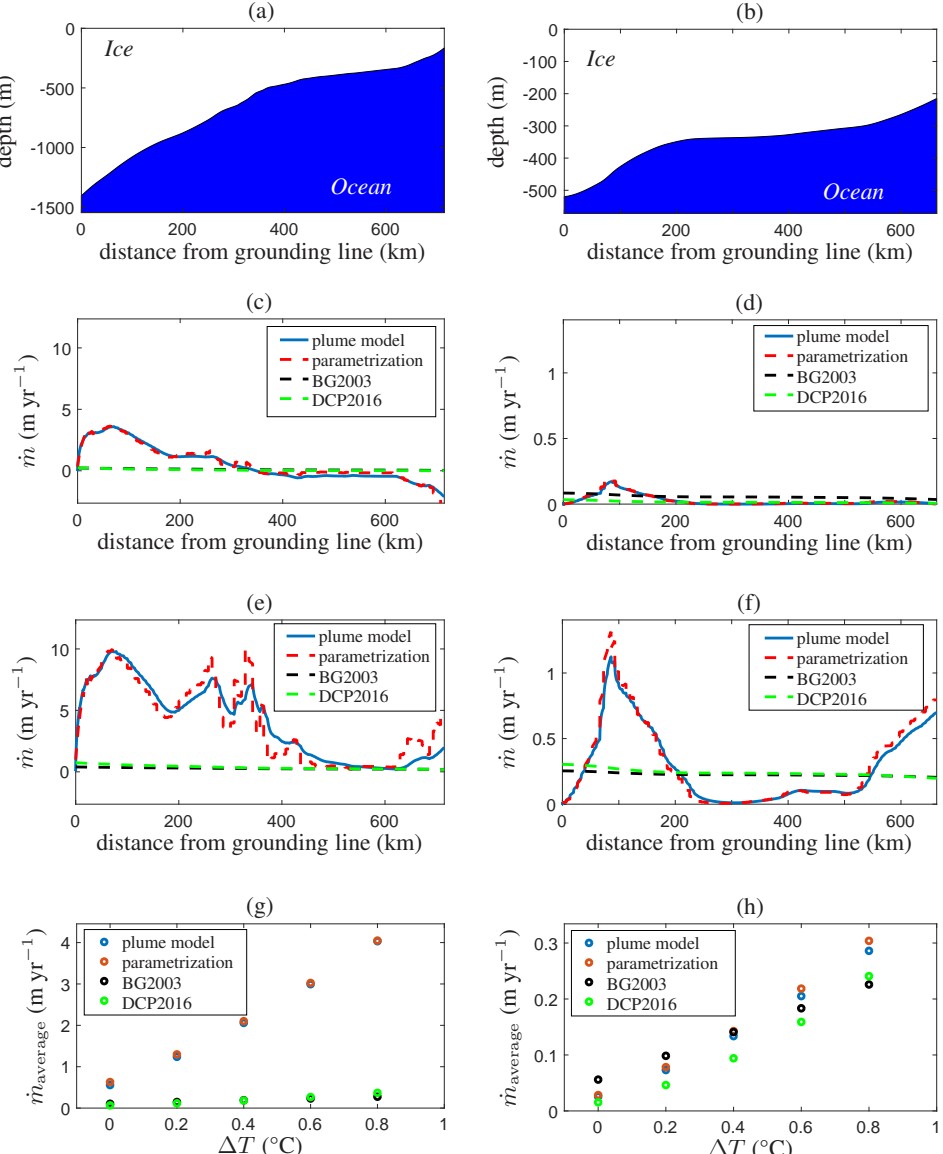

**Figure 5.** Comparison of the plume model (Section 2.1) with the 1-D basal melt parametrization (Section 2.2) , as well as the parametrizations of Beckmann and Goosse (2003) (BG2003) and DeConto and Pollard (2016) (DCP2016), for flow lines along Filchner-Ronne ice shelf (left column) and Ross ice shelf (right column), both with uniform ocean temperature $T_a = -1.9 \,°C + \Delta T$ and constant salinity $S_a = 34.65$ psu. (a,b) Geometry of the ice-shelf base. (c,d) Melt pattern for $\Delta T = 0 \,°C$. (e,f) Melt pattern for $\Delta T = 0.8 \,°C$. (g,h) Melt rates average along the flow line as a function of $\Delta T$. Note that the black curve is nearly identical to the green curve and might appear below it. Also note the difference in vertical scale between the left and right columns. The flow-line locations are indicated in Fig. 6.

puting the unknown ocean temperatures by assuming that the model output for the melt rates matches the (averaged) observations. For this purpose, we use the results of Rignot et al. (2013), who calculated the area-averaged melt rates for each
5 Antarctic ice shelf, based on a combination of observational data and regional climate model output for the different terms in the local ice-shelf mass balance. Other datasets for recent Antarctic basal melt rates exist (e.g. Depoorter et al. 2013), as well as more recent data for ice-shelf thinning (Paolo et al.,
10 2015) from which the basal melt rates can be calculated when

combined with the other terms in the mass balance (e.g. velocity and surface melt rates). These alternative datasets for the (area-averaged) basal melt rates are expected to be at least of the same order of magnitude, which we deem sufficient for the purpose of the current study. Since it is impossible to re- 15 solve each individual ice shelf from the Rignot et al. (2013) dataset with the currently used 20-km resolution (Fig. 4), we consider a set of 13 ice-shelf groups and determine the area-averaged basal melt for each group from the data of Rignot et al. (2013). The definition of these groups along with the 20

calculated average melt rates are shown in Fig. 6. Note that the shelves have been grouped based on their geographical location, but also for more practical reasons such as the possibility of distinguishing their boundaries on the 20-km grid.

As a starting point for constructing the effective ocean temperature, we consider the observational data of the World Ocean Atlas 2013 (WOA13, Locarnini et al. 2013), which contains a global dataset of (annual mean) ocean temperatures within a range of depths $(0 - 5500 \text{ m})$. Restricting ourselves to the temperature data for latitudes south of 60°S, we average the ocean temperatures over depth intervals $[z_1, z_2]$, where $z_1$ is the level of the bed (i.e. the deepest level for which data is available) with the additional constraint $z_1 \geq -1000 \text{ m}$, and $z_2 = \min\{0, z_1 + 400 \text{ m}\}$. This results in a relatively smooth 2-D temperature field containing an inherent dependence on the bottom topography, which can be considered a first estimate for the ocean water flowing into the ice-shelf cavities. The depth-averaged temperature field is now remapped on the same 20-km grid as the topography data (see Section 2.3 and Fig. 4) and interpolated using natural-neighbour interpolation (i.e. a weighted version of nearest-neighbour interpolation, giving smoother results) to obtain data in the entire domain of interest. The resulting temperature field, called $T_0$, is shown in Fig. 7a. One should note that both the depth-averaging and the interpolation procedures introduce biases in the resulting field. In particular, the rather simple interpolation technique also interpolates ocean temperatures between ice-shelf cavities separated by the continent or grounded ice, which is not realistic as it propagates temperatures into cavities that the corresponding ocean water cannot reach. Using the natural-neighbour interpolation method appears to limit these effects. However, the details of the resulting field $T_0$ are somewhat arbitrary as it needs to be adjusted in order to obtain melt rates that agree with the data of Rignot et al. (2013).

The aim is now to modify this depth-averaged, interpolated temperature field $T_0$ in such a way that the basal melt parametrization yields melt rates close to those shown in Fig. 6 for each ice-shelf group. As explained earlier, this modification is necessary for eliminating biases in $T_0$ caused by the sparse observations and numerical interpolation, and also because the flow dynamics of the ocean are not resolved. The field $T_0$ is now modified by adding a 2-D field of temperature differences $(\Delta T)$, which, in turn, is the result of *linearly interpolating* individual values of $\Delta T$ in 29 carefully chosen sample points, with $\Delta T = 0$ on the domain boundary. The sample points and values of $\Delta T$ have been determined by trial and error and are certainly not a unique nor optimal configuration. The points are mainly located in regions that are most affected by interpolation between strictly separated cavities (e.g. grounding line of FRIS) or extrapolation of warm open-ocean temperatures into cavities (e.g. Dronning Maud Land, shelf groups 2 and 3 in Fig. 6). The resulting effective temperature field, $T_{\text{eff}} = T_0 + \Delta T$, is shown in Fig. 7b, which also indicates the positions of the afore-

mentioned sample points along with the used values of $\Delta T$ in these points. Note that for technical reasons explained in Appendix A, we have applied a lower limit to the effective temperature equal to the pressure freezing point at surface level. With the current choice $S_a = 34.6$ psu, this implies $T_{\text{eff}} \geq -1.9$ °C. Comparing Figs. 7a and b, we see that the main effect of $\Delta T$ is a decrease in the ocean temperature over most of the continental shelf and most ice-shelf cavities (in particular for Ross and Amery ice shelves), and a slight increase in the ocean temperature in West Antarctica and some regions in East Antarctica (e.g. shelf group 6 in Fig. 6). Again, note that the details in the procedure for calculating $T_0$ and $\Delta T$ are somewhat arbitrary, since increasing one term would require decreasing the other term in order to obtain similar values for $T_{\text{eff}}$ with similar basal melt rates.

Fig. 8 shows the basal melt rates computed by the parametrization using the effective temperature $T_{\text{eff}}$ of Fig. 7b as forcing. An area-averaged value is obtained for each of the 13 ice-shelf groups in Fig. 6 and compared with the observational values from the Rignot et al. (2013) data. By construction, the modelled basal melt rates correspond closely to the observational values and fall within the error estimates. A notable exception is the value for Filchner-Ronne ice shelf (FRIS), which is $0.32 \pm 0.08 \text{ m yr}^{-1}$ according to the observations, whereas the parametrization gives a value just above $0.5 \text{ m yr}^{-1}$. This discrepancy is caused by the lower bound of $-1.9$ °C imposed on the effective temperature, whereas in reality the temperatures can reach values below $-2.0$ °C (e.g. Nicholls et al. 2009). As we can see in Fig. 7b, the ocean water below FRIS is almost entirely at this minimum temperature, making it impossible to further improve the basal melt rate without using unfeasibly low values for $T_{\text{eff}}$. This rather technical constraint might be relaxed in various ways, as briefly discussed in Appendix A, possibly improving the melt rates in very cold cavities.

Nevertheless, the plume parametrization in conjunction with the constructed effective temperature field appears to yield realistic present-day melt rates for all shelf groups. By construction, the effective temperature shown in Fig. 7b contains an inherent dependence on the bottom topography, with typically lower temperatures above the continental shelves (and thus in the ice-shelf cavities), while still retaining the spatial variation in temperature of the surrounding deep ocean (e.g. higher temperatures for West-Antarctica, leading to higher melt rates for ice-shelf groups 11 and 12 as defined in Fig. 6).

### 3.3 Comparison of 2-D melt-rate patterns

The effective grounding-line depth and effective slope in Fig. 4, the effective ocean temperature in Fig. 7b and the assumption $S_a = 34.6$ psu constitute the full set of input parameters necessary for evaluating the plume parametrization on the entire 2-D geometry. The resulting 2-D field of basal melt rates under all Antarctic ice shelves is shown in Fig. 9a

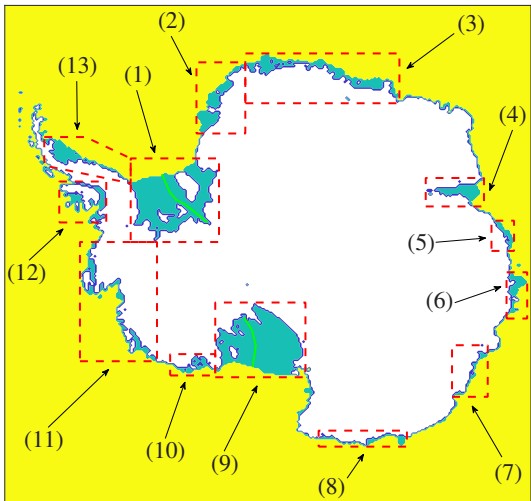

| Ice-shelf group | | Average basal melt (m yr$^{-1}$) |
|---|---|---|
| 1 | *Filchner, Ronne* | $0.32 \pm 0.08$ |
| 2 | *Stancomb, Brunt, Riiser-Larsen, Quar, Ekström, Atka* | $0.2 \pm 0.1$ |
| 3 | *Jelbart, Fimbul, Vigrid, Nivl, Lazarev, Borchgrevink, Baudoin, Prince Harald, Shirase* | $0.5 \pm 0.1$ |
| 4 | *Amery, Publications* | $0.6 \pm 0.4$ |
| 5 | *West* | $1.7 \pm 0.7$ |
| 6 | *Shackleton, Tracy, Tremenchus, Conger* | $2.7 \pm 0.5$ |
| 7 | *Totten, Moscow University, Holmes* | $7.1 \pm 0.5$ |
| 8 | *Mertz, Ninis, Cook East, Rennick, Lillie* | $1.7 \pm 0.4$ |
| 9 | *Ross* | $0.12 \pm 0.07$ |
| 10 | *Sulzberger, Swinburne, Nickerson, Land* | $1.5 \pm 0.2$ |
| 11 | *Getz, Dotsen, Crosson, Thwaites, Pine Island, Cosgrove, Abbot, Venable* | $5.6 \pm 0.3$ |
| 12 | *Stange, George VI, Bach, Wilkins* | $3.0 \pm 0.4$ |
| 13 | *Larsen B-C-D-E-F-G* | $0.5 \pm 0.6$ |

**Figure 6.** The 13 groups of ice shelves used for constructing the effective ocean temperature field. Average melt rates and error estimates (one standard deviation) for each group are calculated from the data of Rignot et al. (2013) for individual ice shelves. Green lines indicate the approximate positions of the flow lines used in Fig. 5.

(note that this is the same data used for the area-averaged melt rates in Fig. 8, but now plotted as a spatial field rather than averaged values over the ice shelves). A general pattern that can be observed, especially on the bigger ice shelves, consists of regions of higher melt close to the grounding line and lower melt or patches of refreezing closer to the ice front, the latter being most apparent at the ice fronts of shelf groups 1, 2 and 9. This pattern is a consequence of the underlying plume model, as shown in Section 3.1 for data along a flow line. Moreover, the highest melt rates occur in West Antarctica (shelf groups 11 and 12) and some specific shelves in East Antarctica (shelf groups 6 and 7), where the constructed effective temperature is significantly higher than elsewhere. The general melt patterns within individual cavities appear to be in line with observations, e.g. Rignot et al. (2013). However, one should note that the Rignot et al. (2013) melt pattern shows a greater spatial variability, with more patches of (stronger) refreezing occurring between patches of melting

(Fig. 11a). Especially beneath FRIS and Ross ice shelf, the melt pattern appears quite complex and local deviations from the general pattern can be considerable (Fig. 11b). These discrepancies in the current parametrization might have different reasons, such as the coarse resolution or the fact that we disregard the details of the ocean circulation within the ice-shelf cavities, as well as effects due to the Coriolis force and both seasonal and vertical variability in the temperature and salinity fields.

Furthermore, Figure 9 shows the melt rates patterns of the plume parametrization zoomed in on three regions, giving more insight in the orders of magnitude of the highest melt rates. The high near-grounding-line melt rates for FRIS have values between 1 and 10 m yr$^{-1}$, while those for Ross ice shelf appear one order of magnitude smaller. On the other hand, the West Antarctic melt rates shown in Fig. 10b have values around 10 m yr$^{-1}$ or more due to the higher ocean temperatures here. It should be noted, however, that the latter

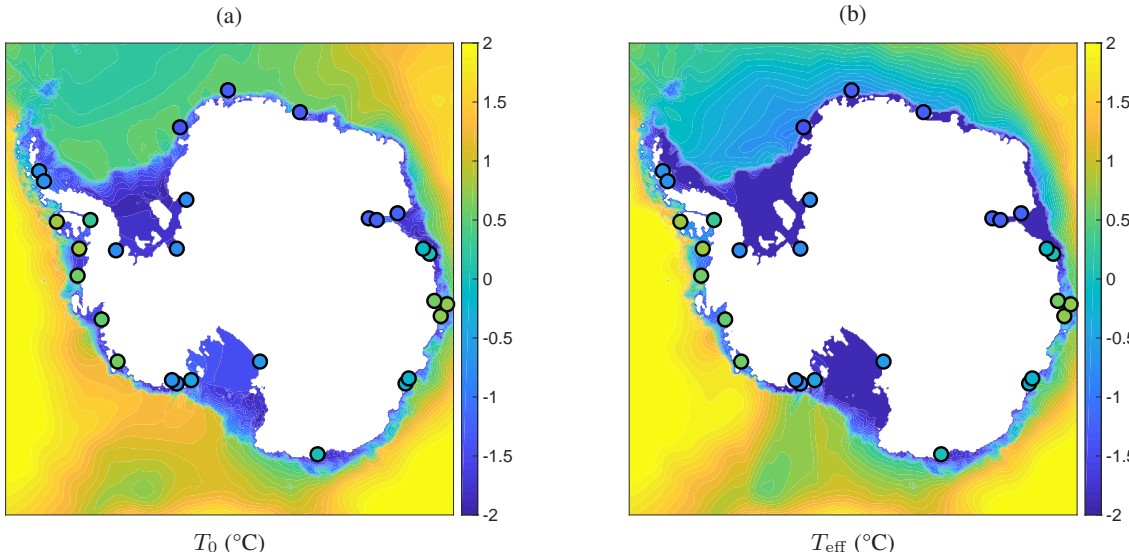

**Figure 7.** (a) Depth-averaged and interpolated ocean temperature, $T_0$, calculated from annual mean WOA13 data. (b) Effective ocean temperature $T_{\mathrm{eff}} = \max\{T_0 + \Delta T, -1.9\}$ constructed from $T_0$ as described in Section 3.2. The circles indicate the positions of the sample points in which the values of $\Delta T$ are imposed. The colour of each circle corresponds to the imposed value of $\Delta T$ (same colour scale), ranging from $-1.4\ ^\circ$C to $0.8\ ^\circ$C. The full $\Delta T$ field is obtained by linearly interpolating these values.

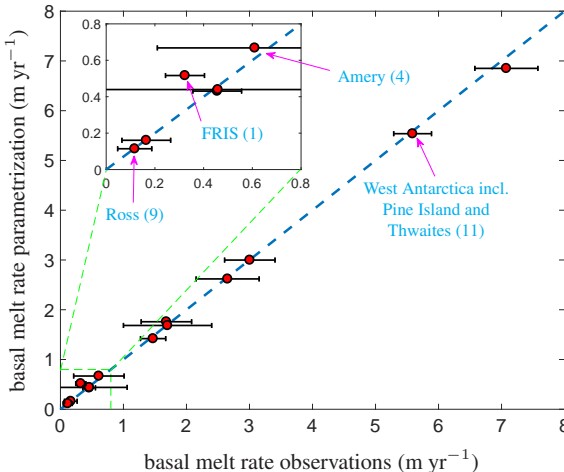

**Figure 8.** Area-averaged basal melt rates for each ice-shelf group in Fig. 6 obtained with the plume parametrization and the effective temperature field of Fig. 7b. The modelled melt rates are plotted against the averaged observational values given in Fig. 6. For four important shelf groups, the data points are explicitly labelled along with the corresponding group number in Fig. 6. The horizontal error bar is one standard deviation uncertainty in the observations.

values are still lower than those observed in the Rignot et al. (2013) data, where local melt rates close the grounding line can reach 100 m yr$^{-1}$, while the average melt rates over the full area of Pine Island and Thwaites are 16.2 m yr$^{-1}$ and 17.7 m yr$^{-1}$, respectively.

For comparison, we also evaluate the quadratic parametrization of DeConto and Pollard (2016), described in

Section 3.1, using the same geometric data and the effective temperature field of Fig. 7b as input. The resulting basal melt rate pattern is shown in Fig. 9b. Comparing this figure to Fig. 9a shows that the quadratic parametrization yields significantly lower melt rates than the plume parametrization, at least with the current effective temperature as input. The only visible patches of basal melt are located in the aforementioned regions where the ocean temperature is high, as well as near the grounding line of Filchner-Ronne ice shelf. Therefore, if the effective temperature in Fig. 7b is indeed characteristic of the true temperatures in the ice-shelf cavities, the quadratic parametrization would require significant tuning in order to obtain a similar agreement with observed melt rates as currently found with the plume parametrization. For completeness, we mention that the linear parametrization of Beckmann and Goosse (2003) yields even lower melt rates due to its low temperature sensitivity, as discussed in Section 3.1.

To further clarify the differences between the two parametrizations in Fig. 9, we have repeated the steps outlined in Section 3.2 and constructed a second effective temperature field based on the quadratic parametrization by De-Conto and Pollard (2016) instead of the plume parametrization. The resulting temperature field is shown in Fig. 12a. Note that the difference between this field and the one in Fig. 7b only lies in the values chosen for $\Delta T$ and not in the underlying interpolated observations ($T_0$). For simplicity, the $\Delta T$ values have been imposed in the same sample points as used for Fig. 7b. Comparing the two effective temperature fields in Figs. 7b and 12a shows that much higher ocean temperatures are required for the quadratic parametrization

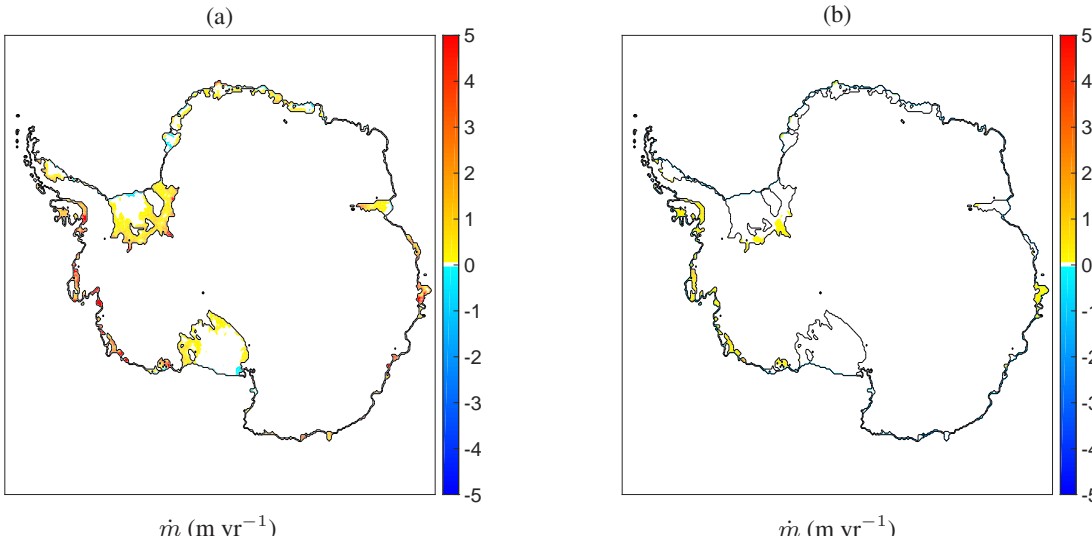

**Figure 9.** Basal melt rates in meter per year with the Bedmap2 topographic data and the effective temperature field of Fig. 7b as obtained from: (a) the plume parametrization with additional input parameters from Fig. 4; (b) the quadratic parametrization of DeConto and Pollard (2016).

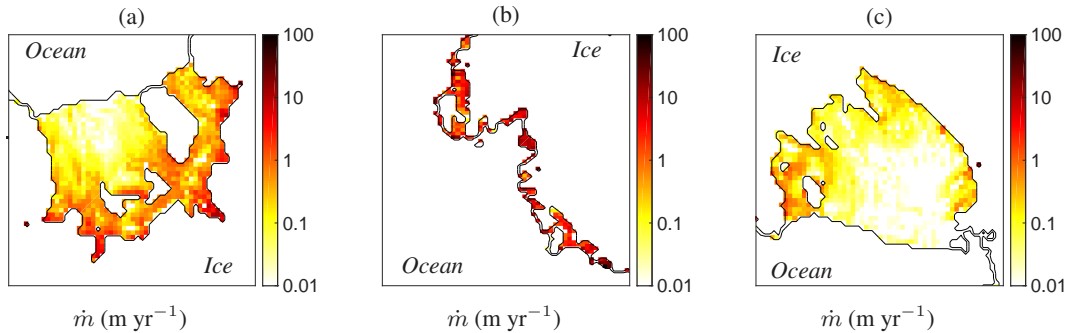

**Figure 10.** As Fig. 9a, but with a logarithmic color scale (negative and zero values shown white) and zoomed in on (a) Filcher-Ronne ice shelf (group 1), (b) West Antarctica including Pine Island and Thwaites (group 11), (c) Ross ice shelf (group 9).

to give realistic area-averaged melt rates. The $\Delta T$ values imposed in the sample points indicated in Fig. 12a range from $-0.5$ °C to $5.4$ °C, while those used for Fig. 7b range from $-1.4$ °C to $0.8$ °C. Furthermore, we can calculate the root mean square values of $T_{\mathrm{eff}} - T_0$ over the entire domain (disregarding the continental points), yielding $0.3$ °C for Fig. 7b and $1.1$ °C for Fig. 12a. Hence, the effective temperature in Fig. 7b lies closer to the underlying observational data $T_0$ than the field in Fig. 12a.

The basal melt rates resulting from the quadratic parametrization and the new effective temperature field are shown in Fig. 12b. Clearly, the higher ocean temperatures cause significantly higher melt rates than those shown in Fig. 9b. However, compared with the plume parametrization in Fig. 9a, the spatial distribution of these melt rates is more uniform, showing less prominent melt peaks near grounding lines and no patches of refreezing. It appears that the quadratic temperature dependence together with the (slight)

depth dependence through the pressure freezing point $T_f$ (equation (1b)) is not sufficient for obtaining realistic melt rates without significantly increasing the input ocean temperature, which can be considered equivalent to using different tuning factors for different ice shelves. On the other hand, the plume parametrization, containing an additional geometry dependence through the grounding-line depth and local slope, appears to yield the required melt rates rather naturally with ocean temperatures constructed in a plausible way, and it results in a more realistic spatial pattern with highest basal melt rates near the grounding line as well as areas of refreezing.

## 4   Discussion

The plume parametrization in combination with the 2-D algorithm of Section 2.3 and the effective temperature field of

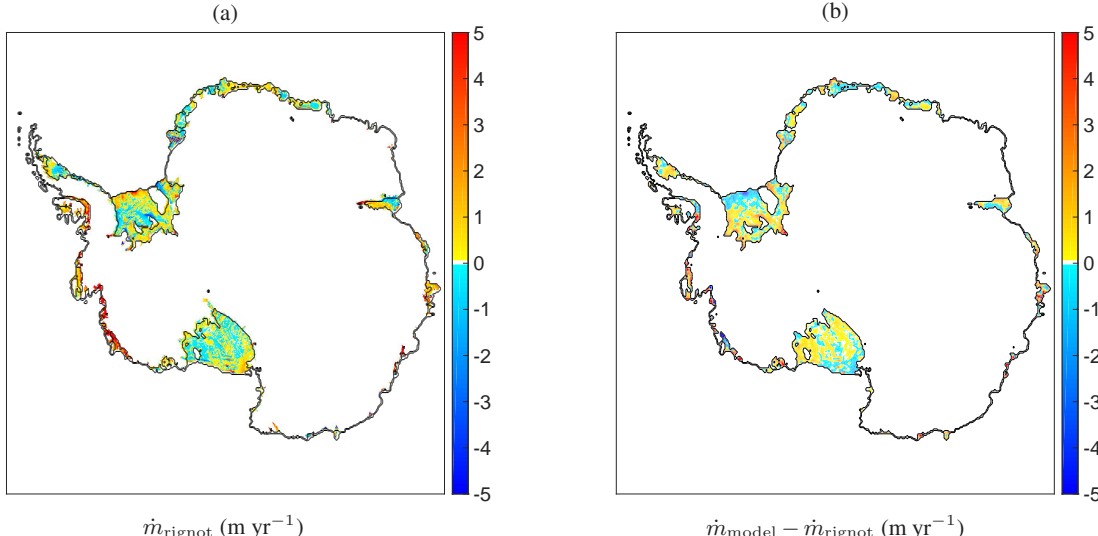

$\dot{m}_{\mathrm{rignot}}$ (m yr$^{-1}$)                $\dot{m}_{\mathrm{model}} - \dot{m}_{\mathrm{rignot}}$ (m yr$^{-1}$)

**Figure 11.** Basal melt rates in meter per year extracted from the Rignot et al. (2013) observational dataset (courtesy of Dr Jeremie Mouginot): (a) raw data plotted together with the currently used mask; (b) difference between the plume parametrization (Fig. 9a) and the observations interpolated on the 20-km grid.

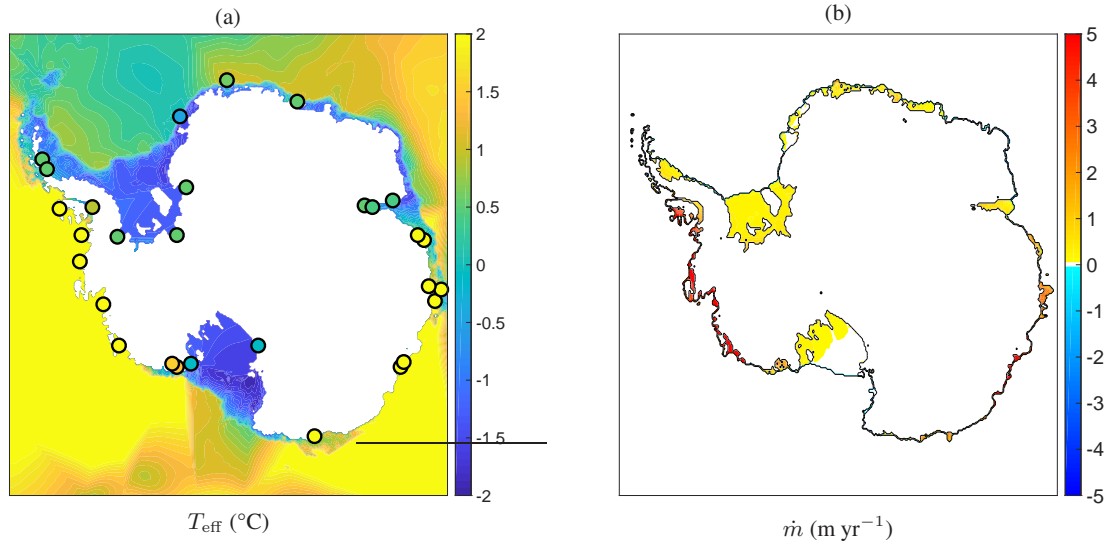

$T_{\mathrm{eff}}$ (°C)                            $\dot{m}$ (m yr$^{-1}$)

**Figure 12.** (a) Effective temperature field constructed in a similar way as Fig. 7b, but with different values for $\Delta T$ (indicated by the circles and ranging from $-0.5$ °C to 5.4 °C), chosen in order to match the melt rates of the quadratic parametrization of DeConto and Pollard (2016) with the data of Rignot et al. (2013). (b) Basal melt rates obtained with the quadratic parametrization of DeConto and Pollard (2016) using the Bedmap2 topographic data and the effective temperature in (a) as input.

Section 3.2 is able to capture a more complex spatial pattern of basal melt rates and a high temperature sensitivity, which is an important step forward compared to the simpler models based only on Eqs. 1. However, the plume parametrization also relies on several rather strong assumptions, which we discuss below. First of all, both the original plume model and the parametrization have a quasi-1-D formulation, assuming homogeneity in the spanwise direction. Even though we attempt to translate this formulation to two dimensions with the algorithm in Section 2.3, there are undoubtedly errors associated with the underlying 1-D assumptions. As already discussed in Section 2.3, an important 2-D effect is the additional degree of freedom associated with the widening of the plume, which influences the plume dynamics and the melt rates through the mass budget equation (Hattermann, 2012; Hattermann et al., 2014).

Furthermore, the current algorithm for finding the plume paths in 2-D is not unique and more realistic and efficient

methods might be possible, e.g. by extrapolating the plume outward from the grounding line instead of searching for surrounding grounding-line points from each shelf point. Also, the current algorithm was developed for the relatively coarse resolution of $20 \times 20$ km, suitable for use in an ice-sheet model, and takes into account only the local slope and overal grounding-line depth, whereas higher resolution runs might benefit from a different and more precise method. For example, the current method inevitably includes unrealistic plume paths along points where the basal slope reserves, which might give problems at higher resolutions. On the other hand, a higher resolution would also entail a more rapid variation of the basal slope, potentially causing high melt peaks (Section 3.1) that would be smoother in the original plume model. This would introduce the need for a smoothing algorithm for higher resolutions. All in all, the current formulation should be considered as a relatively simple parametrization of the net circulation within an ice-shelf cavity, providing non-local features to the basal melt calculation that are not present in the simpler models. Further work is needed to determine whether the realism of the current formulation can be improved.

Another very important feature that has been neglected in the derivation is the vertical variation in the temperature and salinity fields. In reality, stratification and the existence of different water masses have a crucial effect on plume buoyancy, e.g. by causing the plume to detach from the ice-shelf base at levels of neutral buoyancy. In such cases, new plumes are formed at the detachment depth and the relation between the plume and the grounding-line depth breaks down, creating multiple modes in the sub-shelf circulation and associated basal melt (Jacobs et al., 1992). As explained in Sec. 2.2, the current formulation is based on the assumption that the freezing-point length scale (7) is dominant w.r.t. the length scale associated with stratification, as well as those associated with rotation and the initial meltwater flux at the grounding line. This assumption indeed works well in conjunction with constant values for $T_a$ and $S_a$, describing a net circulation for which the buoyancy is parametrized in terms of $T_a - T_f$, as shown more precisely in Appendix A. In this framework, the values of $T_a$ and $S_a$ determine the overall magnitude of plume buoyancy, while the variation along the plume path is described by the depth-dependence of the freezing point $T_f$. This is also the reason why the small horizontal variations in $S_a$ have only a small effect on the overall buoyancy and can be neglected, as was done in Section 3. However, for obtaining a fully realistic melt rate pattern it will be important to also include the effects of vertical and seasonal variations in $T_a$ and $S_a$, e.g. in order to capture seasonal intrusion of warmer surface waters (mode 3 melting; Jacobs et al. 1992; Hattermann et al. 2012; Stern et al. 2013).

An important uncertainty in the current study is the construction of the effective temperature field (Section 3.2). In principle, this is done due to the lack of detailed ocean temperature observations beneath the ice shelves. One should note, however, that in attempting to eliminate the biases caused by the sparse data, we are also correcting for errors in the parametrization itself, since the construction is done by constraining the modelled melt rates. In this respect, the *effective* temperature field (or more precisely, $\Delta T$) should be regarded as part of the modelling framework. It would be crucial for the complete validation of the model to perform additional temperature sensitivity studies to see how the plume parametrization might respond to an evolving ocean. Ideally, this is done in the context of a coupled ice-ocean model. On the large scales currently considered, lack of detail within the ice-shelf cavities will likely remain a problem also when using an ocean general circulation model. Since the current formulation is based on constant ocean properties within individual cavities, a method to determine $T_{\text{eff}}$ from an ocean model could be extrapolating the model temperature within a characteristic depth-range at the ice front and using a (possibly different) $\Delta T$ to constrain the output melt rate, similar to the construction presented here.

On a more technical note, the current construction of $T_{\text{eff}}$ was not based on a sophisticated optimization algorithm, but it is merely a simple method to determine an essentially spatially variable field directly from the observations. An alternative method, which might be more consistent with the derivation of the parametrization, would be to introduce separate values for the ocean temperature for each individual cavity, as the ambient temperature in the current context represents the net inflow into the cavity and not the temperature of meltwater that is produced or mixed locally. On the other hand, the current method is more generic in the sense that it removes the need for defining individual cavities in the model once $\Delta T$ (i.e. the constraint on the melt rates) has been determined. It should be noted that the current method using only 29 sample points might become problematic in dynamical simulations that include grounding-line retreat. Hence, in such a context a more sophisticated method might be necessary. Furthermore, it is not yet clear if a fixed $\Delta T$ is a realistic assumption for an evolving ocean, and introducing the aforementioned additional variations of $T_a$ and $S_a$ might require different considerations altogether.

Finally, it is interesting to note the existence of alternative methods for describing the net circulation within the ice-shelf cavities. A recent example is a box model that simulates the upward flow under the ice shelf in a similar quasi-1-D context by describing the fluxes of heat and salt between a limited number of predefined boxes (Olbers and Hellmer, 2010). This method has recently been extended to two dimensions and coupled to an ocean model (Reese et al., 2017), yielding Antarctic basal melt patterns similar to the ones given by the plume parametrization. Both methods are similar in the sense that they essentially describe the same type of physical process while not accounting for features such as stratification and 2-D effects, as discussed above. One could argue that a systematically derived approximation to the governing equations is preferred over a simple box model. On the other

hand, a box model might be easier to implement and produce similar results in a more efficient way. A more detailed comparison of these two methods is beyond the scope of this work.

## 5   Conclusions

In this study, we have the presented the application of a basal melt parametrization, based on the dynamics of buoyant meltwater plumes, to all ice shelves in Antarctica. The physical basis of this parametrization is the plume model of Jenkins (1991), which describes the fluxes of mass, momentum, heat and salinity within a meltwater plume travelling up from the grounding line along the ice-shelf base. Details of the proposed parametrization have been discussed in earlier works (Jenkins, 2011, 2014) for idealized one-dimensional geometries along an ice-shelf flow line. In particular, the basal melt rate given by the plume model follows a rather universal scaling law depending on the ice-shelf geometry (basal depth $z_b$, local slope angle $\alpha$, and grounding-line depth $z_{gl}$) as well as the ambient ocean temperature $T_a$ and the pressure freezing point $T_f$.

Here, the plume parametrization has been tested for two realistic ice-shelf geometries along a flow line and, for the first time, applied to a completely two-dimensional geometry covering all the Antarctic ice shelves. The one-dimensional tests along flow lines of Filchner-Ronne and Ross ice shelves (Section 3.1) reveal the typical characteristics of the parametrization, namely higher melt rates near the grounding line and in regions of high basal slope. Patches of refreezing can occur further away from the grounding line. Moreover, the plume parametrization exhibits a nonlinear dependence on the ocean temperature, and the increase in melting resulting from higher ocean temperature is dependent on the ice-shelf geometry. In contrast, simpler parametrizations based solely on the local balance of heat at the ice-ocean interface are not able to capture the complex melt pattern nor the temperature sensitivity.

Applying the essentially one-dimensional plume parametrization to a two-dimensional geometry is not trivial and, ideally, it would require a detailed knowledge of both the ice-shelf geometry and the ocean circulation in the ice-shelf cavities. The method discussed in Section 2.3 provides a solution to this issue by constructing a field of effective grounding-line depths and slope angles for each shelf point from topographic data. The resulting values for $z_{gl}$ and $\alpha$ can be interpreted as reflecting the average effect of all plumes that reach the shelf point. This method provides a straightforward way to extend the parametrization from 1-D to 2-D for a given topography and ice mask, but it is not unique. As discussed in the previous section, a fully realistic 2-D formulation of the plume dynamics would require additional considerations.

However, since the temperature sensitivity of the plume parametrization can be considerable, a more important factor for the two-dimensional model is finding an ocean temperature field that is characteristic for the ocean water flowing into the ice-shelf cavities. In this respect, the results in Sections 3.2 and 3.3 show that the depth-averaged and interpolated data from observations require a plausible offset $\Delta T$ between $-1.4$ °C and $0.8$ °C in order to obtain an effective temperature $T_{\text{eff}}$ (Fig. 7b) with which the plume parametrization gives basal melt rates close to the present-day observations of Rignot et al. (2013). In contrast, a much higher offset $\Delta T$ between $-0.5$ °C to $5.4$ °C is required for obtaining the same melt rates with the quadratic parametrization of DeConto and Pollard (2016), as shown in Fig. 12. The same low temperature sensitivity of the melt rates from the latter parametrization is also apparent in Pollard and DeConto (2012), where different tuning factors in the basal melt parametrization are used for different sectors along the Antarctic coastline, and in DeConto and Pollard (2016), where offsets of 3 °C and 5 °C are added to the ocean temperature in the Amundsen and Bellinghausen seas (resulting from an ocean model) in order to obtain the correct present-day basal melt rates and grounding-line retreat.

All in all, the presented plume parametrization, together with the constructed effective temperature field, gives reasonable results for the spatial pattern of present-day basal melt in Antarctica. The inherent geometry dependence, based on the plume dynamics, gives a more natural spatial variation that cannot be captured with local heat-balance models, a major aspect being the occurrence of refreezing. Of course, the current discussion only assumes a steady state regarding the ice dynamics and the ocean temperature. The question remains how an ice-dynamical model would behave when coupled to the plume parametrization, both for present-day forcing and for a varying climate. As a next step, it is important to perform such transient simulations of an ice model coupled to the plume parametrization and conduct sensitivity experiments. For such simulations, the effective temperature in Fig. 7b, even though it is a constructed field, can prove to be a valuable reference state to which temperature anomalies can be added, as briefly discussed in Section 4. Eventually, coupled ice-ocean simulations (e.g. DeConto and Pollard 2016) might benefit from this approach by using both ocean-model output and this reference state to determine an appropriate temperature forcing for this type of basal melt parametrizations.

## Appendix A:  Details of the basal melt parametrization

Here we present more details of the basal melt parametrization summarized in Section 2.2, starting with the theoretical arguments behind its mathematical form. The precise form of the parametrization is, however, the result of an empir-

ical study of the plume model results (Jenkins, 2014) and described at the end of this appendix.

First of all, we consider a simplified form of the plume equations (2)-(4), (6), where we neglect all advection terms except the crucial mass flux $\Phi_m := \dfrac{\mathrm{d}DU}{\mathrm{d}X}$, since without this flux there would be no plume. Furthermore, we replace the salinity equation by an equation for the density contrast $\Delta\rho$ as defined in (4) (similar to Jenkins 2011), neglect the direct effect of the melt rate $\dot{m}$ in the mass and heat equations w.r.t. the entrainment flux (retaining it only for the buoyancy flux), neglect heat conduction into the ice in the ice-ocean interface condition, and take $S_i = 0$. In the case of constant ocean properties $(T_a, S_a)$, as considered also for the empirical derivation of the plume parametrization, this set of assumptions yields the following simplified system:

$$\Phi_m = E_0 U \sin\alpha, \tag{A1a}$$

$$\Phi_m U = D\frac{\Delta\rho}{\rho_0}g\sin\alpha - C_d U^2, \tag{A1b}$$

$$\Phi_m T = (E_0 U \sin\alpha)T_a - C_d^{1/2}\Gamma_{TS}U(T - T_f), \tag{A1c}$$

$$\Phi_m \frac{\Delta\rho}{\rho_0} = \beta_S \dot{m}S_a - \beta_T \dot{m}(T_a - T_f)$$
$$\qquad - \beta_T C_d^{1/2}\Gamma_{TS}U(T - T_f), \tag{A1d}$$

$$\frac{L}{c_w}\dot{m} = C_d^{1/2}\Gamma_{TS}U(T - T_f), \tag{A1e}$$

$$T_f = \lambda_1 S + \lambda_2 + \lambda_3 z_b. \tag{A1f}$$

This is an algebraic system that can be solved rather easily for $(U, T, \Delta\rho, \dot{m})$ as functions of the ambient properties $(T_a, S_a)$, the freezing point $T_f$ and the basal slope angle $\alpha$. The solution can be written compactly as follows:

$$\dot{m} = C_d^{1/2}\Gamma_{TS} \cdot U \cdot \left(\frac{\Delta T}{L/c_w}\right), \tag{A2a}$$

$$U = (gD\Delta\rho)^{1/2} \cdot \left(\frac{\sin\alpha}{C_d + E_0\sin\alpha}\right)^{1/2}, \tag{A2b}$$

$$\Delta T = T - T_f = \left(\frac{E_0\sin\alpha}{C_d^{1/2}\Gamma_{TS} + E_0\sin\alpha}\right) \cdot (T_a - T_f), \tag{A2c}$$

$$\Delta\rho = \left(\frac{C_d^{1/2}\Gamma_{TS}}{E_0\sin\alpha}\right)\left(\frac{\Delta T}{L/c_w}\right)Q_0^2(T_a, T_f, S_a), \tag{A2d}$$

with

$$Q_0(T_a, T_f, S_a) = \sqrt{\beta_S S_a - \beta_T\left(\frac{L}{c_w} + T_a - T_f\right)} \tag{A2e}$$

By substituting the expressions above in (A2a), we obtain three geometrical factors in the melt rate expression, corre-

sponding to the factor $g(\alpha)$ in the melt scale (8):

$$g(\alpha) = \left(\frac{\sin\alpha}{C_d + E_0\sin\alpha}\right)^{1/2}\left(\frac{C_d^{1/2}\Gamma_{TS}}{C_d^{1/2}\Gamma_{TS} + E_0\sin\alpha}\right)^{1/2}$$
$$\cdot \left(\frac{E_0\sin\alpha}{C_d^{1/2}\Gamma_{TS} + E_0\sin\alpha}\right) \tag{A3}$$

What remains is to find the required quadratic temperature dependence in (8). First note that the factor $Q_0$, essentially determining the magnitude of buoyancy, can be taken approximately constant for constant $S_a$ and $T_a - T_f \ll L/c_w$, which is a reasonable assumption with the values in Table 3. Second, the expressions in (A2) depend on the plume thickness $D$, which is still an unknown variable. However, for a simple geometry with a constant and small slope $\alpha$ and slowly varying $U(X)$, the plume thickness can be explicitly solved from the mass equation (A1a) and directly related to the depth difference and, hence, the temperature difference:

$$D = E_0(\sin\alpha)X \approx E_0(z_b - z_{gl}) = E_0 \cdot l \cdot \hat{X}$$
$$\sim (T_a - T_{f,gl})\hat{X}, \tag{A4}$$

where we have used (7) to incorporate the length scale and the dimensionless coordinate $\hat{X}$. A linear thickening of the plume is indeed a reasonable approximation for a constant slope that is also seen in the plume model output, with slight deviations when the plume decelerates. Third, the temperature differences $T_a - T_f$ and $T_a - T_{f,gl}$ are related in a rather straightforward way:

$$T_a - T_f = T_a - T_{f,gl} - \lambda_3(z_b - z_{gl})$$
$$= (T_a - T_{f,gl})\left(1 - \frac{\lambda_3(z_b - z_{gl})}{T_a - T_{f,gl}}\right)$$
$$\approx (T_a - T_{f,gl})\left(1 - \hat{X}\right) \tag{A5}$$

Using (A4) and (A5) in (A2) now yields the following dependence for the melt rate:

$$\dot{m} \sim U\Delta T \sim D^{1/2}\Delta\rho^{1/2}\Delta T \sim D^{1/2}\Delta T^{3/2}$$
$$\sim D^{1/2}(T_a - T_f)^{3/2}$$
$$\sim (T_a - T_{f,gl})^2 \cdot \hat{X}^{1/2}\left(1 - \hat{X}\right)^{3/2}, \tag{A6}$$

which is the required quadratic dependence on $T_a - T_{f,gl}$.

In summary, we have shown how the assumption of a simple geometry with constant slope and constant ocean properties in the simplified system (A1) leads to the form of the melt rate scale (8). As a consequence of the derivation, we also found a relation $\dot{m} \sim \hat{X}^{1/2}(1 - \hat{X})^{3/2}$, showing how the melt rate rather naturally depends on the scaled coordinate $\hat{X}$ defined in (7) (disregarding the factor $f(\alpha)$ for the moment; see below). However, this particular function of $\hat{X}$ does correspond to the general melt curve in Fig. 2. In particular, it

only yields positive values for $0 < \hat{X} < 1$ and does not capture refreezing. The message is that at this point, although we can formally derive the melt rate scale $M$ with the correct temperature and slope dependence, it is still necessary to do an empirical scaling of the plume model results in order to obtain the correct function of $\hat{X}$. This empirical "fine-tuning" then leads to the exact form of the parametrization described below, including parameters $M_0$, $x_0$, $\gamma_1$, $\gamma_2$ as well the polynomial fit of $\hat{M}(\hat{X})$. A more thorough analysis of the plume equations would be required to derive the correct form of the melt curve in a similar way as sketched here, possibly including more physical phenomena that were neglected here, such as stratification.

The precise form of the parametrization can now be described as follows. For a given point at the ice-shelf base with local depth $z_b$ and local slope angle $\alpha$, we can determine the corresponding grounding-line depth $z_{gl}$ and ambient ocean properties $T_a$ and $S_a$. As summarized in Table 1, these quantities, together with a set of constant parameters, serve as the input of the parametrization. The basal melt rate $\dot{m}$ in meter per year at the particular ice-shelf point is now calculated as follows. First we define the characteristic freezing point:

$$T_{f,gl} = T_f(S_a, z_{gl}) = \lambda_1 S_a + \lambda_2 + \lambda_3 z_{gl}, \tag{A7}$$

and an empirically derived effective heat exchange coefficient, essentially depending on plume temperature, as discussed in Sec. 2.2:

$$\Gamma_{TS} = \Gamma_T \left( \gamma_1 + \gamma_2 \cdot \frac{T_a - T_{f,gl}}{\lambda_3} \cdot \frac{E_0 \sin\alpha}{C_d^{1/2}\Gamma_{TS0} + E_0\sin\alpha} \right). \tag{A8}$$

The empirically derived melt rate scale $M$ in meter per year (Eq. (8)) is now calculated from:

$$M = M_0 \cdot g(\alpha) \cdot (T_a - T_{f,gl})^2, \tag{A9}$$

indeed having the general form derived at the beginning of this appendix. Furthermore, the length scale $l$ (Eq. (7)) is given by:

$$l = \frac{T_a - T_{f,gl}}{\lambda_3} \cdot \frac{x_0 C_d^{1/2}\Gamma_{TS} + E_0\sin\alpha}{x_0(C_d^{1/2}\Gamma_{TS} + E_0\sin\alpha)}, \tag{A10}$$

where the second factor, corresponding to $f(\alpha)$ in (7), provides a slope-dependent scaling of the point of transition between melting ($\dot{m} > 0$) and refreezing ($\dot{m} < 0$) (see Fig. 2), as discussed in Sec. 2.2. The empirically derived dimensionless scaling factor $x_0 = 0.56$ ensures that the transition point occurs at the same dimensionless position for all plume model results. We can now determine the dimensionless coordinate:

$$\hat{X} = \frac{z_b - z_{gl}}{l}, \tag{A11}$$

**Table A1.** Coefficients for the polynomial fit of the dimensionless melt curve $\hat{M}(\hat{X})$.

| | |
|---|---|
| $p_{11}$ | $6.388 \times 10^4$ |
| $p_{10}$ | $-3.521 \times 10^5$ |
| $p_9$ | $8.467 \times 10^5$ |
| $p_8$ | $-1.166 \times 10^6$ |
| $p_7$ | $1.015 \times 10^6$ |
| $p_6$ | $-5.820 \times 10^5$ |
| $p_5$ | $2.219 \times 10^5$ |
| $p_4$ | $-5.564 \times 10^4$ |
| $p_3$ | $8.927 \times 10^3$ |
| $p_2$ | $-8.952 \times 10^2$ |
| $p_1$ | $5.528 \times 10^1$ |
| $p_0$ | $1.371 \times 10^{-1}$ |

and calculate the basal melt rate from:

$$\dot{m} = M \cdot \hat{M}(\hat{X}), \tag{A12}$$

where $\hat{M}(\hat{X})$ is the dimensionless melt curve shown in Fig. 2 and given by the following polynomial function:

$$\hat{M}(\hat{X}) = \sum_{k=0}^{11} p_k \hat{X}^k, \tag{A13}$$

for which the coefficients $p_k$ are given in Table A1.

Note that we require $0 \le \hat{X} \le 1$ in order to remain within the valid domain of the polynomial fit and avoid unbounded values of $\hat{M}$. It is rather straightforward to show that $\hat{X} \le 1$ is guaranteed for $T_a \ge \lambda_1 S_a + \lambda_2$, i.e. the ocean temperature should be above the freezing point at surface level ($z = 0$). By combining equations (A7), (A10) and (A11) and taking the limit $T_a \to \lambda_1 S_a + \lambda_2$, we obtain $\hat{X} \to (1 - z_b/z_{gl})F^{-1}$, where $F$ denotes the second (slope-dependent) factor in (A10). Because all the terms appearing in this factor $F$ are positive and $x_0 < 1$, we have $F \ge 1$. Together with $z_{gl} \le z_b \le 0$, this implies $\hat{X} \le 1$ in this particular limit for the ocean temperature. Since $T_a$ appears in the denominator of $\hat{X}$ in (A11), ocean temperatures above this limit will yield smaller values for $\hat{X}$. Hence, the $\hat{X} \le 1$ is guaranteed for $T_a \ge \lambda_1 S_a + \lambda_2$. Note that this is the reason why we have applied this lower limit to the effective temperature $T_{\text{eff}}$ in Fig. 7b. The physical reason for the constraint $\hat{X} \le 1$ is that the plume has lost momentum beyond this value (see Jenkins 2011). Alternatives for constraining the temperature could therefore be forcing $\dot{m} = 0$ for $\hat{X} > 1$ (which would, however, lead to a discontinuity in the melt curve in Fig. 2) or simply forcing $\hat{X} \le 1$ explicitly.

*Competing interests.* The authors declare that they have no conflict of interest.

*Acknowledgements.* The authors thank Dr Jeremie Mouginot for providing the spatial data of basal melt rates from Rignot et al. (2013). Dr Tore Hatterman and Dr Xylar Asay-Davis are thanked for their thorough review and useful comments on the manuscript. Financial support for W.M.J. Lazeroms was provided by the Netherlands Organisation for Scientific Research (NWO-ALW-Open 824.14.003). The lead author wishes to acknowledge the hospitality of Eindhoven University of Technology where part of the work was done.

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
