# Peer review of "Modelling present-day basal melt rates for Antarctic ice shelves using a parametrization of buoyant meltwater plumes"

_The Cryosphere, 2017_

## Referee Comment (RC1) · T. Hattermann (Referee) · 25 May 2017

**General comments**

The manuscript under review presents a new approach to parameterize the spatially resolved basal mass balance beneath floating ice shelves. The applied method uses the scaling of basal melt rates as a function of geometrical parameters and ambient ocean temperatures that has been derived in previous studies from a one-dimensional inclined plume model. Validation of this parameterization for the one-dimensional case is given through direct comparison with the underlying plume model. Aiming at generalizing the approach for two-dimensional applications, algorithms are proposed to compute local geometrical input parameters for an arbitrary ice shelf geometry. To

evaluate the method, circum-Antarctic melt rates are computed by tuning the ambient ocean temperatures to reproduce realistic area averaged melt rates and subsequently comparing the associated spatially resolving melt patterns with maps of observed basal melting.

The authors convincingly show that their method provides a significant improvement compared to the two referred simplistic approaches of Beckmann and Goosse (2003) and DeConto and Pollard (2016) that scale basal melting solely based on the temperature difference between the ambient ocean and the local freezing point. While following directly from a model of the underlying physics, the method is of general nature and (presumably) contains few enough free parameters to prevent over-fitting. Based on this, the presented work is inevitably a sound and useful contribution for better understanding and modelling ice-ocean interactions and should be published.

Said this, the following three concerns need to be addressed to make the merit and scope of this work fully available to the reader.

1 - The origin of the general melt rate curve needs clarification.

In the present manuscript, it is unclear how this central element of the parameterization was obtained and to which extent its derivation has been part of the current study and how much is based on previous works. Although not being explicitly mentioned, an expression of the curve shown in Fig. 2 seems to be given in eqn. A3, but also here lacking a proper derivation or a reference thereof. Meanings of the individual terms are discussed in the text, but neither their exponents nor the fact that they should be multiplied including the factor 10. A more explicit description of what has been done (and by whom) to establish this relation is needed. To ease the line of argument, it would probably also be useful to add a summary of the nature of the basal melt parameterization found by Jenkins 2014 (i.e. the existence of a general melt rate curve) at the beginning of section 2.2., e.g. by moving the paragraph on p. 8, l. 6-14 up front, including eqn. A3, and stating that the remainder of the section will review the meaning

of the respective terms.

2 – The rationale for the extension of the general melt rate curve for the two-dimensional case needs to be explained and discussed.

A basic assumption of the underlying plume model is that the geometry of the ice shelf is uniform in the direction perpendicular to the plane in which the plume is rising (p. 4, l. 15-16). This leads to a system of one dimensional equations (p. 5 l. 6-12) that yields the general melt rate curve when being evaluated for the parameter space of interest. One restriction that is imposed by this one-dimensionality and that is inherit to the general melt rate curve, is that changes in plume thickness and hence the susceptibility of relevant properties (such as plume temperature and buoyancy that in turn controls its speed) are fully predicted by the sum of the fluxes through the upper and lower interface with the ice and ambient ocean respectively. For a two-dimensional configuration, however, one would expect that the width of the plume becomes a dynamic variable of some sort, which through mass conservation affects the plume thickness as well as the width of the interface through which the plume interacts with the ice and ambient ocean. The result would be an increased degree of freedom and I am inclined to believe that this would lead to significant deviations from the general melt rate curve found for the one-dimensional case. This issue is currently lacking attention in the study.

For instance, many ice shelves exhibit asymmetric geometries, being narrower towards the grounding line and wider towards the ice shelf front, as simplistically illustrated in fig 1. Considering that every point of the ice base is covered by a multitude of plumes arising from the deepest grounding line, it is obvious that each individual plume must become wider (and hence thinner) as it ascends towards shallower depth, with direct consequences for its evolution. In fact, augmenting the original plume model of Jenkins (1991) by implementing a varying plume width in a two-dimensional configuration (Hattermann, 2012 section 3.5) it is possible to reproduce melt rates obtained from a general circulation ocean model of a realistic ice shelf geometry for a range of forcing parameters (Hattermann, et al 2014), while for the same setup, the original plume
model is overestimating the melt rates along a one-dimensional flow line by an order of magnitude, primarily because the unscaled (for width) plume predicts too vigorous currents beneath the shallower part of the ice shelf. In essence, the extension into two-dimensions is likely to weaken the influence on the non-local effective grounding line depth, as a thinner and wider plume would more quickly cool and slow down on its rise along the ice base, remaining less of its properties at the source location (possibly also earlier reaching ambient buoyancy and detaching from the ice base, leading to initialization of a new plume at the detaching depth–a case that is not discussed in the manuscript at all).

Much effort has been spent in the current manuscript on reviewing the one-dimensional plume theory that is the basis of the generalized melt rate curve. However, it is currently lacking a discussion and evaluation of the validity of the transfer of that relationship and its underlying physics to higher dimensions and the possible shortcomings therein, such as the above mentioned consequences of mass conservation for an asymmetric distribution of ice shelf area with depth (which is a qualitatively different argument concerning the plume physics than the fact that there might exist multiple plume pathways). I acknowledge that this assessment can be added at various level of detail. Also, in the overall need for simplicity and recognition of other examples of parameterizations that have been used in the past, the presented approach is likely to be justified as is for the purpose of providing boundary conditions for ice sheet models. But the authors need to add some sort of assessment of physical basis for their transfer, which appears to me the major advance of this study.

3 – The evaluation of the performance of the parameterization for the circum-Antarctic case needs improvement, in particular, more information must be provided on the limitations and processes not captured by the present approach.

In the current manuscript, the performance of the method is evaluated by comparison with the simplistic approaches of Beckmann and Goosse (2003) and DeConto and Pollard (2016). In particular, the generalized plume approach is shown to be largely

superior in reproducing a qualitatively realistic spatial pattern of basal melt (increased melting towards the grounding lines) and the need of fewer adjustments of the ambient ocean temperature field to obtain spatially averaged melt rates that match the observations than required by the traditional thermal driving parameterizations. From an ice dynamical modelling perspective, this is certainly an important step forward. However, today, models of a wide range of complexity are used to assess the ice-ocean system (see e.g. Asay-Davis et al., 2016 for a summary) and within the scope of these works, it is desirable to evaluate the proposed parameterization also with results from the other end of the spectrum. A couple of circum-Antarctic ocean general circulation models are readily available (even more regional models, some of already coupled with ice models), providing fields of basal melt rates by explicitly resolving the ice shelf cavity circulation. Although not necessarily yielding realistic results everywhere, all of these simulations provide a self-consistent sets of geometrical parameters and ambient ocean temperatures that can be used to scrutinize the validity of the presented plume parameterization. Applying the new parameterization in the context of a fully resolving ocean model framework of the author's choice appears to be a minor additional effort and I highly recommend that such a comparison is added to this publication, as it would substantially aid the validation of the approach (such as its extension to two dimensions) and greatly improve the understanding and integration of the new method within the context of existing works on simulating basal melting.

Another issue is that the melt rate maps shown in Fig. 10 work well to assess the improvement over the simplistic scaling of DeConto and Pollard (2016), but do not allow to compare the details of the melt rate map with the observations of Rignot et al. (2013) that is used as a reference (p. 20 l. 10-14). In particular, the truncation of the color scale to melt rates of 2 m/yr excludes a quantitative assessment of the maximum melt rates that can be an order of magnitude larger at some grounding lines with important effect on the ice dynamics. Within this scope, it is currently also not accessible to the reader, how the tuning points for the ambient ocean temperature field were chosen and by which algorithm the temperature in these points has been optimized to match the

area averaged melt rates (see specific comments for details). Eventually, there are a couple of processes that are known to influence basal melting around Antarctic, but are not captured by this parameterization, with examples being the influence of regionally varying tidal current strength on the boundary layer heat exchange (Maksinon et al. 2011), as well as the enhanced heat exchange due to winds (Hattermann et al. 2014, Dinnimann et al. 2015) as well as intrusions of solar heated summer water near the ice fronts (Hattermann et al. 2012, Stern et al. 2013). Hence, their influence must either be omitted or be included in the fitting of the temperature field, a limitation of the new approach that needs to be discussed.

Also, to some extent the precision of language and figure quality should be improved.

Specific comments

p. 2, l. 11 & 15: What kind of "steady-state" is referred to and what is meant by the "steady nature" of the parameterizations? Does the new parameterization differ in a manner that it is time-varying of some sort?

p. 2, l. 14: Please clarify the ambiguous formulation "geometry below the ice shelves".

p. 2, l. 27-28: How does the referred mechanism in which upward flowing plumes induce inflow of warm water into the cavity relate to the approach presented? To my understanding, this possible feedback on the ambient ocean temperatures is not part of the plume model or the derived parametrizations, opposing the subsequent statement in line 32.

p. 3, l. 1-2 / p. 8, l. 16 ff / p. 22, l. 6-8: With the above general comment in mind, please reflect on the validity of the underlying physics, in particular the non-local dependence on grounding line depth, when extending the plume parameterization to two dimensions.

p. 3, l. 15-18: It is not always clear, which parts of these sections review the results of previous works and which parts are original contributions of the present study. It is

mentioned that results are summarized from Jenkins (2011) and Jenkins (2014), while particular advances of the present study are not discriminated in detail. To some extent, the problem may arise, because a central reference of the plume theory is contained in a conference presentation, which is not available for reading. However, explicitly clearly labeling review information and original material of this paper at the beginning of the subsections, should sufficiently mediate this issue.

p. 4, l. 3-5: It would be useful to explain how the ocean current that drives mixing relate to the temperature, hence leading to the non-linearity referred to here. Does this refer to the effect of increased buoyancy by decreased salinity due to more meltwater input for higher temperatures?

p. 6, l. 8: For clarity, mention which simplification is applied, i.e. the assumption of a constant ratio between Gamma_T and Gamma_S.

p. 6, l. 23 ff.: The derivation of the general melt rate curve appears somewhat fragmented and I am currently not able to retrace its origin based on the information given in this section. In particular, it is not clear how the terms in eqn. 7 to 9 combine into a single expression. Specifically, it is unclear how Jenkin's extension of eqn. 7 looks like and what is described by the universal length scale mentioned on p. 7, l. 21 or how it is used. Also the discussion of the two different melt formulations (p.7, l.21-27) is confusing in the given context, as is the summarizing statement in p. 7, l. 28 (amplitude of which curves?!). Clarity would probably be added by stating in the beginning of the section that Jenkins 2014 has derived an explicit and universal expression of melt rates as function of distance from the grounding line (possibly including eqn. A3) and explaining that the remainder of the section revises the basic ingredients, to sketch how the relationship was obtained but without providing a stringent derivation.

p. 8, l. 1 & 2: The plume buoyancy is primarily controlled by salinity, while temperature has only little influence on the density for the given parameter range. Even though this is not stated explicitly, I assume that by parameterizing the plume buoyancy through

the temperature difference between the plume and the ambient ocean, an assumption was made on how the temperature difference translates into a salinity difference (i.e. the freshening of the plume is obtained from transforming its respective source water along the melting-freezing mixing line/ Gade line). Does this imply that the general melt rate curve was obtained by assuming that the ambient water at any location along the plume path is the same (or lies along the same Gade line) as at the grounding line where the plume originates? In this case, this would be an important limitation of the theory, which is almost certainly not true for many ice shelves, where different source water types may dominate the ice ocean interaction in different parts of the ice shelf cavity (e.g. different sources of HSSW beneath Filchner-Ronne or the influence of more buoyant surface water near ice shelf fronts).

p. 10, l. 6-12: If my understanding of this algorithm is correct, valid plume paths will also incorporate directions for which the ice base slope reverses somewhere between the given ice shelf point and the respective grounding line since only the local slope and overall grounding line depth are evaluated. What does this imply for the nature and realism of the resulting multitude of valid plume paths?

p. 15, l. 6-9: Should be moved to discussion and supported through proper references.

p. 16, l. 5: For the given temperature range, the buoyancy of the plume is dominated by salinity differences. Please comment how the uniform salinity field affects the response of the melt rate parameterization (or its inherent ingredients).

p. 18, l. 9-11: More information on this tuning process must be provided. How were the respective temperature differences in the 29 sample points determined? Presumingly, some sort of optimization algorithm has been applied, that involves iterative computation of area averaged melt rates and subsequent adjustment of the individual correction points. How well does this procedure converge towards a unique solution for the given cost function? Why were 29 points used and how have they been allocated and how sensitive is the resulting melt rate map to this particular configuration (from Fig. 10a

and Fig. 11a one could get the impression that more spatial detail on the melt rate map correlates with a higher density of correction points)?

p. 18, l. 24-27: It is well known that most of the seawater beneath the FRIS is significantly colder than the surface freezing point. The reason for this is that melt water produced at greater depths is largely recirculated within the cavity and mixes with inflowing water at the surface freezing point, before this interacts with the ice base. Thus a representation of colder ambient water masses would indeed be more realistic in this case.

p. 18, l. 32-33: In fact, the continental shelf temperatures in West Antarctica in Fig. 8a appear rather low compared to observed values well above 0 degC. It would be useful know more about the spatial pattern of basal melt in this region and its comparison compares, in particular if the parameterization is capable of capturing the extremely large melt rates near the grounding lines that are observed here.

p. 21, l. 5-14: Obviously, the new plume parameterization provides significantly improved spatial basal melt patterns compared to the simplistic temperature scaling. However, to this end, it remains somewhat unclear to what extent the obtained spatial pattern of basal melt is a result of underlying dynamics of the parameterization or reflects the optimization of ambient ocean temperatures that were used for the input. Thus, a direct comparison with melt rates from a more comprehensive ocean circulation model remains a desired complement to round off the present study. This, to my mind easy achievable extension of the present work would both help to justify the ad hoc extension for the two dimensional case and scrutinize the predictive capacity of the parameterization that is required for using it in a framework of evolving ice geometry or ocean temperature sensitivity studies.

p. 23, l. 3-7: In addition to the prescription of valid plume paths provided in this study, an extension of the one-dimensional plume theory to higher dimensions needs to account for the effects of mass conservation when the dynamical equations are not

constrained along a path of uniform width. This will have consequences on the validity of the general melt rate curve that need to be addressed here.

Figure 4: Use different colors for open ocean and land areas where the relevant fields are undefined.

Figure 1: Extend range of melt rates, consider using non-linear color scale.

Generally, most spatially resolving circum-Antarctic fields are difficult to assess. Consider the use of zoomed inlets to magnify relevant regions.

Technical corrections

Generally, the manuscript should be edited to improve the precision of language, including the removal of unnecessary conjectures and filling terms (examples being p. 1, l. 23: "Therefore", p. 4, l. 13: "ultimately", p. 7, l. 13: "hence", p. 8, l. 6: "thus", p. 9, l. 12: "easily", p. 9, l. 14: "Now", p. 11, l. 1: "In summary", p. 13, l. 16: "clearly", p. 18, l. 30: "Clearly", p. 20, l. 7: "obviously", p. 20, l. 20: "immediately") as well as first person narratives which is extensively used throughout the manuscript.

p. 1, l. 20: "ocean flow", better use "oceanic heat supply"

p. 2, l. 3: "In the view of these issues", imprecise, clarify: "In order to correctly predict the evolution of the ice sheet"

p. 3, l. 6: "An important part of this work is [the derivation/ the development of] an algorithm"

p. 3, l. 25 & 24: consistently refer to "sea water" when introducing rho_w and c_w.

p. 6, l. 2: if only similar, what is the difference between eqn. 1b and 5c.?

p. 11, l. 6-9: Redundant with p. 9, l. 9-11.

References

Dinniman, M. S., Klinck, J. M., Bai, L. S., Bromwich, D. H., Hines, K. M., & Holland, D.

M. (2015). The Effect of Atmospheric Forcing Resolution on Delivery of Ocean Heat to the Antarctic Floating Ice Shelves. Journal of Climate, 28(15), 6067-6085.

Hattermann, T. (2012) Ice Shelf - Ocean Interaction in the Eastern Weddell Sea, Antarctica, PhD thesis, UiT, The Arctic University of Norway, online available: http://hdl.handle.net/10037/5147

Hattermann, T., Nøst, O. A., Lilly, J. M., & Smedsrud, L. H. (2012). Two years of oceanic observations below the Fimbul Ice Shelf, Antarctica. Geophysical Research Letters, 39(12).

Hattermann, T., Smedsrud, L. H., Nøst, O. A., Lilly, J. M., & Galton-Fenzi, B. K. (2014). Eddy-resolving simulations of the Fimbul Ice Shelf cavity circulation: Basal melting and exchange with open ocean. Ocean Modelling, 82, 28-44.

Makinson, K., Holland, P. R., Jenkins, A., Nicholls, K. W., & Holland, D. M. (2011). Influence of tides on melting and freezing beneath Filchner‐Ronne Ice Shelf, Antarctica. Geophysical Research Letters, 38(6).

Stern, A. A., Dinniman, M. S., Zagorodnov, V., Tyler, S. W., & Holland, D. M. (2013). Intrusion of warm surface water beneath the McMurdo Ice Shelf, Antarctica. Journal of Geophysical Research: Oceans, 118(12), 7036-7048.
* * *
[Figure]

**Fig. 1.** Idealized illustration of assymetric ice shelf geometry. Blue arrows inidcate possible plume pathways that require widening of the plume at shallower depth to span the entire ice base..

[Figure]

---

## Referee Comment (RC2) · X. Asay-Davis (Referee) · 5 Jul 2017

Review of Lazeroms et al. "Modelling present-day basal melt rates for Antarctic ice shelves using a parametrization of buoyant meltwater plumes"

Reviewer: Xylar Asay-Davis

I wish my name to be relayed to the authors, as I do not support the practice of anonymous review.

**General comments:**

This paper presents a new method for computing basal melt rates below Antarctic ice shelves based on a polynomial best-fit to a non-dimensionalized 1D plume model. The major innovation of this work is the methods for computing the parameters (the slope of the ice draft and the height above the grounding line) for the 1D plume fit based on 2D ice and bedrock topography data. The result appears to be a low-cost, physically based method that can capture the large range of observed mean melt rates for groups of Antarctic ice shelves. Melt rate patterns are also argued to be closer to observations than those from other melt parameterizations, though this is not shown quantitatively.

This work represents a *significant* step forward in bridging the gap between more complete representations of sub-ice-shelf dynamics (e.g. in 3D ocean models or 2D plume models) and simplified, ad hoc melt parameterizations that contained little or no physics. Given the computational expense of ocean and plume modeling and the fact that ice-sheet models are not fully coupled into earth system models, there is a need in the ice-sheet modeling community for parameterizations and simplified models like the one proposed here to improve the realism of forcing from basal melting in response to changes in ocean temperature.

The main concern I have with the paper involves the discussion around the temperature correction field $\Delta T$ applied to the observed temperatures from World Ocean Atlas (WOA). First, the claim is made that this correction is necessary because of unknown temperatures below the ice shelves, summer biases of observations and the interpolation method used to produce the base temperature field $T_0$ from WOA. No doubt, these factors do contribute to $\Delta T$. But inaccuracies in the plume model itself are also being swept into $\Delta T$. It is reassuring, as the authors state, that the $\Delta T$ is not unrealistically large (as they show it to be for an alternative parameterization), suggesting the strength of the plume-based parameterization. At the same time, the authors' sensitivity study in Sec. 3.1 shows that melt rates can be highly sensitive to changes in temperature that are of the same order as $\Delta T$. This suggests that the evolution of melt rates, even if they are calibrated to match present-day observations, are likely to be highly sensitive to $\Delta T$. This is not shown or discussed in the paper. An application of this parameterization in ice-sheet simulations forced by time-evolving ocean observations or simulation results would require a method for determining $\Delta T$. The paper would benefit from some more discussion of how the authors foresee $\Delta T$ being computed in these scenarios. Namely, what ocean state should be used

to compute ΔT? Observations? The initial state of the ocean forcing? How sensitive are the melt rates likely to be to this choice?

Another comment is that this paper relies heavily on Jenkins (2014), an EGU talk that does not seem to be available online.  This work is cited 9 times, often with the implication that the reader should be familiar with the equations and notation it uses.  I happen to have attended this particular EGU session but, as remarkable as the talk was, I can't say I remember the notation in detail.  Given how heavily this work relies on Jenkins (2014), it might be worth either providing a permanent URL to that those slides or providing their contents as an appendix here.  Otherwise, I would suggest efforts be made to cut down on how often that work is cited and instead to incorporate its findings directly into the paper.

In addition to the requested discussion above, I recommend a number of minor revisions to the manuscript in the specific comments below.  If these are addressed, I would recommend the manuscript for publication.

**Specific comments:**

In what follows, I will indicate the page number a line number as pp-ll (e.g. 1-1 for page 1, line 1) for simplicity.

2-9: "depend solely on the thickness of the water column beneath the ice shelf"  I'm not aware of any parameterizations that use the thickness of the water column only, and the authors don't give a citation for this.  Instead, most parameterizations I'm aware of depend only on the depth of the ice-ocean interface (the ice draft), with some parameterizations (e.g. Asay-Davis et al. 2016) *also* using the water-column thickness to taper off melting near the grounding line.

2-14: "Due to their steady nature, it is unlikely that the simple basal melt parametrizations contain enough physical details to capture this complex pattern without either significant tuning or extremely detailed ocean-shelf-cavity models." First, I have trouble following what it meant by "their steady nature".  Do the authors mean their lack of dependence on external forcing (e.g. ocean temperature)? Or that they assume steady state? Or something else, perhaps?  Second, "simple basal melt parameterizations" by definition will not be "extremely detailed ocean-shelf-cavity models", so I think the sentence needs to be rephrased to differentiate between parameterizations and detailed physical models.

3-10: "Special attention is given to the construction of an effective ocean temperature field from observations, which is required for providing realistic input data of the temperature within the ice-shelf cavities to the parametrization."  This is part of my concern about how the ΔT field is discussed in this paper.  I don't disagree that there are biases in the the WOA observations but I do not think the authors demonstrate (or can demonstrate) that the correction leads to a more realistic temperature field.  Instead, it is important to acknowledge that the various biases in the WOA observations, the interpolation/extrapolation of those observation, and the plume emulator are all being compensated by tuning ΔT, and this

process will not necessarily mean that the resulting effective temperature is more realistic than WOA.

4-3: "The non-linearity arose because the exchange velocity $\gamma_T$ in Eq. (1a) was expressed as a linear function of the ocean current driving mixing across the boundary layer." This is not quite sufficient to have nonlinearity. It is also important that the strength of the ocean current is itself a function of the thermal driving. Maybe add something like, "...across the boundary layer, which is itself a function of the thermal driving".

5-11, 5-12, 5-14: These are not the standard uses of the symbols $\Gamma_T$ and $\Gamma_S$ (e.g. Jenkins et al. 2010, Jenkins 2011). The exchange coefficients are typically defined to be distinct from the Stanton number, such that $St = (C_D)^{1/2} \Gamma_T$ (and similarly for salt). I would *strongly* recommend switching to this more standard notation or there is likely to be confusion when others try to implement the parameterization. (2c) and (2d) would therefore each need an extra factor of $(C_D)^{1/2}$ and this change would propagate to many other places in the manuscript.

6-12: "This simplified formulation can be used together with the prognostic equations (2) by assuming $T_b = T_f$." My understanding of the 2-equation formulation is not that one necessarily assumes that $T_b = T_f$, but rather that a new equation is adopted with the same form as (2) with $T_b$ substituted by $T_f$. We never need to know what $T_b$ is but if one were to need it (e.g. as an ice-sheet boundary condition), it would be different from $T_f$ because of the significantly lower salinity at the interface.

6-15: "Also note the similarity between Eqs. (6) and the simple melt model described by Eqs. (1), the difference being the inclusion of heat conduction and the parametrization $\gamma_T = \Gamma_{TS} U$." I would say an equally (or perhaps more) important difference is the use of the plume T and S instead of the ambient fields.

6-19: "...different vertical temperature and salinity profiles of the ambient ocean (Jenkins, 2011, 2014)." My understanding is that the polynomial emulator that the authors use does not account for stratification or vertical variations in T and S. This might be worth mentioning explicitly, either here or better yet in the discussion section. Accounting for T and S profiles that vary with depth as well as time would be a potential improvement for the future that might allow the parameterization to produce Mode 3 seasonal melting (as defined in Jacobs et al. 1992) near the calving fronts of "cold" cavities. This could potentially improve the melt pattern.

7-9: "three larger length scales" If it is clear which 3 of the 4 length scales is largest, I missed it. It might be best to explicitly state either which 3 are meant or which one is excluded.

7-17: "...the slope affects the entrainment rate, but not the melt rate..." I carefully read the corresponding section of Jenkins (2011) and I think what is shown is that the term in the mass conservation equation for the melt rate doesn't explicitly contain the slope, whereas the term for the entrainment rate does. However, when the equations are solved, the

resulting melt rate will depend on the slope, since the plume speed and thermal driving (which contribute to this the melt rate, as shown in Jenkins (2011), Eq. (14)) depend on the slope. So I think the phrase should be changed to something like "...the entrainment rate explicitly depends on the slope, whereas the melt rate does not..."

9-9: "In this study, we use remapped data based on the Bedmap2 dataset for Antarctica (Fretwell et al., 2013)," Do the authors perform any kind of a firn correction to the ice thickness, given the assumption of constant ice density in the masking in Table 2? How well does the mask for grounded ice, floating ice and open ocean from Bedmap2 compare with that from the approach in Table 2? The figures suggest that the grounding line might not match well with Bedmap2 (e.g the Amery and deeper parts of the Ross and FRIS) but part of this could be due to the relatively coarse resolution. Without a firn correction, I wouldn't expect the masking from Table 2 to be a good match to the mask provided with Bedmap2.

10-8: "the algorithm searches in this direction for the nearest ice-sheet point." This may be obvious to the authors but I think the method used to search for the nearest ice-sheet point should probably be stated explicitly. This part of the algorithm seems like it could potentially be quite slow, particularly at higher resolution. There might also be approaches (e.g. working out from the grounding line, caching the distance to the G.L. in each direction) that could be used to speed up the process. Is this something the authors have considered?

11-Fig. 3: "$d_n = 1/2(H_{b,1} + H_{b,2})$" why the factor of 1/2 exactly here? Is this because the grounding line is assumed to always fall on the edge halfway between a grounded and a floating point? Also, the reasoning behind the different approaches in (b) and (c) probably deserves a bit more explanation.

11-8: Why such coarse resolution (20 km)? Is the algorithm too costly to apply on finer resolution? Have the authors explored whether it still works at, say, 1 km resolution that seems to be needed to resolve grounding line dynamics? I could imagine that issues with noise due to rapid changes in bed slope (e.g. Fig 5b) would be exacerbated by finer resolution.

12-Fig. 4: There is a strange rim of floating ice around the whole of Antarctica not present in Bedmap2. Is that an artifact of the remapping scheme that was used? Or the masking scheme in Table 2? Perhaps the calving front is being smoothed out over multiple cells, leading to apparent floating ice where none was present in Bedmap2 before remapping? Also, as mentioned above, the grounded vs. floating mask doesn't look like Bedmap2. Is this just the coarser resolution or has something gone wrong either during remapping or the masking procedure in Table 2?

12-2: "The values for the local slope are typically higher both near the grounding line and the ice front, as shown in Fig. 4c." Could the steeper slope the authors see near the ice front be an artifact of smoothing or remapping? The cross sections in Figs. 5a and 6a look quite smooth, even given the 20 km resolution, compared to plots of cross sections from Bedmap2 directly and I have not seen this tendency toward steeper slopes toward the calving front in sections I have taken from Bedmap2.

13-20: "...the discrepancies between the current parametrization and the plume model are largest when the basal slope changes rapidly, because the parameterization responds immediately to the change while the full model has an inherent lag as the plume adjusts to the new conditions." This problem will likely get worse at higher resolution. Might it be worth looking into a certain amount of along-flow smoothing and/or lag of when computing the effective α? Perhaps something for the discussion section.

14-Fig 5, 15-Fig 6: It seems like what is potentially missing here is a comparison with the patterns from Rignot et al. (2013) or another melt rate field inferred from observations. I believe the Rignot data set is available from Jeremie Mouginot on request. The data set from Moholdt et al. (2014) is available from Gier Moholdt on request.

15-4: "This also means that the simplest basal melt parametrizations currently used in some ice-sheet models, namely constant values or monotonic functions of the water-column thickness below the ice shelf, are far from being valid." Again, I don't know of any models using the latter. Perhaps the authors mean "ice draft" instead of "water-column thickness below the ice shelf"?

16-3: "but we will assume that the variations in ocean salinity around Antarctica are so small that the pressure freezing point $T_f$ is only affected by variations in depth." What about buoyancy (via $\Delta\rho$)? Wouldn't this also depend on $S_a$? Also, how has $S_a$ been eliminated from the universal polynomial (given that it doesn't appear anywhere in Appendix A)? By assumption? Or has it been demonstrated in Jenkins (2014) that variations in $S_a$ in the observed range don't have an appreciable effect? Would this still be true if stratification were taken into account?

16-8: "The best possibility is an interpolated field…" First, I would rephrase "the best possibility" to something more like "We decide a more feasible approach was ...". Second, to me it is odd to speak of interpolating the field into the ice-shelf cavities. It seems that this is what the authors did, but in my own modeling I extrapolate the field into a given cavity with no regard for temperatures in cavities on the other side of Antarctica that might figure into interpolation. Indeed, my colleagues and I have run into trouble when we were too naive in our extrapolation technique, extrapolating warm ocean temperatures from the Amundsen and Bellingshausen Seas under deep parts of FRIS. This does not appear to have occurred using the natural neighbors interpolation approach used here but it might still be worth acknowledging that interpolating temperature between ice-shelf cavities that really don't interact with one another is not really physically realistic.

16-16: "requires minimally tuned forcing data to produce realistic output." First, I'm not sure I agree with the assessment that the forcing in "minimally tuned", since the tuning likely has a significant effect on melt rates and their evolution, as discussed above. Second, I'm not sure I would characterize the computation of a field with 29 degrees of freedom (to match 13 mean melt rates) as "tuning", which in my experience refers to attempting to constrain a small number of model parameters rather than a spatially dependent field. Instead, this seems like inversion, much like the approach used to compute basal sliding factors under

grounded ice in many ice sheet models. The authors have also characterized this as bias correction, but I do not necessarily agree with that characterization, as I stated above.

18-2: "interpolated using natural-neighbour interpolation (i.e. a weighted version of nearest-neighbour interpolation, giving smoother results) to obtain data in the entire domain of interest." Again, it seems strange to interpolate *between* cavities. I guess natural-neighbor interpolation effectively extrapolate into cavities as long as the closest open ocean points is in front of *this* cavity and not some other cavity?

18-6: "this modification is necessary for eliminating biases in $T_0$ caused by the sparse observations and numerical interpolation, and also because the flow dynamics of the ocean are not resolved." This may be the principle but in reality the authors are almost certainly also correcting for shortcomings in the parameterization itself.

18-9: "29 carefully chosen points" I think more explanation is needed about how these points were chosen. It appears that they are located at grounding lines near the boundaries between shelves with potentially differing properties. Assuming $\Delta T$ is held fixed during an evolving simulation, will values of $\Delta T$ in regions that are currently grounded be appropriate as the grounding line moves? What might the limitations be? Again, this may belong in the discussion.

18-11: "Note that for technical reasons explained in Appendix A, we have applied a lower limit to the effective temperature equal to the pressure freezing point at surface level." As the authors show in the results section, this seems to be a significant limitation on the approach, particularly when applied to "cold" cavity shelves like FRIS. Perhaps some discussion is warranted on how this restriction might be relaxed in the future, as I will discuss more below.

18-18: The whole preceding paragraph for determining $\Delta T$ is the most worrisome aspect of the algorithm to me. The choice of $\Delta T$ (resulting from the details of how $T_0$ is computed) will potentially determine a lot about how melt rates evolve with time in response to changes in ocean temperature.

18-28: "...yield realistic present-day melt rates for all shelf groups. Therefore, we can conclude that the effective temperature shown in Fig. 8b is a realistic forcing field, at least within the current modelling framework." I don't think the authors can make this statement. The field $\Delta T$ was inverted to yield realistic melt rates for the 13 ice-shelf groups, so the fact that this goal was reached does not suggest that the effective temperature is realistic. A comparison with observations not used to constrain the model would be needed to make such a conclusion. All the authors can conclude here is that their inversion worked as expected (except for FRIS) and that the resulting temperature field looks plausible.

20-10: "This fact, along with the general melt pattern and the correlation with the surrounding ocean temperature, are in line with observations, e.g. Rignot et al. (2013)." This is by construction, so be careful not to attempt to used this as a validation of the model.

20-11: "However, one should note that the Rignot et al. (2013) melt pattern shows a greater spatial variability, with more patches of (stronger) refreezing occurring between patches of positive melt. The lack of such prominent patches of refreezing in the current parametrization might have different reasons, such as the coarse resolution or the fact that we disregard the details of the ocean circulation within the ice-shelf cavities, as well as effects due to stratification and the Coriolis force." Lack of seasonal variability in T and S and also lack of vertical variability (not just in the sense of stratification, but also in the sense of having distinct water masses at different depths) likely also play a role. For example, this is likely why Mode 3 melting is missing (as mentioned above).

20-15: "All in all, the plume parametrization, together with the effective temperature field, appears to give a realistic melt pattern for Antarctica, showing both a large spatial variability and average melt rates that agree with observations." It is definitely a strength of this parameterization compared with its predecessors that it can capture the range observed melt rates. So I definitely think this deserves emphasis. But, again, this is by construction. It is a good property to have but I think the authors should be careful to state that this is not a validation of the model, since the observed melt rates were used in the inversion for $\Delta T$.

21-Fig 10: A comparison with the Rignot et al. (2013) melt rates seems like it would also make sense here. As I said, the data should be available on request. Having theme plotted with the same color map would make them much easier to compare.

21-13: "minimal tuning" As before, I'm not a fan of this phrasing. What was done was an inversion of a field that can be argued to be within a plausible range. To me, this is neither clearly "minimal" nor clearly "tuning".

23-1: "parametrizations based solely on the local balance of heat at the ice-ocean interface are not able to capture the complex melt pattern..." The authors can rightly claim to have a more broadly realistic melt pattern than these previous studies, with both melting and refreezing. But the authors have not really shown that the complex melt patterns resulting from their parameterization are contributing added realism compared to a simpler pattern with a similar distribution of melting and freezing, and complex patterns are not a goal in and of themselves.

23-14: "...data from observations only need a minimal offset $\Delta T$ (between $-1.4°C$ and $0.8°C$)" Again, I would suggest a different phrasing than "minimal". "Plausible"? Also, again I think some discussion is needed about how a time-varying $T_0$ field would be handled. Would $\Delta T$ be held fixed? (The authors seem to imply it would be)? How sensitive will the results likely be to $\Delta T$? Over what kinds of time scales might it be reasonable to hold $\Delta T$ fixed? How should data from ocean models be applied? Should a $\Delta T$ be computed from ocean-model initial conditions to match observed melt rates? Or should ocean observations (e.g. WOA) be used to compute $\Delta T$?

23-22: "All in all, the presented plume parametrization, together with the constructed effective temperature field, gives realistic results for the present-day basal melt in Antarctica, both in terms of area-averaged values (Fig. 9) and the spatial pattern

(Fig. 10a)." I don't think this paper as written has shown that the spatial patterns are realistic, just that the mean values are (by construction) consistent with observations. A more qualitative (or better yet quantitative) comparison of the spatial patterns with Rignot et al. (2013) or with another data set derived from observations would be needed to make the latter assertion here. Alternatively, the claim could be toned down, stating that the pattern is reasonable in a broad sense -- highest melt rates are near the grounding line with refreezing closer to calving fronts.

23-29: "For such simulations, the effective temperature in Fig. 8b, even though it is a constructed field, can prove to be a valuable reference state to which temperature anomalies can be added." As I have said earlier, I think more discussion is needed on how the effective temperature would be used in dynamic ice-sheet simulations using this parameterization. This sentence is a good start but I'd really like to see more.

23-30 "Eventually, coupled ice-ocean simulations (e.g. DeConto and Pollard 2016) can benefit from this approach by comparing ocean-model output to this reference State." Hmm, I hope I'm misunderstanding but it seems like the authors are claiming that their reference temperature should act as a reference field, from which coupled ice sheet-ocean simulations could be validated. If this is not what was intended, please clarify what is meant here. If that *is* what is meant, that's a very bold assertion, given the fact that plume-model biases are also "swept under the rug" during the inversion process for $\Delta T$ used to produce the effective temperature field.

24-9, 24-11 The numerical constants 3.5e-5 and 10 seem to need units of m and m/yr/°C$^2$, respectively. Maybe give them names and put them in a table or something, along with 0.545? Also, how were (A2) and (A3) derived, at least in broad strokes?

24-15: $x_0$ is unitless? How was it derived?

24-22: Reading Jenkins (2011), it seems like X > 1 means there is no momentum left in the plume, so I would expect setting m = 0 beyond this point would be more realistic than restricting $T_a$ to not go below the surface freezing point. By the way, it might be worth discussing why, physically, X should not be allowed to exceed 1.

25: It's not clear to me that Appendix B adds much to the text, other than to emphasize that the algorithms for finding $z_{gl}$ and α are arbitrary. The results are visibly less realistic than for the algorithm presented in the main text and the problems with the constraint on $T_a$ seem more severe. If you add Jenkins (2014) as an appendix, you might consider removing this one.

One final comment. Since your paper was submitted, an alternative method for parameterizing basal melt by Reese et al. (2017) has been submitted and is also on The Cryosphere Discussions (see full citation below). You might consider discussing how their approach compares with yours, including what the strengths and weaknesses of each approach might be for adoption in a full ice-sheet simulation. I don't mean this as shameless

self promotion even though I'm a coauthor.  I really do think these papers are highly relevant to one another.

**Typographic and grammatical corrections:**

1-1: It is a very minor thing but I would suggest another word or phrase besides "decline", which implies to me that the AIS used to be better than it is now.  Perhaps "...major factor in the decline in volume of the Antarctic Ice Sheet" or "...major factor in mass loss from the Antarctic Ice Sheet".  Also, "Ice Sheet" should be capitalized in this case.

2-13: "...a complex spatial pattern, which depends heavily on both the geometry below the ice shelves and the ocean temperature." I do not think the observations demonstrate this, though I agree that it is the case.  Perhaps rephrase something like  "...a complex spatial pattern, which can be inferred to depend heavily on..."

2-17: "within a single ice-shelf cavity"  Maybe consider "within individual ice-shelf cavities" instead, since we aren't talking about one specific ice-shelf cavity but rather about any one of several ice-shelf cavities in isolation.

6-30: "The first governing length scale is associated with the pressure dependence of the freezing point that imposes an external control on the relationship between plume temperature, plume salinity and the melt rate, which is determined by the temperature relative to the freezing point. " I don't follow this sentence.  What is determined by the temperature relative to the freezing point?  Perhaps try to reword or break this sentence into 2.

10-17: "found values" should be "values found"

23-17: "The latter behavior is also apparent in…" It's not entirely clear to me what is meant by "the latter behavior".  I guess it is the low sensitivity of the melt rate to changes in temperature, though this is not explicitly the behavior described in the previous sentence.

**References**

Asay-Davis, X. S., Cornford, S. L., Durand, G., Galton-Fenzi, B. K., Gladstone, R. M., Gudmundsson, G. H., … Seroussi, H. (2016). Experimental design for three interrelated marine ice sheet and ocean model intercomparison projects: MISMIP v. 3 (MISMIP+), ISOMIP v. 2 (ISOMIP+) and MISOMIP v. 1 (MISOMIP1). *Geoscientific Model Development*, *9*(7), 2471–2497. https://doi.org/10.5194/gmd-9-2471-2016

Jacobs, S. S., Helmer, H. H., Doakea, C. S. M., Jenkins, A., & Frolich, R. M. (1992). Melting of ice shelves and the mass balance of Antarctica. *Journal of Glaciology*, *38*(130), 375–387. https://doi.org/10.3198/1992JoG38-130-375-387

Moholdt, G., Padman, L., & Fricker, H. A. (2014). Basal mass budget of Ross and Filchner-Ronne ice shelves, Antarctica, derived from Lagrangian analysis of ICESat altimetry. *Journal of Geophysical Research: Earth Surface*, *119*(11), 2361–2380. https://doi.org/10.1002/2014JF003171

Reese, R., Albrecht, T., Mengel, M., Asay-Davis, X., & Winkelmann, R. (2017). Antarctic sub-shelf melt rates via PICO. *The Cryosphere Discussions*, 1–24. https://doi.org/10.5194/tc-2017-70

---

## Author Comment (AC1) · 15 Sep 2017

**Author response**

*The authors would like to thank both reviewers for their very constructive comments. We are glad to see that the reviewers appreciate the scope of our work, but we acknowledge that the issues raised are very important and need to be addressed to improve the quality of the manuscript. The most important changes are the addition of two new figures, a substantial revision of Sec. 2.2 and Appendix A, the addition of new Discussion section, and the removal of Appendix B. Below we present our response to all reviewer comments in blue.*

**General minor remark**

A typo had occurred in Table 1. The value of the exchange coefficient $\Gamma_T$ should have exponent -3 instead of -5. This has been changed.

**Response to Reviewer 1 (Tore Hattermann)**

General comments

The manuscript under review presents a new approach to parameterize the spatially resolved basal mass balance beneath floating ice shelves. The applied method uses the scaling of basal melt rates as a function of geometrical parameters and ambient ocean temperatures that has been derived in previous studies from a one-dimensional inclined plume model. Validation of this parameterization for the one-dimensional case is given through direct comparison with the underlying plume model. Aiming at generalizing the approach for two-dimensional applications, algorithms are proposed to compute local geometrical input parameters for an arbitrary ice shelf geometry. To evaluate the method, circum-Antarctic melt rates are computed by tuning the ambient ocean temperatures to reproduce realistic area averaged melt rates and subsequently comparing the associated spatially resolving melt patterns with maps of observed basal melting.

The authors convincingly show that their method provides a significant improvement compared to the two referred simplistic approaches of Beckmann and Goosse (2003) and DeConto and Pollard (2016) that scale basal melting solely based on the temperature difference between the ambient ocean and the local freezing point. While following directly from a model of the underlying physics, the method is of general nature and (presumably) contains few enough free parameters to prevent over-fitting. Based on this, the presented work is inevitably a sound and useful contribution for better understanding and modelling ice-ocean interactions and should be published.

Thank you for the positive words and for acknowledging the merit of our work.

Said this, the following three concerns need to be addressed to make the merit and scope of this work fully available to the reader.

1 - The origin of the general melt rate curve needs clarification.

In the present manuscript, it is unclear how this central element of the parameterization was obtained and to which extent its derivation has been part of the current study and how much is based on previous works. Although not being explicitly mentioned, an expression of the curve shown in Fig. 2 seems to be given in eqn. A3, but also here lacking a proper derivation or a reference thereof. Meanings of the individual terms are discussed in the text, but neither their exponents nor the fact that they should be multiplied including the factor 10. A more explicit description of what has been done (and by whom) to establish this relation is needed. To ease the line of argument, it would probably also be useful to add a summary of the nature of the basal melt parameterization found by Jenkins 2014 (i.e. the existence of a general melt rate curve) at the beginning of section 2.2., e.g. by moving the paragraph on p. 8, l. 6-14 up front, including eqn. A3, and stating that the remainder of the section will review the meaning of the respective terms.

It is correct that some of the details of the plume parametrization have only been shown in the conference contribution by Jenkins (2014). We have tried to summarize the most important aspects as best as possible, but obviously this was not clear enough. We apologize for the confusion. One of the reasons to keep this summary rather concise is that a second paper is planned that discusses the derivation of the plume parametrization in full detail. To overcome this gap, we have reformulated most of Sec. 2.2 and added some new details in Appendix A.
One aspect to keep in mind is that the current parametrization is ultimately based on a purely empirical study of the plume model results. The original scaling found in Jenkins (2014), including coefficients and exponents, was not derived analytically but empirically. We have now added some analytical results to the Appendix that (partially) justifies where the mathematical form of the equations comes from. We hope to further extend this analysis in a different paper.
We are grateful for the suggestion to move the final paragraph of Sec. 2.2 to the front. This indeed clarifies the form of the parametrization before the details are discussed.
For clarity, we would like to stress that eq. A3 is strictly speaking *not* the equation describing the curve in Fig. 2, but the equation describing the melt rate scale. The curve in Fig. 2 is the *dimensionless* curve obtained after scaling and described by the polynomial coefficients in Tab. A1. Hopefully this has been clarified in the newly formulated Sec. 2.2.

2 – The rationale for the extension of the general melt rate curve for the two-dimensional case needs to be explained and discussed.

A basic assumption of the underlying plume model is that the geometry of the ice shelf is uniform in the direction perpendicular to the plane in which the plume is rising (p. 4, l. 15-16). This leads to a system of one dimensional equations (p. 5 l. 6-12) that yields the general melt rate curve when being evaluated for the parameter space of interest. One restriction that is imposed by this one-dimensionality and that is inherit to the general melt rate curve, is that changes

in plume thickness and hence the susceptibility of relevant properties (such as plume temperature and buoyancy that in turn controls its speed) are fully predicted by the sum of the fluxes through the upper and lower interface with the ice and ambient ocean respectively. For a two-dimensional configuration, however, one would expect that the width of the plume becomes a dynamic variable of some sort, which through mass conservation affects the plume thickness as well as the width of the interface through which the plume interacts with the ice and ambient ocean. The result would be an increased degree of freedom and I am inclined to believe that this would lead to significant deviations from the general melt rate curve found for the one-dimensional case. This issue is currently lacking attention in the study.

For instance, many ice shelves exhibit asymmetric geometries, being narrower towards the grounding line and wider towards the ice shelf front, as simplistically illustrated in fig 1. Considering that every point of the ice base is covered by a multitude of plumes arising from the deepest grounding line, it is obvious that each individual plume must become wider (and hence thinner) as it ascends towards shallower depth, with direct consequences for its evolution. In fact, augmenting the original plume model of Jenkins (1991) by implementing a varying plume width in a two-dimensional configuration (Hattermann, 2012 section 3.5) it is possible to reproduce melt rates obtained from a general circulation ocean model of a realistic ice shelf geometry for a range of forcing parameters (Hattermann, et al 2014), while for the same setup, the original plume model is overestimating the melt rates along a one-dimensional flow line by an order of magnitude, primarily because the unscaled (for width) plume predicts too vigorous currents beneath the shallower part of the ice shelf. In essence, the extension into two-dimensions is likely to weaken the influence on the non-local effective grounding line depth, as a thinner and wider plume would more quickly cool and slow down on its rise along the ice base, remaining less of its properties at the source location (possibly also earlier reaching ambient buoyancy and detaching from the ice base, leading to initialization of a new plume at the detaching depth–a case that is not discussed in the manuscript at all).

Much effort has been spent in the current manuscript on reviewing the one-dimensional plume theory that is the basis of the generalized melt rate curve. However, it is currently lacking a discussion and evaluation of the validity of the transfer of that relationship and its underlying physics to higher dimensions and the possible shortcomings therein, such as the above mentioned consequences of mass conservation for an asymmetric distribution of ice shelf area with depth (which is a qualitatively different argument concerning the plume physics than the fact that there might exist multiple plume pathways). I acknowledge that this assessment can be added at various level of detail. Also, in the overall need for simplicity and recognition of other examples of parameterizations that have been used in the past, the presented approach is likely to be justified as is for the purpose of providing boundary conditions for ice sheet models. But the authors need to add some sort of assessment of physical basis for their transfer, which appears to me the major advance of this study.

We fully agree that the one-dimensional formulation of the plume model is a rather strong assumption and regret not having spent more attention on this issue, so we take the opportunity to discuss this briefly. A discussion of the neglected two-dimensional effects including the references mentioned above has been added to Section 2.3 and the Discussion section. However, we hope that the reviewer realizes that the current approach is nothing more than a parametrization of the net circulation within the cavity, which can never fully capture all the physics. Still, it would be interesting to explore this and other extensions of the plume model in a later stage.

3 – The evaluation of the performance of the parameterization for the circum-Antarctic case needs improvement, in particular, more information must be provided on the limitations and processes not captured by the present approach.

In the current manuscript, the performance of the method is evaluated by comparison with the simplistic approaches of Beckmann and Goosse (2003) and DeConto and Pollard (2016). In particular, the generalized plume approach is shown to be largely superior in reproducing a qualitatively realistic spatial pattern of basal melt (increased melting towards the grounding lines) and the need of fewer adjustments of the ambient ocean temperature field to obtain spatially averaged melt rates that match the observations than required by the traditional thermal driving parameterizations. From an ice dynamical modelling perspective, this is certainly an important step forward. However, today, models of a wide range of complexity are used to assess the ice-ocean system (see e.g. Asay-Davis et al., 2016 for a summary) and within the scope of these works, it is desirable to evaluate the proposed parameterization also with results from the other end of the spectrum. A couple of circum-Antarctic ocean general circulation models are readily available (even more regional models, some of already coupled with ice models), providing fields of basal melt rates by explicitly resolving the ice shelf cavity circulation. Although not necessarily yielding realistic results everywhere, all of these simulations provide a self-consistent sets of geometrical parameters and ambient ocean temperatures that can be used to scrutinize the validity of the presented plume parameterization. Applying the new parameterization in the context of a fully resolving ocean model framework of the author's choice appears to be a minor additional effort and I highly recommend that such a comparison is added to this publication, as it would substantially aid the validation of the approach (such as its extension to two dimensions) and greatly improve the understanding and integration of the new method within the context of existing works on simulating basal melting.

We agree that evaluating the basal melt parametrization in the context of an ocean general circulation model would be a necessary step for the validation. This is certainly one of the goals that one should work towards. However, we think it would be too ambitious for the current paper and it is not clear if it is as easy as the reviewer claims. The current paper focuses on the evaluation of the melt rates for a fixed present-day temperature field and geometry. Although using an ocean model for determining the ocean temperatures might be less ad hoc than extrapolating the observations, such a model also contains many uncertainties and does not necessarily give more realistic values. In other words,

adding the results of an ocean model would require a detailed description and discussion of the ocean model itself, which would make the current work too extensive, losing focus on the parametrization itself. We hope the reviewer agrees that the problem of modelling basal melt rates on these scales is a very difficult one that requires several careful steps. The first step is showing that a plume parametrization can capture more realistic melt rates than the frequently used parametrizations of the Beckmann & Goosse type, even in its simplest form.

Another issue is that the melt rate maps shown in Fig. 10 work well to assess the improvement over the simplistic scaling of DeConto and Pollard (2016), but do not allow to compare the details of the melt rate map with the observations of Rignot et al. (2013) that is used as a reference (p. 20 l. 10-14). In particular, the truncation of the color scale to melt rates of 2 m/yr excludes a quantitative assessment of the maximum melt rates that can be an order of magnitude larger at some grounding lines with important effect on the ice dynamics.

We agree that this can be improved. The colour scales for the figures have been extended to 5 m/yr and a new figure has been added showing the Rignot data and the difference between these data and the parametrization. Furthermore, we have added zoomed panels for 3 important regions with a logarithmic colour scale, so that the assessment of the maximum melt rates is facilitated.

Within this scope, it is currently also not accessible to the reader, how the tuning points for the ambient ocean temperature field were chosen and by which algorithm the temperature in these points has been optimized to match the area averaged melt rates (see specific comments for details).

The description of the method behind determining the effective temperature field has been improved; see also our replies to the specific comments. We also refer to our reply to Reviewer 2, who has more detailed concerns about the ΔT field. In short, we acknowledge that there are many uncertainties present in our constructed temperature field and we have tried to be more critical throughout the manuscript, including the newly added Discussion section.

Eventually, there are a couple of processes that are known to influence basal melting around Antarctic, but are not captured by this parameterization, with examples being the influence of regionally varying tidal current strength on the boundary layer heat exchange (Maksinon et al. 2011), as well as the enhanced heat exchange due to winds (Hattermann et al. 2014, Dinnimann et al. 2015) as well as intrusions of solar heated summer water near the ice fronts (Hattermann et al. 2012, Stern et al. 2013). Hence, their influence must either be omitted or be included in the fitting of the temperature field, a limitation of the new approach that needs to be discussed.

We have chosen to keep a fairly simple parameterization and a key purpose is to use it in ice dynamical models. Nevertheless we will discuss the simplifications we take in more detail and along the lines suggested by the reviewer. Of the effects mentioned above, mode 3 melting (also mentioned by Reviewer #2)

might be the most important one to discuss, so we added these references to the Discussion.

Also, to some extent the precision of language and figure quality should be improved.

Specific comments

p. 2, l. 11 & 15: What kind of "steady-state" is referred to and what is meant by the "steady nature" of the parameterizations? Does the new parameterization differ in a manner that it is time-varying of some sort?

Thank you for this point. The essential difference between the two parametrizations is rather local vs. non-local. We have changed this in the text.

p. 2, l. 14: Please clarify the ambiguous formulation "geometry below the ice shelves".

This has been changed to "geometry of the ice-shelf base".

p. 2, l. 27-28: How does the referred mechanism in which upward flowing plumes induce inflow of warm water into the cavity relate to the approach presented? To my understanding, this possible feedback on the ambient ocean temperatures is not part of the plume model or the derived parametrizations, opposing the subsequent statement in line 32.

Again a good point. This feedback mechanism is indeed not part of the parametrization, but we prefer to mention it here as a brief summary of the physics in the cavity. To avoid the contradiction in the next paragraph, we have changed "All these physical processes" to "The dynamics of the plume".

p. 3,l. 1-2/p. 8,l. 16ff/p. 22,l. 6-8: With the above general comment in mind, please reflect on the validity of the underlying physics, in particular the non-local dependence on grounding line depth, when extending the plume parameterization to two dimensions.

We hope that the additional lines in Sec. 2.3 and the Discussion section address this issue appropriately.

p. 3, l. 15-18: It is not always clear, which parts of these sections review the results of previous works and which parts are original contributions of the present study. It is mentioned that results are summarized from Jenkins (2011) and Jenkins (2014), while particular advances of the present study are not discriminated in detail. To some extent, the problem may arise, because a central reference of the plume theory is contained in a conference presentation, which is not available for reading. However, explicitly clearly labeling review information and original material of this paper at the beginning of the subsections, should sufficiently mediate this issue.

As explained in our reply to major comment 1, we fully agree that it is an issue that the details of the parametrization have only appeared in a conference contribution. Hopefully, our revision of Sec. 2.2 and Appendix A has resolved this issue. The main original contribution of this paper is the extension of the parametrization to 2-D. We tried to clarify this at the start of the subsections. It might be important to note again that the added analytical solution of the simplified plume equations (Appendix A) has not appeared in Jenkins (2014) and can also be considered a new contribution of the current manuscript.

p. 4, l. 3-5: It would be useful to explain how the ocean current that drives mixing relate to the temperature, hence leading to the non-linearity referred to here. Does this refer to the effect of increased buoyancy by decreased salinity due to more meltwater input for higher temperatures?

We added an additional reference to Holland et al. 2008, where this is explained in full detail. The crucial element is indeed the linear temperature dependence of the ocean current.

p. 6, l. 8: For clarity, mention which simplification is applied, i.e. the assumption of a constant ratio between Gamma_T and Gamma_S.

Thank you for noting this; we have added a sentence that explains it.

p. 6, l. 23 ff.: The derivation of the general melt rate curve appears somewhat fragmented and I am currently not able to retrace its origin based on the information given in this section. In particular, it is not clear how the terms in eqn. 7 to 9 combine into a single expression. Specifically, it is unclear how Jenkin's extension of eqn. 7 looks like and what is described by the universal length scale mentioned on p. 7, l. 21 or how it is used. Also the discussion of the two different melt formulations (p.7, l.21-27) is confusing in the given context, as is the summarizing statement in p. 7, l. 28 (amplitude of which curves?!). Clarity would probably be added by stating in the beginning of the section that Jenkins 2014 has derived an explicit and universal expression of melt rates as function of distance from the grounding line (possibly including eqn. A3) and explaining that the remainder of the section revises the basic ingredients, to sketch how the relationship was obtained but without providing a stringent derivation.

As already explained in the reply to the major comments, we have significantly rewritten this section, taking into account your suggestion to show the form of the parametrization at the start and go into the physical meaning of the terms afterwards.
We hope that the discussion of the two melt formulations make more sense in the revised text. The point is that an empirical expression for Gamma_TS in terms of other quantities is added to complete the procedure for finding the universal length scale. It is one of two extra ingredients for extending the original length scale of Lane-Serff (1995).
The "amplitude of the curves" refers to the plume model results that are scaled by M in order to produce the dimensionless curve in Fig. 2. This has been clarified as well.

p. 8, l. 1 & 2: The plume buoyancy is primarily controlled by salinity, while temperature has only little influence on the density for the given parameter range. Even though this is not stated explicitly, I assume that by parameterizing the plume buoyancy through the temperature difference between the plume and the ambient ocean, an assumption was made on how the temperature difference translates into a salinity difference (i.e. the freshening of the plume is obtained from transforming its respective source water along the melting-freezing mixing line/ Gade line). Does this imply that the general melt rate curve was obtained by assuming that the ambient water at any location along the plume path is the same (or lies along the same Gade line) as at the grounding line where the plume originates? In this case, this would be an important limitation of the theory, which is almost certainly not true for many ice shelves, where different source water types may dominate the ice ocean interaction in different parts of the ice shelf cavity (e.g. different sources of HSSW beneath Filchner-Ronne or the influence of more buoyant surface water near ice shelf fronts).

The revision of Sec. 2.2 and Appendix A should hopefully clarify what the underlying assumptions of the plume parametrization are. It is definitely true that salinity difference are the direct driving mechanism of the plume, and that this is indirectly controlled by the input of meltwater in the plume and, hence, the temperature difference that controls the melting. The theoretical arguments added to Appendix A show how these effects end up being parametrized in terms of the temperature difference alone, indeed under the assumption of constant ambient properties. One can regard this as a limitation, but no parametrization is, of course, able to capture all the physics. In our opinion, this is the simplest way to capture the key physics for producing a net circulation within the ice-shelf cavity. Other effects, such as stratification and different water masses, are certainly important and these should be included and investigated at a later stage. Still, we have added some words about this in the Discussion section.

p. 10, l. 6-12: If my understanding of this algorithm is correct, valid plume paths will also incorporate directions for which the ice base slope reverses somewhere between the given ice shelf point and the respective grounding line since only the local slope and overall grounding line depth are evaluated. What does this imply for the nature and realism of the resulting multitude of valid plume paths?

This is correct. It could be possible to check the slope in between and discard plumes paths for which the slope reverses. But one should keep in mind that this is just a parametrization that is supposed to describe a net circulation. We believe that for the resolution considered here, adding more complexity to the algorithm would have little effect. Yet, it is something that one should look into when considering higher resolutions and more sophisticated plume models, e.g. the 2-D plumes that you have proposed. See also the added lines in the Discussion section.

p. 15, l. 6-9: Should be moved to discussion and supported through proper references.

As discussed in more detail in the reply to Reviewer #2, we believe it was a mistake to mention these models here, because it appears that they are not really used in a proper reference except in our own ice model for testing purposes. These particular sentences have been removed.

p. 16, l. 5: For the given temperature range, the buoyancy of the plume is dominated by salinity differences. Please comment how the uniform salinity field affects the response of the melt rate parameterization (or its inherent ingredients).

Indeed, the main driving mechanism of the plume is the density difference caused by the difference in salinity between meltwater and ambient ocean. As we hopefully clarified in the previous replies and the revision of Sec. 2.2 and Appendix A, the plume buoyancy depends on the difference $S_a - S_i$ (with $S_i$ taken equal to zero) and the temperature differences $T_a - T_f$ and $T - T_f$. The result is that the buoyancy is parametrized entirely in terms of $S_a$ and the temperature difference $T_a - T_f$, which controls the input of meltwater. The bottom line is that the current parametrization does not account for vertical ambient density profiles, and therefore the absolute value of $S_a$ (or more precisely, the relative value w.r.t. $S_i = 0$) only controls the initial buoyancy of the plume at the grounding line, both explicitly and through the freezing point $T_f$. Horizontal variations in $S_a$ around Antarctica are then much smaller than its absolute value and have a very weak effect on the parametrization output. Of course, this would all change if one would include stratification explicitly. We added a line in the text that distinguishes between horizontal and vertical variations and refers to the Discussion section.

p. 18, l. 9-11: More information on this tuning process must be provided. How were the respective temperature differences in the 29 sample points determined? Presumingly, some sort of optimization algorithm has been applied, that involves iterative computation of area averaged melt rates and subsequent adjustment of the individual correction points. How well does this procedure converge towards a unique solution for the given cost function? Why were 29 points used and how have they been allocated and how sensitive is the resulting melt rate map to this particular configuration (from Fig. 10a and Fig. 11a one could get the impression that more spatial detail on the melt rate map correlates with a higher density of correction points)?

We apologize for the confusion here. The 29 sample points have not been determined by a sophisticated algorithm but were chosen by a trial and error. We tried to clarify this in the text. One criterion is to limit the biases near grounding lines resulting from the interpolation between ice shelves (e.g. FRIS). But there are also regions (e.g. Dronning Maud Land) where warm open ocean temperatures are extrapolated into cavities due to the lack of cavity points in the observations. This causes higher values $T_0$ that would overestimate the melt rates (as we found) and require a negative $\Delta T$.
This is not a unique optimal solution (as we added in the text), but merely a necessary exercise in order to obtain the required input field. See also our reply to Reviewer 2. By his suggestion, we have changed the tone of this section,

clarifying that the method is simply an inversion of the basal melt parametrization, yielding ocean temperatures that are not necessarily realistic.

Since ΔT is the result of linear interpolation, we don't expect the melt rates to be very sensitive to the number of sample points. Rather, the melt rates might be sensitive to the *values* of ΔT (see also Sec. 3.1), but this seems a different matter. The current number 29 appears to be around the minimum number of points necessary to obtain the correct *average* melt rates for each shelf group. Adding more points likely has little effect on the average melt rates.

To put it differently, one could also have chosen to "tune" a single effective temperature value for each ice-shelf cavity. The current method aims at obtaining the values from observations and makes the values slightly spatially variable.

The spatial variation in the melt rate maps *within a cavity* is really a property of the parametrization itself, as we showed in Sec. 3.1. Of course, the spatial variation over the entire domain is related to the temperature field as well, but this is the aim of the construction.

We also added a few lines about this to the Discussion section.

p. 18, l. 24-27: It is well known that most of the seawater beneath the FRIS is significantly colder than the surface freezing point. The reason for this is that melt water produced at greater depths is largely recirculated within the cavity and mixes with inflowing water at the surface freezing point, before this interacts with the ice base. Thus a representation of colder ambient water masses would indeed be more realistic in this case.

This might indeed be one of the reasons why the melt rates under FRIS are overpredicted. However, there are two things one should take into account here. First, we only use the annual mean of the temperature observations (we apologize for not clarifying this earlier), so the coldest temperatures are likely not present in the effective temperature field. Second, the plume parametrization in its present form essentially only describes a net circulation with (ideally) a single length scale (hence a single ambient temperature) per cavity. The ambient temperature in this sense represents the net inflow into the cavity and not the temperature of melt water that is produced or mixed locally. We have added some lines in the Discussion section about this. Still, the parametrization clearly does not reproduce the situation under FRIS in a satisfying way. Also note the comments of Reviewer 2 about relaxing the lower bound for the effective temperature.

p. 18, l. 32-33: In fact, the continental shelf temperatures in West Antarctica in Fig. 8a appear rather low compared to observed values well above 0 degC. It would be useful know more about the spatial pattern of basal melt in this region and its comparison compares, in particular if the parameterization is capable of capturing the extremely large melt rates near the grounding lines that are observed here.

Fig. 8a is obtained directly from the WOA13 observations, which do not seem to contain these temperatures above 0 degrees. Maybe the confusion is caused by the fact that we used annual mean data. This is clarified in the caption. Also note

that the ΔT values in West Antarctica (circles in Fig. 8b) are all positive, so the effective temperature used for calculating the melt rates will be higher than the values of $T_0$ here. As far as the spatial detail is concerned, we have added an additional figure to Sec. 3.3 that should clarify this.

p. 21, l. 5-14: Obviously, the new plume parameterization provides significantly improved spatial basal melt patterns compared to the simplistic temperature scaling. However, to this end, it remains somewhat unclear to what extent the obtained spatial pattern of basal melt is a result of underlying dynamics of the parameterization or reflects the optimization of ambient ocean temperatures that were used for the input.

As discussed above, the spatial variation within a cavity is an inherent feature of the parametrization, which is clearly shown in Sec. 3.1. On the other hand, the melt rates can indeed be rather sensitive to the ocean temperatures (also shown in Sec. 3.1).

Thus, a direct comparison with melt rates from a more comprehensive ocean circulation model remains a desired complement to round off the present study. This, to my mind easy achievable extension of the present work would both help to justify the ad hoc extension for the two dimensional case and scrutinize the predictive capacity of the parameterization that is required for using it in a framework of evolving ice geometry or ocean temperature sensitivity studies.

We appreciate the suggestion, but as already mentioned in our reply to the major comments, we are not sure if the suggested additional study with an ocean general circulation model is as easy as the reviewer claims. The current study is meant to show how the plume parametrization in its simplest form behaves when applied to all Antarctic ice shelves. The construction of the necessary ocean temperature input field is indeed an important uncertainty, but the results from an ocean model would not necessarily be more realistic. A carefully developed sensitivity experiment would be required to introduce this coupling, which can be the topic of an entire follow-up study. Nevertheless, some lines about this issue have been added to the Discussion section.

p. 23, l. 3-7: In addition to the prescription of valid plume paths provided in this study, an extension of the one-dimensional plume theory to higher dimensions needs to account for the effects of mass conservation when the dynamical equations are not constrained along a path of uniform width. This will have consequences on the validity of the general melt rate curve that need to be addressed here.

As noted before, this issue is now addressed in Sec. 2.3 and the Discussion section, while parts of the Conclusion section have been toned down.

Figure 4: Use different colors for open ocean and land areas where the relevant fields are undefined.

It turned out to be difficult to add different colours for the land and ocean areas in Matlab. As a compromise, we made both areas white and drew the contours of the borders between the 3 mask areas, as was done for Figs. 10 - 13.

Figure 1: Extend range of melt rates, consider using non-linear color scale.

(Assuming that this refers to Figs. 10 - 13) The color scale has been extended to +/- 5 m/yr so that it becomes easier to compare directly to the Rignot data, which we also added (see the new Fig 12). We considered a non-linear color scale, but this appears to lower the overall contrast between high and low melt rates rather than highlight the highest melt rates. However, we did use it for the new zoomed figure (see below).

Generally, most spatially resolving circum-Antarctic fields are difficult to assess. Consider the use of zoomed inlets to magnify relevant regions.

A new figure was added (Fig. 11) with three panels zooming in on FRIS, West Antarctica and Ross, respectively. The colour scale is logarithmic and essentially only shows the positive regions (negative and zero values have been made white). Coming back to your previous remark, Fig. 11b shows now that the melt rates around Pine Island and Thwaites have an order of magnitude of 10 m/yr. Although these values are quite high, they are not as extreme as the values of 50-100 m/yr found locally in Rignot data. A remark about this was added to the text.

Technical corrections

Generally, the manuscript should be edited to improve the precision of language, including the removal of unnecessary conjectures and filling terms (examples being p. 1, l. 23: "Therefore", p. 4, l. 13: "ultimately", p. 7, l. 13: "hence", p. 8, l. 6: "thus", p. 9, l. 12: "easily", p. 9, l. 14: "Now", p. 11, l. 1: "In summary", p. 13, l. 16: "clearly", p. 18, l. 30: "Clearly", p. 20, l. 7: "obviously", p. 20, l. 20: "immediately") as well as first person narratives which is extensively used throughout the manuscript.

Most of the suggested corrections have been applied, though sometimes we think that these words are useful for aiding the reader. The first-person narratives have been changed only in a few instances. We do not really agree that this form is used too extensively. It is a style that is widely used nowadays in scientific articles, and changing everything to passive voice does not necessarily make the text easier to read or more objective.

p. 1, l. 20: "ocean flow", better use "oceanic heat supply"

Maybe "oceanic heat exchange" is even better?

p. 2, l. 3: "In the view of these issues", imprecise, clarify: "In order to correctly predict the evolution of the ice sheet"

corrected

p. 3, l. 6: "An important part of this work is [the derivation/ the development of] an algorithm"

corrected

p. 3, l. 25 & 24: consistently refer to "sea water" when introducing rho_w and c_w.

corrected

p. 6, l. 2: if only similar, what is the difference between eqn. 1b and 5c.?

"similar" has been changed to "equivalent". The difference would somehow be the salinity (Sw or Sb), but the equation itself is of course the same.

p. 11, l. 6-9: Redundant with p. 9, l. 9-11.

The reference to Bedmap2 on page 9 has been removed.

**Response to Reviewer 2 (Xylar Asay-Davis)**

General comments:

This paper presents a new method for computing basal melt rates below Antarctic ice shelves based on a polynomial best-fit to a non-dimensionalized 1D plume model. The major innovation of this work is the methods for computing the parameters (the slope of the ice draft and the height above the grounding line) for the 1D plume fit based on 2D ice and bedrock topography data. The result appears to be a low-cost, physically based method that can capture the large range of observed mean melt rates for groups of Antarctic ice shelves. Melt rate patterns are also argued to be closer to observations than those from other melt parameterizations, though this is not shown quantitatively.

This work represents a *significant* step forward in bridging the gap between more complete representations of sub-ice-shelf dynamics (e.g. in 3D ocean models or 2D plume models) and simplified, ad hoc melt parameterizations that contained little or no physics. Given the computational expense of ocean and plume modeling and the fact that ice-sheet models are not fully coupled into earth system models, there is a need in the ice-sheet modeling community for parameterizations and simplified models like the one proposed here to improve the realism of forcing from basal melting in response to changes in ocean temperature.

Thank you for this compliment. This is exactly the aim of our work and it is reassuring that this is acknowledged as a significant step.

The main concern I have with the paper involves the discussion around the temperature correction field $\Delta T$ applied to the observed temperatures from World Ocean Atlas (WOA). First, the claim is made that this correction is necessary because of unknown temperatures below the ice shelves, summer biases of observations and the interpolation method used to produce the base temperature field $T_0$ from WOA. No doubt, these factors do contribute to $\Delta T$. But inaccuracies in the plume model itself are also being swept into $\Delta T$. It is reassuring, as the authors state, that the $\Delta T$ is not unrealistically large (as they show it to be for an alternative parameterization), suggesting the strength of the plume-based parameterization. At the same time, the authors' sensitivity study in Sec. 3.1 shows that melt rates can be highly sensitive to changes in temperature that are of the same order as $\Delta T$. This suggests that the evolution of melt rates, even if they are calibrated to match present-day observations, are likely to be highly sensitive to $\Delta T$. This is not shown or discussed in the paper. An application of this parameterization in ice-sheet simulations forced by time-evolving ocean observations or simulation results would require a method for determining $\Delta T$. The paper would benefit from some more discussion of how the authors foresee $\Delta T$ being computed in these scenarios. Namely, what ocean state should be used to compute $\Delta T$? Observations? The initial state of the ocean forcing? How sensitive are the melt rates likely to be to this choice?

The computation of the effective temperature field (more precisely ΔT) is indeed the main uncertainty in our study. We regret that it has led to so much confusion and agree that the method should be explained more clearly and its consequences discussed more critically. First of all, some changes were made to Sec. 3.2 that hopefully describe the method to calculate ΔT more clearly. Furthermore, a new Discussion section was added where we discuss various aspects of the model, including the calculation of ΔT and the temperature sensitivity. A proper temperature sensitivity study of the model is certainly a necessary step, preferably in combination with a dynamical ice sheet model and possibly also with ocean model results.

Another comment is that this paper relies heavily on Jenkins (2014), an EGU talk that does not seem to be available online. This work is cited 9 times, often with the implication that the reader should be familiar with the equations and notation it uses. I happen to have attended this particular EGU session but, as remarkable as the talk was, I can't say I remember the notation in detail. Given how heavily this work relies on Jenkins (2014), it might be worth either providing a permanent URL to that those slides or providing their contents as an appendix here. Otherwise, I would suggest efforts be made to cut down on how often that work is cited and instead to incorporate its findings directly into the paper.

As already mentioned to Reviewer #1, we fully agree that this is a weak point. The reason why we were rather concise in providing details from Jenkins (2014) is that a second paper is planned in which the derivation of the parametrization will be discussed in detail.
Instead of providing a permanent link to the slides, which probably would not completely solve the confusion and is also not free of objections, we have decided to restructure and rewrite Sec. 2.2., also after suggestions by Reviewer #1. Furthermore, we have added some recently found analytical arguments to Appendix A that further explain the form of the different factors in the parametrization.
The important thing to keep in mind is that the current parametrization is merely the result of an empirical study with the plume model. All notation and equations needed to run the model are fully described in the current manuscript. The revision of Sec. 2.2 and additional theoretical arguments in Appendix A further explain the background of the parametrization. Hopefully this clarifies where it all comes from without having to rely on the EGU talk. We hope to formalize the analysis in a future publication.

In addition to the requested discussion above, I recommend a number of minor revisions to the manuscript in the specific comments below. If these are addressed, I would recommend the manuscript for publication.

Specific comments:

In what follows, I will indicate the page number a line number as pp-ll (e.g. 1-1 for page 1, line 1) for simplicity.

2-9: "depend solely on the thickness of the water column beneath the ice shelf" I'm not aware of any parameterizations that use the thickness of the water column only, and the authors don't give a citation for this. Instead, most parameterizations I'm aware of depend only on the depth of the ice-ocean interface (the ice draft), with some parameterizations (e.g. Asay-Davis et al. 2016) *also* using the water-column thickness to taper off melting near the grounding line.

We apologize for the confusion. This type of parametrization is present in our own ice model IMAU-ICE. It makes sense that such a parametrization is not really used in any publication, because it completely lacks a physical basis, giving the exact opposite behaviour from what we describe here (zero melt at the grounding line and monotonically increasing towards the ice front). It seems best to remove the references to such crude models from the manuscript. We have added a reference to the type of parametrization in your paper instead.

2-14: "Due to their steady nature, it is unlikely that the simple basal melt parametrizations contain enough physical details to capture this complex pattern without either significant tuning or extremely detailed ocean-shelf-cavity models." First, I have trouble following what it meant by "their steady nature". Do the authors mean their lack of dependence on external forcing (e.g. ocean temperature)? Or that they assume steady state? Or something else, perhaps?
Second, "simple basal melt parameterizations" by definition will not be "extremely detailed ocean-shelf-cavity models", so I think the sentence needs to be rephrased to differentiate between parameterizations and detailed physical models.

Thank you for this suggestion. We have changed this sentence in such a way that it refers more specifically to the local (steady was indeed not a good term) ice-ocean heat flux on which the "simple parametrizations" are based. Our point was that the local heat flux formulation could be used either by itself as a parametrization or as a boundary condition for a coupled ice-ocean model. But indeed, it is not correct to say that a simple parametrization is used with an extremely detailed physical model.

3-10: "Special attention is given to the construction of an effective ocean temperature field from observations, which is required for providing realistic input data of the temperature within the ice-shelf cavities to the parametrization." This is part of my concern about how the ΔT field is discussed in this paper. I don't disagree that there are biases in the the WOA observations but I do not think the authors demonstrate (or can demonstrate) that the correction leads to a more realistic temperature field. Instead, it is important to acknowledge that the various biases in the WOA observations, the interpolation/extrapolation of those observation, and the plume emulator are all being compensated by tuning ΔT, and this process will not necessarily mean that the resulting effective temperature is more realistic than WOA.

*We fully agree with this view and have rephrased this particular sentence in the Introduction. As discussed below for Sec. 3.2, we have also added some more critical lines to the Discussion section.*

4-3: "The non-linearity arose because the exchange velocity $\gamma_T$ in Eq. (1a) was expressed as a linear function of the ocean current driving mixing across the boundary layer." This is not quite sufficient to have nonlinearity. It is also important that the strength of the ocean current is itself a function of the thermal driving. Maybe add something like, "...across the boundary layer, which is itself a function of the thermal driving".

*This is indeed a necessary requirement. We have added some additional information, as also requested by Reviewer #1.*

5-11, 5-12, 5-14: These are not the standard uses of the symbols $\Gamma_T$ and $\Gamma_S$ (e.g. Jenkins et al. 2010, Jenkins 2011). The exchange coefficients are typically defined to be distinct from the Stanton number, such that St = $(C_D)^{1/2} \Gamma_T$ (and similarly for salt). I would *strongly* recommend switching to this more standard notation or there is likely to be confusion when others try to implement the parameterization. (2c) and (2d) would therefore each need an extra factor of $(C_D)^{1/2}$ and this change would propagate to many other places in the manuscript.

*Thanks for noting this discrepancy. We have corrected it throughout the manuscript.*

6-12: "This simplified formulation can be used together with the prognostic equations (2) by assuming $T_b = T_f$" My understanding of the 2-equation formulation is not that one necessarily assumes that $T_b = T_f$, but rather that a new equation is adopted with the same form as (2) with $T_b$ substituted by $T_f$. We never need to know what $T_b$ is but if one were to need it (e.g. as an ice-sheet boundary condition), it would be different from $T_f$ because of the significantly lower salinity at the interface.

*This is true. We have reformulated this.*

6-15: "Also note the similarity between Eqs. (6) and the simple melt model described by Eqs. (1), the difference being the inclusion of heat conduction and the parametrization $\gamma_T = \Gamma_{TS} U$." I would say an equally (or perhaps more) important difference is the use of the plume T and S instead of the ambient fields.

*This has been added.*

6-19: "...different vertical temperature and salinity profiles of the ambient ocean (Jenkins, 2011, 2014)." My understanding is that the polynomial emulator that the authors use does not account for stratification or vertical variations in T and S. This might be worth mentioning explicitly, either here or better yet in the discussion section. Accounting for T and S profiles that vary with depth as well as time would be a potential improvement for the future that might allow the parameterization to produce Mode 3 seasonal melting (as defined in Jacobs et al.

1992) near the calving fronts of "cold" cavities. This could potentially improve the melt pattern.

We apologize if this was not clearly stated. The parametrization is indeed derived without taking into account vertical variations in T and S (as we now clarified in Sec. 2.2 and Appendix A). The original plume model does allow for vertical profiles. It also remains possible to use varying temperatures also in the parametrization, although this is not consistent with its derivation. We added a sentence at the start of Sec. 2.2 to clarify this and also refer to it in the new Discussion section.

7-9: "three larger length scales" If it is clear which 3 of the 4 length scales is largest, I missed it. It might be best to explicitly state either which 3 are meant or which one is excluded.

To be precise, Jenkins 2011 actually discusses all four length scales mentioned here. The crucial thing here is "beyond the initial zone near the grounding line where the initial source of buoyancy dominates", implying that the third length scale mentioned here can be disregarded. To help the argument, we simply changed "three larger length scales" to "these length scales", which hopefully avoids similar confusion.

7-17: "...the slope affects the entrainment rate, but not the melt rate..." I carefully read the corresponding section of Jenkins (2011) and I think what is shown is that the term in the mass conservation equation for the melt rate doesn't explicitly contain the slope, whereas the term for the entrainment rate does. However, when the equations are solved, the resulting melt rate will depend on the slope, since the plume speed and thermal driving (which contribute to this the melt rate, as shown in Jenkins (2011), Eq. (14)) depend on the slope. So I think the phrase should be changed to something like "...the entrainment rate explicitly depends on the slope, whereas the melt rate does not..."

This is a good point. We have changed it accordingly.

9-9: "In this study, we use remapped data based on the Bedmap2 dataset for Antarctica (Fretwell et al., 2013)," Do the authors perform any kind of a firn correction to the ice thickness, given the assumption of constant ice density in the masking in Table 2? How well does the mask for grounded ice, floating ice and open ocean from Bedmap2 compare with that from the approach in Table 2? The figures suggest that the grounding line might not match well with Bedmap2 (e.g the Amery and deeper parts of the Ross and FRIS) but part of this could be due to the relatively coarse resolution. Without a firn correction, I wouldn't expect the masking from Table 2 to be a good match to the mask provided with Bedmap2.

To our knowledge, no firn correction is adopted in the remapping procedure. This indeed causes a discrepancy between the current mask and the Bedmap2 mask. We think this might not cause big problems for the current coarse resolution (see also below), but it would be important to take into account if one

aims for very accurate simulations with high resolutions. Some words about this have been added to end of this section.

10-8: "the algorithm searches in this direction for the nearest ice-sheet point." This may be obvious to the authors but I think the method used to search for the nearest ice-sheet point should probably be stated explicitly. This part of the algorithm seems like it could potentially be quite slow, particularly at higher resolution. There might also be approaches (e.g. working out from the grounding line, caching the distance to the G.L. in each direction) that could be used to speed up the process. Is this something the authors have considered?

Some lines have been added that should further clarify the searching method. We admit that this is perhaps the simplest and one of the least efficient searching algorithms one could apply. But for the current resolution, the speed of the algorithm appears to be acceptable, also in the (preliminary) runs we have done with the parametrization coupled to our ice sheet model. But for high resolution it could indeed become much slower, and probably some revision of the algorithm would be needed. Some words on the efficiency have also been added to the Discussion section.

11-Fig. 3: "d n = 1/2(H b,1 + H b,2) " why the factor of 1/2 exactly here? Is this because the grounding line is assumed to always fall on the edge halfway between a grounded and a floating point? Also, the reasoning behind the different approaches in (b) and (c) probably deserves a bit more explanation.

This is indeed the assumption behind this additional interpolation. We have found that in some cases the depth difference between the first encountered ice-sheet point and the previous shelf point can be considerable. The additional interpolation is meant to smoothen these discrepancies. For higher resolutions, such an additional step might not be necessary. Also, there might be more sophisticated ways to estimate the location of the grounding line. Furthermore, the reason for having the two different approaches in (b) and (c) is simply to account for both positive and negative basal slopes behind the grounding line, assuming that the basal slope of the ice shelf is always positive. A brief discussion of this step has been added after the description of step 2 of the algorithm.

11-8: Why such coarse resolution (20 km)? Is the algorithm too costly to apply on finer resolution? Have the authors explored whether it still works at, say, 1 km resolution that seems to be needed to resolve grounding line dynamics? I could imagine that issues with noise due to rapid changes in bed slope (e.g. Fig 5b) would be exacerbated by finer resolution.

20 km is the resolution we are aiming for in our ice sheet model (IMAU-ICE). It seemed natural to use this resolution in the current study because we wish to calculate basal melt rates within the framework of future dynamic runs of the ice sheet model on a continental scale. As a crude test, we also applied the parametrization to the original 1 km resolution of Bedmap2. In that case, one resolves more detailed topographic features of the ice shelf base (channels) with

typically higher melt rates "following" these channels. But it is not clear if resolving such detailed features is realistic: they do not seem to appear in the Rignot map and we are also not resolving the 2D ocean circulation, which might be a more important effect. The current algorithm is nothing more than a description of a net circulation within the cavity. This is also mentioned in the Discussion section.

12-Fig. 4: There is a strange rim of floating ice around the whole of Antarctica not present in Bedmap2. Is that an artifact of the remapping scheme that was used? Or the masking scheme in Table 2? Perhaps the calving front is being smoothed out over multiple cells, leading to apparent floating ice where none was present in Bedmap2 before remapping? Also, as mentioned above, the grounded vs. floating mask doesn't look like Bedmap2. Is this just the coarser resolution or has something gone wrong either during remapping or the masking procedure in Table 2?

The strange rim seems to be an artefact of Matlab's contourf routine (i.e. the mask values are interpolated in order to determine the contour lines). We apologize for the confusion here. We changed Fig. 4a such that each pixel is plotted separately. The rim around the coastline has mostly disappeared, though there are still some isolated pixels here and there that are the direct result of our mask applied to the remapped Bedmap2 data. Note that we also changed the colours in the other panels of Fig. 4 as requested by Reviewer #1.

12-2: "The values for the local slope are typically higher both near the grounding line and the ice front, as shown in Fig. 4c." Could the steeper slope the authors see near the ice front be an artifact of smoothing or remapping? The cross sections in Figs. 5a and 6a look quite smooth, even given the 20 km resolution, compared to plots of cross sections from Bedmap2 directly and I have not seen this tendency toward steeper slopes toward the calving front in sections I have taken from Bedmap2.

We have rephrased this sentence somewhat. It does seem true that higher slopes near the ice front are not very typical, neither in the original Bedmap2 data nor our generated slopes. But it does occur in a few places, notably FRIS, though maybe the slopes here are not as high as near the grounding line. Also, please note that Figs. 5 and 6 are not taken from Bedmap2, but from flow line data from Bombosch & Jenkins (1995) and Shabtaie & Bentley (1987).

13-20: "...the discrepancies between the current parametrization and the plume model are largest when the basal slope changes rapidly, because the parameterization responds immediately to the change while the full model has an inherent lag as the plume adjusts to the new conditions." This problem will likely get worse at higher resolution. Might it be worth looking into a certain amount of along-flow smoothing and/or lag of when computing the effective α? Perhaps something for the discussion section.

This is a good suggestion. We have added some lines to the Discussion section.

14-Fig 5, 15-Fig 6: It seems like what is potentially missing here is a comparison with the patterns from Rignot et al. (2013) or another melt rate field inferred from observations. I believe the Rignot data set is available from Jeremie Mouginot on request. The data set from Moholdt et al. (2014) is available from Gier Moholdt on request.

Thank you for this suggestion. We were able to obtain the data from Jeremie Mouginot, but we think it would be more appropriate to add a direct comparison to Sec. 3.3, because Fig. 5 and Fig. 6 are mainly meant to show the general 1-D behaviour of the plume model / parametrization. The full 2-D case including the algorithm of Sec. 2.3 is not yet discussed here. Please see our reply to your comment on Fig. 10 below.

15-4: "This also means that the simplest basal melt parametrizations currently used in some ice-sheet models, namely constant values or monotonic functions of the water-column thickness below the ice shelf, are far from being valid." Again, I don't know of any models using the latter. Perhaps the authors mean "ice draft" instead of "water-column thickness below the ice shelf"?

As mentioned earlier, these parametrizations occurring in our ice model are probably just for testing and not worth mentioning here, especially since they seem to lack a proper reference.

16-3: "but we will assume that the variations in ocean salinity around Antarctica are so small that the pressure freezing point T f is only affected by variations in depth." What about buoyancy (via Δρ)? Wouldn't this also depend on S a? Also, how has S a been eliminated from the universal polynomial (given that it doesn't appear anywhere in Appendix A)? By assumption? Or has it been demonstrated in Jenkins (2014) that variations in S a in the observed range don't have an appreciable effect? Would this still be true if stratification were taken into account?

The newly added analytical derivation in the Appendix should clarify how the buoyancy is actually parametrized in terms of T_a – T_f. We also refer to our reply to Reviewer #1's comment on the same line. What matters most is the salinity difference between the ambient ocean and the meltwater. This indeed depends directly on the absolute value of S_a, as well as indirectly through T_f, but the *horizontal* variations in S_a are assumed small. The factor Q0 appearing in the Appendix is then approximately constant, and this constant essentially ends up in the parameter M0. Stratification would be an entirely different matter that is not captured by the current parametrization. We added some lines in this paragraph and in the new Discussion section that should cover this issue.

16-8: "The best possibility is an interpolated field..." First, I would rephrase "the best possibility" to something more like "We decide a more feasible approach was ...".

This phrase has been changed accordingly.

Second, to me it is odd to speak of interpolating the field into the ice-shelf cavities. It seems that this is what the authors did, but in my own modeling I extrapolate the field into a given cavity with no regard for temperatures in cavities on the other side of Antarctica that might figure into interpolation. Indeed, my colleagues and I have run into trouble when we were too naive in our extrapolation technique, extrapolating warm ocean temperatures from the Amundsen and Bellingshausen Seas under deep parts of FRIS. This does not appear to have occurred using the natural neighbors interpolation approach used here but it might still be worth acknowledging that interpolating temperature between ice-shelf cavities that really don't interact with one another is not really physically realistic.

We agree that one of the main drawbacks of a too simple interpolation technique is the occurrence of biases that are extrapolated from one cavity into another cavity, without taking into account that there might be a continent in between. We certainly encountered this problem as well and, as explained below, this was indeed one of the reasons for choosing the natural-neighbour approach, as it appears to minimize these effects. Still, these biases are still present and we should acknowledge this. Some sentences were added on page 18, which hopefully clarify this point ("One should note that both … agree with the data of Rignot et al. (2013)").

16-16: "requires minimally tuned forcing data to produce realistic output." First, I'm not sure I agree with the assessment that the forcing in "minimally tuned", since the tuning likely has a significant effect on melt rates and their evolution, as discussed above. Second, I'm not sure I would characterize the computation of a field with 29 degrees of freedom (to match 13 mean melt rates) as "tuning", which in my experience refers to attempting to constrain a small number of model parameters rather than a spatially dependent field. Instead, this seems like inversion, much like the approach used to compute basal sliding factors under grounded ice in many ice sheet models. The authors have also characterized this as bias correction, but I do not necessarily agree with that characterization, as I stated above.

"Inversion" indeed seems to be a much better term. Thanks for this suggestion. We have tried to remove the phrases with "minimally tuned" and "realistic" and replace them with clearer and more objective terms.

18-2: "interpolated using natural-neighbour interpolation (i.e. a weighted version of nearest-neighbour interpolation, giving smoother results) to obtain data in the entire domain of interest." Again, it seems strange to interpolate *between* cavities. I guess natural-neighbor interpolation effectively extrapolate into cavities as long as the closest open ocean points is in front of *this* cavity and not some other cavity?

This is indeed what happens, at least if one would use *nearest*-neighbour interpolation. The weighting / smoothening in *natural*-neighbour interpolation probably causes temperatures from one cavity to "leak" into a separate cavity more easily. But this effect still seems much smaller than for e.g. linear

interpolation. We hope the added sentences (mentioned above) clarify that the interpolation method is still not perfect in this sense.

18-6: "this modification is necessary for eliminating biases in $T_0$ caused by the sparse observations and numerical interpolation, and also because the flow dynamics of the ocean are not resolved." This may be the principle but in reality the authors are almost certainly also correcting for shortcomings in the parameterization itself.

This is true and should be mentioned explicitly. We have chosen to add it to the Discussion section.

18-9: "29 carefully chosen points" I think more explanation is needed about how these points were chosen. It appears that they are located at grounding lines near the boundaries between shelves with potentially differing properties. Assuming ΔT is held fixed during an evolving simulation, will values of ΔT in regions that are currently grounded be appropriate as the grounding line moves? What might the limitations be? Again, this may belong in the discussion.

The 29 sample points have been chosen by a trial and error. We tried to clarify this in the text. One criterion is indeed to limit the biases near grounding lines resulting from the interpolation between ice shelves. But there are also regions (e.g. Dronning Maud Land) where warm open ocean temperatures are extrapolated into cavities due to the lack of cavity points in the observations. This causes higher values $T_0$ that would overestimate the melt rates (as we found) and require a negative ΔT.
Regarding the use of ΔT in evolving simulations, we realize that this is much less trivial than we might have anticipated. Some critical lines have been added in the Discussion section. A retreating grounding line is indeed one of the difficult issues here.

18-11: "Note that for technical reasons explained in Appendix A, we have applied a lower limit to the effective temperature equal to the pressure freezing point at surface level." As the authors show in the results section, this seems to be a significant limitation on the approach, particularly when applied to "cold" cavity shelves like FRIS. Perhaps some discussion is warranted on how this restriction might be relaxed in the future, as I will discuss more below.

It is certainly unsatisfying that this rather technical constraint still has a significant effect on the average melt rate for FRIS. We discuss this issue further below and in the Appendix. In the end, it seems to be just a consequence of the crudeness of the polynomial fit, rather than a physical constraint. The last sentence of the next paragraph, which claimed that the constraint has a physical meaning, has also been toned down.

18-18: The whole preceding paragraph for determining ΔT is the most worrisome aspect of the algorithm to me. The choice of ΔT (resulting from the details of how $T_0$ is computed) will potentially determine a lot about how melt rates evolve with time in response to changes in ocean temperature.

We have to admit that it is not entirely certain yet how this procedure should be translated to an evolving ocean, but some suggestions have been added to the Discussion section. With the current computation of $T_{eff}$ we simply aim at obtaining a reasonable reference field that leads to the observed average melt rates. We hope that the critical notes added to the Discussion section address this issue.

18-28: "...yield realistic present-day melt rates for all shelf groups. Therefore, we can conclude that the effective temperature shown in Fig. 8b is a realistic forcing field, at least within the current modelling framework." I don't think the authors can make this statement. The field $\Delta T$ was inverted to yield realistic melt rates for the 13 ice-shelf groups, so the fact that this goal was reached does not suggest that the effective temperature is realistic. A comparison with observations not used to constrain the model would be needed to make such a conclusion. All the authors can conclude here is that their inversion worked as expected (except for FRIS) and that the resulting temperature field looks plausible.

True, this statement was too optimistic. We have removed it.

20-10: "This fact, along with the general melt pattern and the correlation with the surrounding ocean temperature, are in line with observations, e.g. Rignot et al. (2013)." This is by construction, so be careful not to attempt to use this as a validation of the model.

This is a good point. We have reformulated this sentence and also added a direct comparison with the Rignot data as suggested below.

20-11: "However, one should note that the Rignot et al. (2013) melt pattern shows a greater spatial variability, with more patches of (stronger) refreezing occurring between patches of positive melt. The lack of such prominent patches of refreezing in the current parametrization might have different reasons, such as the coarse resolution or the fact that we disregard the details of the ocean circulation within the ice-shelf cavities, as well as effects due to stratification and the Coriolis force." Lack of seasonal variability in T and S and also lack of vertical variability (not just in the sense of stratification, but also in the sense of having distinct water masses at different depths) likely also play a role. For example, this is likely why Mode 3 melting is missing (as mentioned above).

Thanks for this suggestion. We clarified it in the text.

20-15: "All in all, the plume parametrization, together with the effective temperature field, appears to give a realistic melt pattern for Antarctica, showing both a large spatial variability and average melt rates that agree with observations." It is definitely a strength of this parameterization compared with its predecessors that it can capture the range observed melt rates. So I definitely think this deserves emphasis. But, again, this is by construction. It is a good property to have but I think the authors should be careful to state that this is not

a validation of the model, since the observed melt rates were used in the inversion for ΔT.

We have removed this sentence here to make the description of the model results more objective. Also, a proper discussion of the parametrization seems to belong in the Discussion and Conclusion sections.

21-Fig 10: A comparison with the Rignot et al. (2013) melt rates seems like it would also make sense here. As I said, the data should be available on request. Having theme plotted with the same color map would make them much easier to compare.

We have added a new figure (Fig. 12) with both the original Rignot dataset and the difference with the parametrization.

21-13: "minimal tuning" As before, I'm not a fan of this phrasing. What was done was an inversion of a field that can be argued to be within a plausible range. To me, this is neither clearly "minimal" nor clearly "tuning".

This sentence has been rephrased.

23-1: "parametrizations based solely on the local balance of heat at the ice-ocean interface are not able to capture the complex melt pattern…" The authors can rightly claim to have a more broadly realistic melt pattern than these previous studies, with both melting and refreezing. But the authors have not really shown that the complex melt patterns resulting from their parameterization are contributing added realism compared to a simpler pattern with a similar distribution of melting and freezing, and complex patterns are not a goal in and of themselves.

It does not appear that we claim to have a fully realistic melt pattern in this particular sentence. Indeed, compared to the simpler models the new parametrization is more realistic because the simpler models cannot capture refreezing at all. We have tried to be more objective and critical throughout the manuscript, which hopefully solves this issue.

23-14: "…data from observations only need a minimal offset ΔT (between −1.4°C and 0.8°C)" Again, I would suggest a different phrasing than "minimal". "Plausible"? Also, again I think some discussion is needed about how a time-varying T 0 field would be handled. Would ΔT be held fixed? (The authors seem to imply it would be)? How sensitive will the results likely be to ΔT? Over what kinds of time scales might it be reasonable to hold ΔT fixed? How should data from ocean models be applied? Should a ΔT be computed from ocean-model initial conditions to match observed melt rates? Or should ocean observations (e.g. WOA) be used to compute ΔT?

We have again removed the phrase "minimal" and changed it by "plausible", which we find a better alternative. As mentioned above, the Discussion section now addresses the issue of using ΔT in the context of an ocean model.

23-22: "All in all, the presented plume parametrization, together with the constructed effective temperature field, gives realistic results for the present-day basal melt in Antarctica, both in terms of area-averaged values (Fig. 9) and the spatial pattern (Fig. 10a)." I don't think this paper as written has shown that the spatial patterns are realistic, just that the mean values are (by construction) consistent with observations. A more qualitative (or better yet quantitative) comparison of the spatial patterns with Rignot et al. (2013) or with another data set derived from observations would be needed to make the latter assertion here. Alternatively, the claim could be toned down, stating that the pattern is reasonable in a broad sense -- highest melt rates are near the grounding line with refreezing closer to calving fronts.

We think this is a good point, even though we have added the requested comparison of Rignot et al. (2013) in Fig. 12. There are still many features in the Rignot data that the current parametrization cannot capture. We have slightly rephrased this particular sentence.

23-29: "For such simulations, the effective temperature in Fig. 8b, even though it is a constructed field, can prove to be a valuable reference state to which temperature anomalies can be added." As I have said earlier, I think more discussion is needed on how the effective temperature would be used in dynamic ice-sheet simulations using this parameterization. This sentence is a good start but I'd really like to see more.

As mentioned already, this has been added to the Discussion section, with a reference to that section added here.

23-30 "Eventually, coupled ice-ocean simulations (e.g. DeConto and Pollard 2016) can benefit from this approach by comparing ocean-model output to this reference State." Hmm, I hope I'm misunderstanding but it seems like the authors are claiming that their reference temperature should act as a reference field, from which coupled ice sheet-ocean simulations could be validated. If this is not what was intended, please clarify what is meant here. If that *is* what is meant, that's a very bold assertion, given the fact that plume-model biases are also "swept under the rug" during the inversion process for $\Delta T$ used to produce the effective temperature field.

Sorry for the confusion, this is really not what we meant. As noted before, we fully agree that $T_{eff}$ and $\Delta T$ contain model uncertainties as well and should be regarded as part of the model. In fact, $\Delta T$ also includes uncertainties in the observations used to constrain the melt rates. The idea would be to somehow use this information together with the ocean model output to obtain some reasonable evolving temperature anomaly that forces the melt rates. But as we tried to explain in the Discussion this is not completely clear yet, and of course validation of the coupled models is an entirely different matter altogether. We have tried to clarify this final sentence of the Conclusion section.

24-9, 24-11 The numerical constants 3.5e-5 and 10 seem to need units of m and m/yr/°C2 , respectively. Maybe give them names and put them in a table or something, along with 0.545? Also, how were (A2) and (A3) derived, at least in broad strokes?

These parameters have been renamed and added to table 1. The (empirical) derivation of (A2) and (A3) is discussed in Sec. 2.2. We have added some clearer references to this section. Also, the newly added analytical derivation in this appendix should further clarify the background of these equations.

24-15: x 0 is unitless? How was it derived?

It is indeed unitless and determines the transition point between melting and freezing. Like the other parameters, it was derived empirically from the plume model results. We hope this has been clarified in the text.

24-22: Reading Jenkins (2011), it seems like X > 1 means there is no momentum left in the plume, so I would expect setting m = 0 beyond this point would be more realistic than restricting T a to not go below the surface freezing point. By the way, it might be worth discussing why, physically, X should not be allowed to exceed 1.

This is indeed what happens physically. The suggestion to put m = 0 for X > 1 does seem to be a good alternative. We have added some words about this in the Appendix. However, the current melt curve (Fig. 2) does not exactly go to zero, so that putting m = 0 outside of the domain would lead to a discontinuity. Another alternative might be to just constrain X to lie between 0 and 1. All in all, this seems to be a rather technical problem pertaining to the crudeness of the polynomial fit. A proper analysis of the equations as we outlined at the end of the appendix might also lead to a "cleaner" formulation of the parametrization that overcomes this issue entirely.

25: It's not clear to me that Appendix B adds much to the text, other than to emphasize that the algorithms for finding z gl and α are arbitrary. The results are visibly less realistic than for the algorithm presented in the main text and the problems with the constraint on T a seem more severe. If you add Jenkins (2014) as an appendix, you might consider removing this one.

This appendix has been removed. Indeed, it seems better to focus on the current method and improve the description of the parametrization and the discussion of its limitations.

One final comment. Since your paper was submitted, an alternative method for parameterizing basal melt by Reese et al. (2017) has been submitted and is also on The Cryosphere Discussions (see full citation below). You might consider discussing how their approach compares with yours, including what the strengths and weaknesses of each approach might be for adoption in a full ice-sheet simulation. I don't mean this as shameless self promotion even though I'm a coauthor. I really do think these papers are highly relevant to one another.

Thank you for pointing our attention to this other paper. It is certainly interesting and useful to briefly discuss the differences and similarities between these two methods. Though it is difficult to fully grasp the behaviour of the PICO model just from reading the paper, it appears that both this model and our parametrization essentially describe the same physics: a net circulation within the cavity that requires an additional algorithm to be extended to 2-D. Judging from the reviewer comments of the other manuscript, both methods are also similar in terms of the physical processes that are not taken into account (e.g. stratification, Coriolis force etc.). If this is correct, then the main difference with our parametrization appears to be that it is a box model rather than based on a smooth fit function of the governing equations. It probably goes too far to add a detailed discussion of advantages and disadvantages of box models compared to parametrizations that are closer to the governing equations. In the end both are only approximations and both could give useful results. Of course, a lot depends also on the implementation and efficiency. Though one could argue that ultimately a systematically derived analytical solution/approximation of the plume equations might be preferred (we are not claiming that we have done this, but it would be an interesting next step). A few lines about this have been to the Discussion section.

Typographic and grammatical corrections:

1-1: It is a very minor thing but I would suggest another word or phrase besides "decline", which implies to me that the AIS used to be better than it is now. Perhaps "...major factor in the decline in volume of the Antarctic Ice Sheet" or "...major factor in mass loss from the Antarctic Ice Sheet". Also, "Ice Sheet" should be capitalized in this case.

corrected

2-13: "...a complex spatial pattern, which depends heavily on both the geometry below the ice shelves and the ocean temperature." I do not think the observations demonstrate this, though I agree that it is the case. Perhaps rephrase something like "...a complex spatial pattern, which can be inferred to depend heavily on..."

corrected

2-17: "within a single ice-shelf cavity" Maybe consider "within individual ice-shelf cavities" instead, since we aren't talking about one specific ice-shelf cavity but rather about any one of several ice-shelf cavities in isolation.

corrected

6-30: "The first governing length scale is associated with the pressure dependence of the freezing point that imposes an external control on the relationship between plume temperature, plume salinity and the melt rate, which is determined by the temperature relative to the freezing point. " I don't

follow this sentence. What is determined by the temperature relative to the freezing point? Perhaps try to reword or break this sentence into 2.

The final clause has been removed because it indeed adds little clarification, especially since we have added more equations that explicitly show this relation.

10-17: "found values" should be "values found"

corrected

23-17: "The latter behavior is also apparent in..." It's not entirely clear to me what is meant by "the latter behavior". I guess it is the low sensitivity of the melt rate to changes in temperature, though this is not explicitly the behavior described in the previous sentence.

This sentence has been rephrased.

[revised manuscript text omitted]

---

## Author Comment (AC2) · 15 Sep 2017

Please find our response to all reviewer comments in the attached PDF file. The changes in the manuscript (generated by latexdiff) can be found at the end of the PDF file.

Please also note the supplement to this comment:
https://www.the-cryosphere-discuss.net/tc-2017-58/tc-2017-58-AC2-supplement.pdf

---

## Author Response (AR2)

**Author response to minor revision**

*We thank both reviewers and the editor for their positive opinion of our revised manuscript. The remaining minor comments have been addressed accordingly, as shown in the appended marked-up version of the manuscript. Below we add a brief explanation to some specific comments, page numbers referring to the old marked-up version.*

**General comment**

*Unfortunately, an error was found in Eq. (A1c). In order to obtain the simplified solution in Eq. (A2), also the term containing the melt rate and T_f in the heat balance equation was neglected. This has been corrected.*

**Referee #1: Tore Hattermann**

The general melt rate curve / Sec. 2.2

*We agree with the editor that the discussion of the physical meaning of the various length scales could be useful for non-specialists. Therefore, we prefer to keep this explanation in Sec. 2.2. and reserve the Appendix for the detailed description of the parametrization.*

The algorithm to determine the local slope and grounding line depth also includes (unphysical) paths along reversed slopes of the ice base (could either be included in p. 12, line 15 and following or in the discussion).

*This has been added to the Discussion section.*

Although stratification being mentioned as an uncertainty, its role should be clarified ....

*Some lines were added in the Discussion section, explicitly referring to the different modes of basal melt and the breakdown of the current non-local relation with the grounding-line depth.*

On page 25, line 15, I suggest rephrasing to "with ocean temperatures being constructed in a plausible way". Especially beneath FRIS, temperatures are certainly not plausible, they are known to be below -2 degC for most of the cavity (various Nicholls et al. papers), which is worth telling the reader on p. 21, line 5 and following.

*The sentence on P25L15 has been rephrased. A reference to Nicholls et al. (2009) concerning the sub-shelf temperatures has been added to P21L5.*

**Editor (Kenny Matsuoka)**

P4L27, P4L32, and elsewhere: Change "ocean water" to "seawater" (or vice versa to keep it consistent throughout the paper).

*We chose to change everything to "ocean water", because the term "ocean" occurs in more instances in the manuscript.*

P7L14-15: Small letter L is defined here as "a temperature- and geometry-dependent length scale". However, if I understand correctly, it is referred as "the universal length scale l" at P8L23. Because many length scales are discussed in the section 2.2, it is confusing for me. When you define l at P7L14, can you also mention why it has a universal feature?

*We understand the confusion and tried to improve the transition between the paragraphs at P8L23. The main point of the preceding paragraphs is to explain where the main part of length scale l in (7) (namely the factor (T_a – T_f) / lambda3) comes from. The following paragraphs then mention the other ingredients necessary to scale the plume model results on a universal melt-rate curve. Hopefully this clarifies which length scales are meant and that the length scale in (7) is the one used in the parametrization.*

P7L28: I think that equation numbers "(9) and (8)" are incorrect. Maybe they are associated with LateX system bug but please double check equation numbers.

*This was indeed a bug/mistake in latexdiff and should only apply to the previous marked-up version where equations with the same equation label were simultaneously crossed out and added elsewhere. The final tex file should not have this issue and all equations numbers seem correct.*

P10L6: "The elevation of the upper ice surface Hs". H is used for ice thickness and elevations, whereas z is defined as depth. And later, "d" is used to define grounding-line depth. It is confusing.

*We understand the point made by the editor. The symbols for the "intermediate" grounding-line depths have been changed to $z_n$. However, the other symbols are commonly used within their respective subfields: Hs, Hi and Hb are common in ice-sheet models (e.g. De Boer et al. 2015, The Cryosphere 9), whereas D has also been used for the plume thickness in e.g. Jenkins (1991) and Jenkins (2011). Therefore, we would suggest keeping these symbols to be consistent with these papers. Changing them could be equally confusing for other readers.*

P10L6 and P10L10, subscript b is used to refer both bedrock and the lower surface of the ice shelf, which can be confusing. Also, "bedrock" is not accurate; seafloor is often covered with sediments, and I think that the authors want to refer "seafloor", instead of "bedrock".

*As mentioned above, $H_b$ is a common symbol in ice-sheet models for what is commonly referred to as "bedrock elevation", without distinguishing between the solid ground under the ice sheet and the seafloor under the ice shelves. We thank the editor for mentioning the inaccuracy and added "seabed" to the specification of this quantity, which is now called $H_{bed}$ to avoid confusion with $z_b$.*

P18L10: Is the salinity correct? You say 34.6 here, but you used 34.65 for the 1D experiments (P15L11).

*These salinities are both correct, our apologies for the confusion. There seems to be a discrepancy between the two different scripts used to produce the results in Sections 3.1 and 3.2, respectively. The salinity difference of 0.05 will have a negligible effect on the results, changing the pressure freezing point by approximately 0.003 degrees.*

P24L2: do you think Rignot's estimate of 100 m/a is accurate? I accept your general message here that models predict basal melt smaller than the observation but doubt validity of this extreme number.

*It is difficult to say if this value is accurate, but in the data we received from Jeremie Mouginot these extreme values only seem to occur locally and close to the grounding line of e.g. Pine Island Glacier. The area-averaged value for this ice shelf calculated by Rignot et al. (2013) is 16.2 m/yr, showing that most of the area has basal melt rates that are less extreme. We added the average melt rates of Pine Island and Thwaites to this sentence to highlight that the 100 m/yr is only a local value.*

Table 1: third row in the constant parameters. The subscript should be "TS", not "TS0".

*Sorry for the confusion. Gamma_TS0 is actually a different quantity than Gamma_TS, because Gamma_TS is a non-constant function containing, among others, the constant parameter Gamma_TS0, which can be seen as an initial guess. This is already explained in Sec. 2.2 and Appendix A (eq. A10), but we tried to clarify it, also changing the order and description in Table 1.*

Figure 2:

*Different colours are now used for melting and refreezing, as requested, with specification in the caption.*

Figure 3:

*The requested labels (with "Bed" instead of "Seafloor") have been added.*

Figure 5 & 6:

*As requested, these two figures have been combined into one two-column figure, with the results of each ice shelf vertically aligned. The figures and caption have been modified accordingly, as well as references in the text. The brown area denoting the seafloor has been removed because it only had an aesthetic purpose. The approximate positions of the flow lines have been indicated in Fig. 6 (previously Fig. 7).*

Fig. 10 (previously Fig. 11):

*It proved difficult to add a color mask, but we decided to add labels denoting "Ice" and "Ocean" areas instead.*

[revised manuscript text omitted]